# Multi-modal meta-analysis of cancer cell line omics profiles identifies ECHDC1 as a novel breast tumor suppressor

Alok Jaiswal[1,*,†] (iD), Prson Gautam[1], Elina A Pietilä[2] (iD), Sanna Timonen[1,3,4], Nora Nordström[1], Yevhen Akimov[1] (iD), Nina Sipari[5], Ziaurrehman Tanoli[1], Thomas Fleischer[6], Kaisa Lehti[2,7,8] (iD), Krister Wennerberg[1,9] & Tero Aittokallio[1,6,10,11,**] (iD)

## Abstract

Molecular and functional profiling of cancer cell lines is subject to laboratory-specific experimental practices and data analysis protocols. The current challenge therefore is how to make an integrated use of the omics profiles of cancer cell lines for reliable biological discoveries. Here, we carried out a systematic analysis of nine types of data modalities using meta-analysis of 53 omics studies across 12 research laboratories for 2,018 cell lines. To account for a relatively low consistency observed for certain data modalities, we developed a robust data integration approach that identifies reproducible signals shared among multiple data modalities and studies. We demonstrated the power of the integrative analyses by identifying a novel driver gene, ECHDC1, with tumor suppressive role validated both in breast cancer cells and patient tumors. The multi-modal meta-analysis approach also identified synthetic lethal partners of cancer drivers, including a co-dependency of PTEN deficient endometrial cancer cells on RNA helicases.

**Keywords** cancer driver; data integration; multi-omics data; reproducibility; synthetic lethality

**Subject Categories** Cancer; Methods & Resources; Molecular Biology of Disease

**Mol Syst Biol. (2021) 17: e9526**

## Introduction

Cancer cell lines have immensely served the purpose of expanding our understanding of cancer biology, and also accelerated the process of developing new targeted therapeutics (Gillet *et al*, 2013; Ben-David *et al*, 2018). Analogous to the patient tumor profiling efforts (Zehir *et al*, 2017; Hutter & Zenklusen, 2018), high-throughput "omics" technologies have enabled a deep molecular and genetic characterization of large panels of human cancer cell lines. As a result, a high-resolution molecular portrait of the genome (Shankavaram *et al*, 2009; Barretina *et al*, 2012; Daemen *et al*, 2013; Klijn *et al*, 2015; Iorio *et al*, 2016; Marcotte *et al*, 2016; Ghandi *et al*, 2019), transcriptome (Shankavaram *et al*, 2009; Barretina *et al*, 2012; Daemen *et al*, 2013; Klijn *et al*, 2015; Iorio *et al*, 2016; Marcotte *et al*, 2016; Ghandi *et al*, 2019), proteome (Gholami *et al*, 2013; Lawrence *et al*, 2015; Coscia *et al*, 2016; Roumeliotis *et al*, 2017; Lapek *et al*, 2017; Nusinow *et al*, 2020), epigenome (Shankavaram *et al*, 2009; Barretina *et al*, 2012; Daemen *et al*, 2013; Iorio *et al*, 2016; Ghandi *et al*, 2019), and phospho-proteome (Shankavaram *et al*, 2009; Barretina *et al*, 2012; Daemen *et al*, 2013; Marcotte *et al*, 2016; Ghandi *et al*, 2019) across diverse panels of cancer cell lines is becoming available. Complementing these efforts, functional and phenotypic profiling of cancer cell lines using loss-of-function screens (Koh *et al*, 2012; Aguirre *et al*, 2016; Marcotte *et al*, 2016; Tsherniak *et al*, 2017; Wang *et al*, 2017; Meyers *et al*, 2017; McDonald *et al*, 2017; Behan *et al*, 2019) and small-molecule drug response profiling has also been carried out by several laboratories

---

1 Institute for Molecular Medicine Finland (FIMM), Helsinki Institute of Life Science (HiLIFE), University of Helsinki, Helsinki, Finland
2 Individualized Drug Therapy, Research Programs Unit, University of Helsinki, Helsinki, Finland
3 Hematology Research Unit Helsinki, University of Helsinki and Helsinki University Hospital Comprehensive Cancer Center, Helsinki, Finland
4 Translational Immunology Research Program and Department of Clinical Chemistry and Hematology, University of Helsinki, Helsinki, Finland
5 Viikki Metabolomics Unit, Helsinki Institute of Life Science (HiLIFE), University of Helsinki, Helsinki, Finland
6 Department of Cancer Genetics, Institute for Cancer Research, Oslo University Hospital, Oslo, Norway
7 Department of Microbiology, Tumor and Cell Biology, Karolinska Institutet, Stockholm, Sweden
8 Department of Biomedical Laboratory Science, Norwegian University of Science and Technology, Trondheim, Norway
9 Biotech Research & Innovation Centre (BRIC) and Novo Nordisk Foundation Center for Stem Cell Biology (DanStem), University of Copenhagen, Copenhagen, Denmark
10 Department of Mathematics and Statistics, University of Turku, Turku, Finland
11 Oslo Centre for Biostatistics and Epidemiology (OCBE), University of Oslo, Oslo, Norway
*Corresponding author. Tel: +1 617 7089255; E-mail: ajaiswal@broadinstitute.org
**Corresponding author. Tel: +358 50318246; E-mail: tero.aittokallio@helsinki.fi
†Present address: The Broad Institute of MIT and Harvard, Cambridge, MA, USA

(Barretina *et al,* 2012; Garnett *et al,* 2012; Basu *et al,* 2013; Iorio *et al,* 2016; Gautam *et al,* 2016).

Recently, the reproducibility of pre-clinical data and findings from the high-throughput profiling studies in cancer cell lines has been extensively investigated due to concerns of inconsistency between laboratories (Haibe-Kains *et al,* 2013; Haverty *et al,* 2016; Mpindi *et al,* 2016; Jaiswal *et al,* 2017; Niepel *et al,* 2019; Gautam *et al,* 2019; Dempster *et al,* 2019). In particular, the consistency of high-throughput drug sensitivity phenotypes has been questioned and re-analyzed by multiple groups (Haibe-Kains *et al,* 2013; Mpindi *et al,* 2016; Bouhaddou *et al,* 2016; Geeleher *et al,* 2016; Safikhani *et al,* 2016). Similarly, functional gene dependency estimates based on genome-wide RNAi screens have been reported to be relatively inconsistent, mainly due to the off-target effects inherent to the RNAi technique (Jaiswal *et al,* 2017), while the CRISPR-based genome-wide knockout screens have been shown to provide fairly good agreement (Dempster *et al,* 2019). Furthermore, given the nature of cell culture techniques by which cell lines are passaged and seeded from a small population, it is likely that even identical cell lines accumulate genomic variability and differences in their clonal composition from one research laboratory to another (Ben-David *et al,* 2018). This type of evolutionary variability introduces an additional level of complexity which influences the repeatability of phenotypic profiles, research findings, and biological conclusions (Gillet *et al,* 2013; Ben-David *et al,* 2018).

In addition to the experimental issues, it is also known that the technology platform being used for high-throughput measurements as well as the computational methods used in their data processing are important contributors to the consistency of research results (Mpindi *et al,* 2016; Haverty *et al,* 2016). Many of the technology platforms for molecular profiling are still in a nascent stage of development, and thus, the resulting data are error-prone, even when using state-of-the-art data processing and normalization procedures. Moreover, there exist major differences in the set of cell lines profiled between research sites, hence making the comparisons and integration of profiling data intricate and biased due to missing omics profiles for certain cell lines. Therefore, there is a need for a comprehensive and quantitative analysis of the relative consistency of molecular, genetic, and phenotypic characteristics of cancer cell lines from different research laboratories and technology platforms, with the aim to improve the robustness of the conclusions drawn from these studies.

In this study, we first performed a systematic statistical meta-analysis to estimate the reproducibility of various types of molecular profiles, or "modalities", of cancer cell lines. Subsequently, we built on these analyses, with the aim to identify robust and reproducible gene signatures with consistent evidence across multiple research laboratories and data modalities, and hence, more likely to be implicated in cancer. To do so, we developed a novel multi-omics integrative approach for jointly analyzing heterogeneous datasets generated from multiple studies for multiple modalities, which also accounts for differences in the panels of cell lines profiled between the research sites. Using 53 omics datasets from 12 research laboratories encompassing 9 data modalities for 2,018 cancer cell lines, we demonstrate how our data-driven approach is able to identify well-known driver genes of established relevance in breast cancer, as well as novel targets for therapeutic opportunities. We expect the comprehensive multi-modal data resource and the integrated approach will provide useful guidelines on how to integrate heterogeneous data from multiple omics studies, which may lead to novel biological discoveries in various cancer types.

## Results

### Compilation of available omics data modalities of cancer cell lines

We processed, re-analyzed, and harmonized curated datasets of various data modalities for cancer cell lines that were originally generated at 12 research sites (see Methods and Protocols, Appendix Figs S4 and S5). We focused on analyzing data modalities available as quantitative measurements for various attributes of protein-coding genes, including methylation, mutational status, copy number alteration status, gene and protein expression, and protein phosphorylation. We further considered functional profiles such as gene dependency estimates from loss-of-function screens and drug response measurements, and calculated an additional functional data modality by transforming drug response profiles to target protein addiction signatures (Fig 1A). Overall, a given cell line had maximally omics data across nine modalities generated at one of the research laboratory sites (Fig 1B). The number of cell lines profiled for a given data modality ranged from 171 (protein expression) to 1,689 (mutation profiles, Fig 1C), making the data integration challenging for the meta-analysis (Dataset EV9).

For instance, the National Cancer Institute (NCI) program (NCI-60) has extensively characterized a panel of 60 cancer cell lines

**Figure 1.** **Overview of data modalities and their consistency.**

A  Overview of datasets, research sites, and molecular modalities that were analyzed in the study.

B  The number of cell lines having data for the 9 types of modalities that were analyzed in the study.

C  The number of cell lines for which data were available for each of the modality types.

D  Correlation of the different types of data modalities of cancer cell lines profiled at multiple research sites. Spearman's correlation was calculated between identical cell lines for the shared set of genes that were overlapping between any two datasets. Gray distributions show the correlation of non-identical cell lines between datasets from various research sites for comparison. $N_g$ and $N_c$ indicate the median [ranges] of the number of genes and cell lines, respectively, across the pairwise comparisons made between datasets from different research sites. More details on the breakdown of $N_c$ and $N_g$ by data modality and research site is available in Appendix Figs S5B and S6, respectively, and Appendix Fig S7C shows the correlation *P*-values adjusted for the sample size ($N_g$). For the point mutation view, only those genes having mutations with an associated functional consequence were considered in the Matthews correlation analysis. Only those datasets for which the mutation profiles were obtained using the whole-exome sequencing technology were considered in this study. Horizontal lines mark the median value. Target addiction score (TAS), Drug Sensitivity Score (DSS), Gene dependency (FUNC), protein phosphorylation (PHOS), protein expression (PEXP), gene expression (GEXP), copy number variation (CNV), point mutation (MUT) and methylation (METH) profiles.

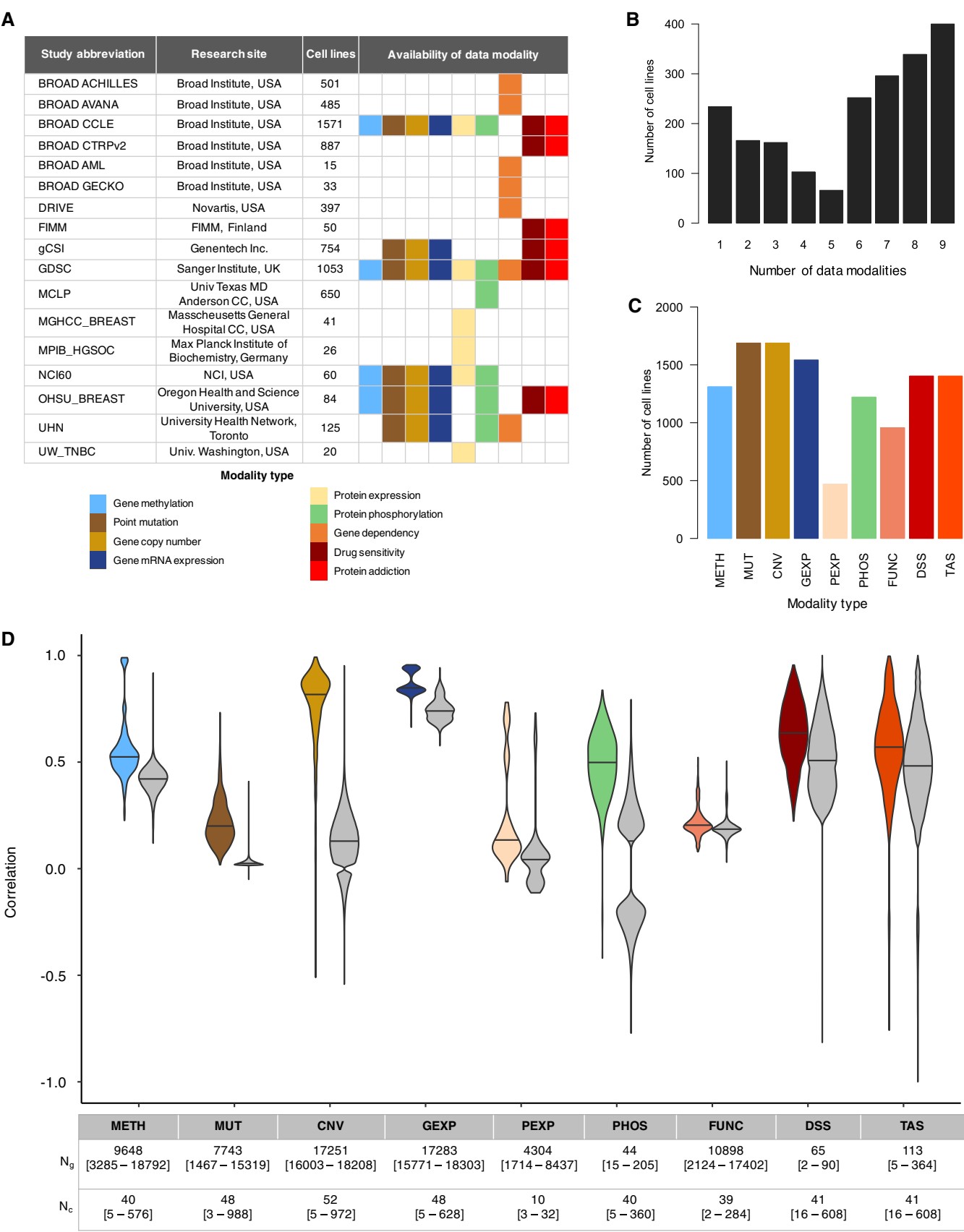

**Figure 1.**

representing nine different cancer types (Shankavaram *et al*, 2009). In contrast, more large-scale efforts such as the Genomics of Drug Sensitivity in Cancer (GDSC) (Yang *et al*, 2012; Garnett *et al*, 2012; Iorio *et al*, 2016; Roumeliotis *et al*, 2017; Behan *et al*, 2019), Cancer Cell Line Encyclopedia (CCLE) (Barretina *et al*, 2012; Basu *et al*, 2013; Seashore-Ludlow *et al*, 2015; Tsherniak *et al*, 2017; Meyers *et al*, 2017; Ghandi *et al*, 2019), and the Genentech Cell Screening Initiative (gCSI) (Klijn *et al*, 2015) have characterized approximately 1,000, 1,500 and 675 cancer cell lines, respectively, representing a wide variety of cancer types. These studies have also performed phenotypic profiling of drug sensitivity against a library of small molecules (Fig 1A). Likewise, the DepMap project has systematically characterized the functional-genomic landscape of ~ 500 cancer cell lines using genome-wide RNAi screens and several versions of genome-wide CRISPR-Cas9 loss-of-function libraries (Aguirre *et al*, 2016; Tsherniak *et al*, 2017; Meyers *et al*, 2017; Wang *et al*, 2017; Fig 1A). MD Anderson Cell Lines Project (MCLP) has additionally profiled protein phosphorylation levels using reverse phase protein arrays (RPPA) for 340 unique cancer signaling related proteins in ~ 650 cancer cell lines (Li *et al*, 2017), and at CCLE for 174 proteins in ~ 900 cell lines (Appendix Figs S5 and S6).

Complementing these large-scale pan-cancer programs, we also re-analyzed datasets from more targeted efforts that have profiled cell lines of a specific cancer type; these smaller-scale studies were included in this meta-analysis to increase the information content on selected tissue lineages. Specifically, multi-modal datasets were generated at the University Health Network (UHN) (Koh *et al*, 2012; Marcotte *et al*, 2016) at Toronto and the Oregon Health and Science University (OHSU; Daemen *et al*, 2013; Costello *et al*, 2014) studies for > 80 breast cancer cell lines. Furthermore, proteome-scale expression levels in breast, ovarian, and colorectal cancer cell lines have been generated using mass spectrometry (MS) at the University of Washington (UW_TNBC) (Lawrence *et al*, 2015), Massachusetts General Hospital Cancer Center (MGHCC_BREAST) (Lapek *et al*, 2017), Max Planck Institute of Biochemistry (MPIB_HGSOC) (Coscia *et al*, 2016) and the Sanger Institute (GDSC) (Roumeliotis *et al*, 2017); however, the total number of cell lines profiled for MS protein expression was only 171. Further, an in-house drug sensitivity profiling dataset of > 50 pan-cancer cell lines generated at the Institute for Molecular Medicine Finland (FIMM) was also utilized in the study (Mpindi *et al*, 2016; Smirnov *et al*, 2018; Gautam *et al*, 2019).

To enable the meta-analysis between studies, we only considered datasets that were generated in a sufficiently larger panel of cell lines ($n > 10$) and therefore excluded datasets below the threshold. In statistical analyses, we assumed that the same cell lines profiled at each site were cultured independently. All together, we processed and re-analyzed 53 datasets, encompassing nine modalities generated at the 12 research study sites. In total, we analyzed data for 2,018 cancer cell lines having measurements for at least one of the data modalities. A substantial proportion of cell lines ($n = 1,047$) had data available for $\geq 6$ modalities, thus serving as a comprehensive resource for further analyses (Fig 1B and C). Even though most cell lines had data available from multiple sites, there were ~ 700 cell lines that had data available from only one study site (Appendix Fig S5A and B). We reasoned that the substantial overlap between cell lines across multiple molecular layers between more than two sites provides a solid basis to perform a quantitative

assessment of the reproducibility of the multiple modalities of cancer cell lines, which allowed us to fine tune the parameters for a robust integration of data modalities from multiple research sites (see Methods and Protocols).

## Reproducibility of molecular modalities of cancer cell lines from multiple sites

We performed a systematic correlation analysis to evaluate the consistency of gene-level quantitative measurements of the various data modalities from identical cell lines profiled across different research sites. Overall, we observed a wide variation in the degree of agreement between the research laboratories (Fig 1D, Appendix Fig S7A). Consistent with previous observations (Haibe-Kains *et al*, 2013; Klijn *et al*, 2015; Haverty *et al*, 2016), copy number variation (CNV) profiles and transcriptomic profiles of cell lines were highly correlated between different study sites (Spearman's correlation $r_{CNV} = 0.76$ [−0.51 to 0.99] and $r_{GEXP} = 0.87$ [0.66 to 0.96]), in contrast to mutational profiles ($r^2_{MUT} = 0.22$ [0.02–0.73]) (Fig 1D). We observed a considerable range of variation in the pairwise correlation of CNV profiles between different sites, suggesting that the cell lines with poor agreement may have undergone clonal divergence during cell culture. For all the data modalities, we observed that non-identical cell lines from same tissue types had slightly elevated correlation compared to non-identical cell lines from different tissues, but the opposite for the PEXP modality (Appendix Fig S7A). Moreover, the cell lines that had weaker correlation in CNV profiles also tended to have weaker correlation in MUT profiles (Appendix Fig S7B, Dataset EV10).

In general, methylation profiles of cell lines, corresponding to methylation levels of CpG sites located at transcription start sites of genes, were moderately consistent ($r_{METH} = 0.56$ [0.23–0.99]) (Fig 1 D). Likewise, protein-level phosphorylation profiles were only modestly reproducible between different sites, suggesting that the targeted reverse phase protein array (RPPA) technique is relatively noisy ($r_{PHOS} = 0.49$ [−0.42 to 0.84]). The correlation of the global proteome expression profiled with MS was even lower, on average, and it also exhibited a wide range of variability in the relatively small number of available breast and ovarian cancer cell lines ($r_{PEXP} = 0.29$ [−0.09 to 0.78]). However, when considering the dimension of the profiles (median of 44 for PHOS, and 4,304 for PEXP), the global protein expression correlations had higher significance on average (Appendix Fig S7C). As observed previously (Mpindi *et al*, 2016; Haverty *et al*, 2016), we also found that the reproducibility of drug sensitivity profiles between sites was moderately high ($r_{DSS} = 0.63$ [0.22–0.95]), similar to the reproducibility of TAS profiles ($r_{TAS} = 0.56$ [−0.75 to 0.99]). In contrast, gene dependency estimates based on loss-of-function RNAi and CRISPR screens exhibited rather poor reproducibility ($r_{FUNC} = 0.21$ [0.08 to 0.52]).

Given that the distributions of data modalities are quite different, the correlation estimates (either Spearman's or Matthew's coefficient) are not directly comparable. To set a reference point for the pairwise comparisons, we further estimated the correlation of non-identical cell lines between the different studies (Fig 1D, gray distributions). This analysis is also useful for assessing the expected baseline correlation of different modality types. As expected, the average correlation of mRNA expression profiles of even non-identical cell lines was generally high ($r_{GEXP} = 0.75$), suggesting that the

transcriptomic landscapes are quite similar across cancer cell lines and tissue origins (Fig 1D, Appendix Fig S7A). Compared to the average correlation of non-identical cell lines, we observed a 1.17-fold increase in the mean correlation of the identical cell lines for gene expression profiles ($P < 10^{-10}$, Wilcoxon test). We observed a similar fold increase for methylation (1.33-fold, $P < 10^{-10}$), gene dependency (1.13-fold, $P < 10^{-10}$) and drug response profiles (1.59-fold, $P < 10^{-10}$). In contrast, a much higher fold increase in the correlation of identical cell lines was observed for CNV (5.9-fold, $P < 10^{-10}$), point mutation (7.8-fold, $P < 1.0 \times 10^{-10}$), protein phosphorylation (27.9-fold, $P < 10^{-10}$), and protein expression profiles (12.2-fold, $P < 1.5 \times 10^{-08}$).

### Reproducibility of technology platforms used to generate the data modalities

Correlation analysis implied the existence of bi-modal distribution of consistency estimates for some of the data modalities (Fig 1D, Appendix Fig S8A–I). We therefore further stratified the correlation analyses separately for each of the experimental technologies to investigate whether the observed variability could be explained by the platform used to generate the data. We observed a significantly higher reproducibility of methylation profiles between studies generated using the Illumina 450K BeadChip, compared to the correlation of methylation profiles of datasets generated using Bisulfite sequencing ($r_{\mathrm{METH}} = 0.97$ for 450K/450K vs. $r_{\mathrm{METH}} = 0.51$ for 450K/Bisulfite, $P < 1.0 \times 10^{-10}$, Wilcoxon test) (Appendix Fig S8J). As expected, a higher correlation was observed between those studies in which the transcriptomic profiles of cell lines were measured using RNA sequencing compared to the microarray-based profiles ($r_{\mathrm{GEXP}} = 0.93$ for RNA-seq vs. $r_{\mathrm{GEXP}} = 0.84$ for arrays, $P < 1.0 \times 10^{-10}$, Wilcoxon test) (Appendix Fig S8). Similarly, RPPA-based protein phosphorylation profiles were slightly better correlated with studies based on RPPA than with MS-based phospho-proteomic profiles ($r_{\mathrm{PHOS}} = 0.45$ for MS/RPPA vs. $r_{\mathrm{PHOS}} = 0.49$ for RPPA/RPPA, $P = 0.03$) (Appendix Fig S8K). Likewise, drug sensitivity screens and TAS profiles based on CellTiter-Glo (CTG) assay were significantly more correlated in comparison to those based on fluorescent nucleic acid stain probes such as Syto60 ($r_{\mathrm{DSS}} = 0.72$ for CTG/CTG vs. $r_{\mathrm{DSS}} = 0.55$ for CTG/Syto60, $P < 10^{-10}$) (Appendix Fig S8L).

In the comparison of gene dependency profiles obtained either from genome-wide RNAi knock-down or CRISPR knockout screening techniques (Fig 2A), we observed a relatively low correlation between functional studies based on genome-wide RNAi screens ($r_{\mathrm{FUNC-RNAi}} = 0.22$) (Fig 2B), in line with previous reports showing that gene dependency profiles based on this technique are less robust (Jaiswal *et al*, 2017; Gautam *et al*, 2019). In contrast, genome-wide CRISPR screens exhibited a moderate consistency between studies ($r_{\mathrm{FUNC-CRISPR}} = 0.36$), significantly higher compared to genome-wide RNAi screens ($P < 10^{-10}$). As reported before (Gautam *et al*, 2019), the correlation between studies based on RNAi and CRISPR screens was also quite poor ($r_{\mathrm{FUNC-RNAi/CRISPR}} = 0.19$) (Fig 2A). Moreover, the agreement between the two screens performed at SANGER and BROAD Avana library was slightly lower compared to the screens performed exclusively at BROAD ($r_{\mathrm{FUNC}} = 0.35$ for BROAD Avana/SANGER vs. $r_{\mathrm{FUNC}} = 0.43$ for BROAD Avana/GeCKO/AML, $P = 2.7 \times 10^{-08}$). These results demonstrate how

laboratory-specific factors contribute to differences in the quantitative estimates of gene dependency profiles.

When investigating potential reason for the bi-modal distribution of correlation estimates for the MS-proteomic datasets, we found that the agreement of protein expression profiles varied depending on the sample preparation method (Fig 2C), Specifically, the BROAD, MGHCC_BREAST, and SANGER studies utilized tandem mass tag (TMT)-based peptide labeling before protein abundance quantification, whereas the other studies used a non-labeled (NL) approach. The correlation between TMT-labeled and NL proteome profiles was poor ($r_{\mathrm{PEXP}} = 0.11$), compared to proteome profiles generated at different study sites using the same method ($r_{\mathrm{PEXP}} = 0.63$ for NL/NL and $r_{\mathrm{PEXP}} = 0.52$ for TMT/TMT) (Fig 2C and D). In addition to differences in labeling, we found that the data normalization procedure also contributed to the differences in reproducibility. The TMT-labeled proteomic profiles are typically bridge-normalized, i.e., the bridge sample intensity in each plex is subtracted by log-ratio transformation. We observed much higher correlation between the BROAD (TMT-labeled and NL studies when using non-bridge-normalized intensities, compared to bridge-normalized intensities (Appendix Fig S9). However, there was a slightly better agreement in the coefficient of variation (CV) calculated for the common set of proteins between MHGCC_BREAST (TMT-labeled) and UW_TNBC (NL) ($r_{\mathrm{CV-PEXP}} = 0.44$, Fig 2E). This suggests that both the labeling and normalization procedures have a drastic impact on the estimates of protein abundance, which may lead to variability in the proteomic profiles.

### An analytical framework for meta-analysis and integration of multi-modal datasets

The availability of various data modalities of molecular profiles of cancer cell lines from multiple studies and laboratories, that show only a moderate overlap and consistency, poses a challenge for integrative approaches that leverage the multiple levels of profiling information to identify robust driver genes and biological processes. We hypothesized that genes that have a consistent molecular pattern shared across multiple studies and modalities are more likely to have a functional consequence relevant for cancer. Toward this end, we developed a non-parametric, rank-based framework, named cell line-specific gene Identification Pipeline (CLIP), which enables a systematic meta-analysis and integration of all the datasets collected and processed in this study (Fig 3). To boost the statistical power toward finding robust and reproducible signals in these data, the CLIP framework accounts for the substantial variability in the consistency of the various types of modalities (Fig 3A) between laboratories (Fig 3B) (see Methods and Protocols for details).

To solve the data sparsity challenge, we developed a "bottom-up" meta-analysis approach based on the concept of cancer cell line-specific (CCS) genes. A CCS gene exhibits a molecular feature that is unique for a given cell line in reference to the other cell lines, i.e., CCS gene has a context-specific property, which may also potentially contribute to the unique biological characteristics of the particular cell line. Statistically, CCS genes have the tendency to be located toward the extremes of a data modality distribution. For instance, the expression of ERBB2 gene is much higher in ERBB2 (HER2) driven breast cancer cell lines, compared to cell lines from other tissue types (Appendix Fig S10). The measure of CCS property

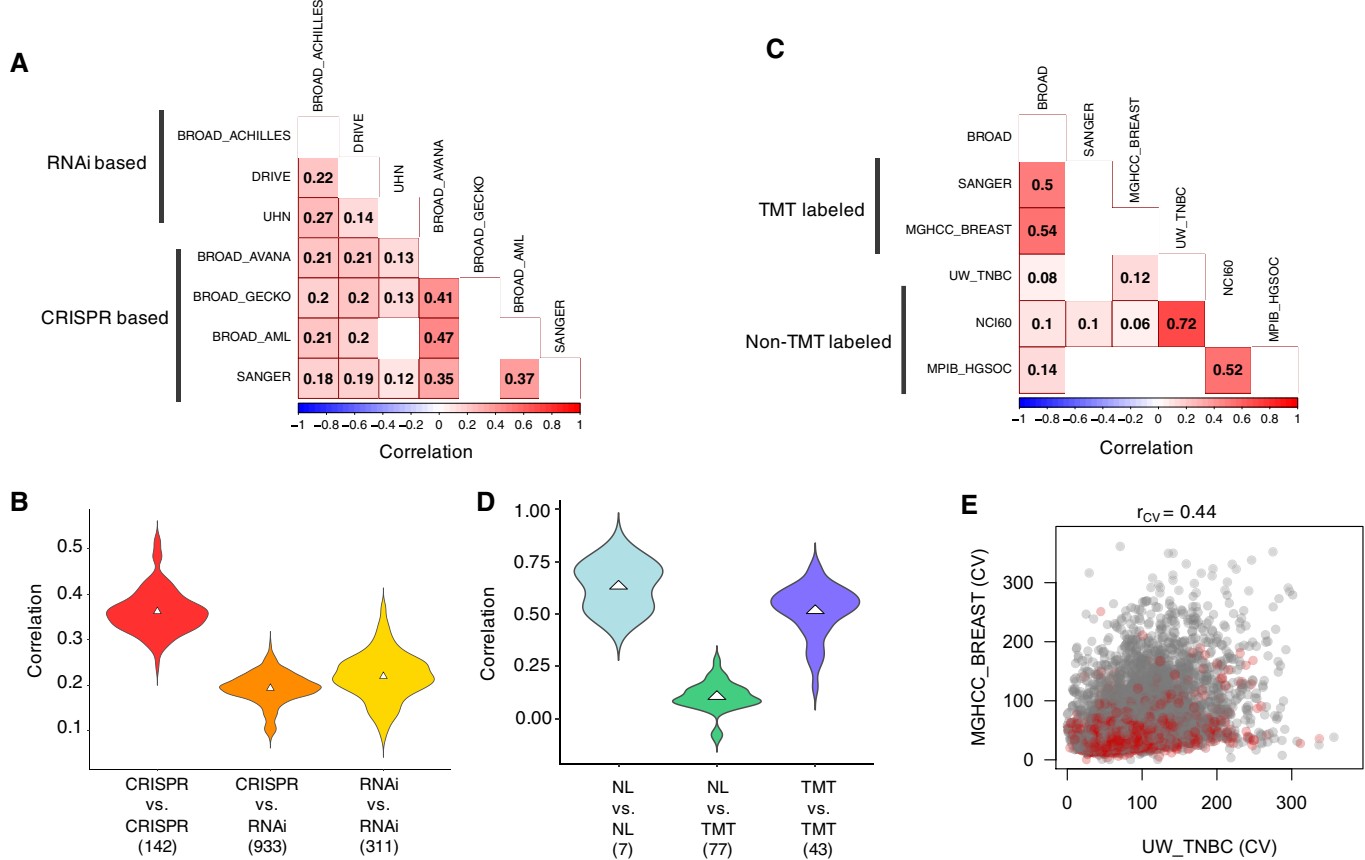

**Figure 2. Contribution of laboratory-specific factors to the reproducibility of functional gene dependency profiles and MS-based proteomic profiles.**

A   Correlation matrix plot of average Spearman's correlation of gene dependency profiles of cancer cell lines calculated based on genome-wide RNAi screens and CRISPR screens. Number of overlapping cell lines between any two datasets used for estimating the average correlation ranges between 2 and 284, with a mean of 46.4. The empty cells indicate that no identical cell lines were profiled between the two datasets.

B   Distribution of Spearman's correlation of gene dependency profiles between different study sites. Triangles represent mean correlation values. Numbers below the labels represent the number of overlapping cell lines based on which the distributions were drawn.

C   Average Spearman's correlation of MS-based proteomic profiles between different study sites generated using different peptide labeling procedures. The empty cells indicate that no identical cell lines were profiled between the two datasets. Number of overlapping cell lines between any two datasets used for estimating the average correlation ranges between 3 and 27, with a mean of 7.8.

D   Distribution of Spearman's correlation of MS-based proteomic profiles between different study sites. Numbers below the labels represent the number of overlapping cell lines based on which the distributions were drawn. Triangles correspond to the median value.

E   Coefficient of variation (CV) of proteins detected and quantified in UW TNBC study (non-TMT-labeled) vs. MGHCC BREAST study (TMT-labeled). Both studies had a maximal overlap of breast cancer cell lines for a robust estimation of CV. Housekeeping genes are highlighted as red dots. Spearman's correlation (rcv) was calculated to estimate the agreement in the CV estimates of common set of proteins between the two studies.

**Figure 3. Overview of the cell line-specific gene Identification Pipeline (CLIP) for integration of molecular datasets from multiple studies.**

A   CLIP performs a meta-analysis of datasets from multiple sites for each data modality type: Target addiction score (TAS), Gene dependency (FUNC), protein phosphorylation (PHOS), protein expression (PEXP), gene expression (GEXP), copy number variation (CNV), point mutation (MUT) and methylation (METH) profiles.

B   For each modality type, CLIP iterates over datasets available from multiple sites and quantifies the cancer context specificity (CCS) property for every gene G in cell line j.

C   For all unique cell lines, the CSS property is quantified for each gene G in a dataset D. For continuous modalities (METH, GEXP, PEXP, PHOS, FUNC, TAS), we defined the Outlier Evidence Score ($OES_{G,D,j}$), calculated by normalizing the observed value by the mean in the dataset for each gene ($X_i$). SD is defined as the standard deviation. For binary modalities (CNV-GAIN, CNV-LOSS and MUT), we defined the Proportion Score ($PS_{G,D,j}$) for each gene G in cell line j, calculated as the frequency of the alteration ($F_{D,j}$) normalized by the total samples in each dataset ($N_{D,j}$).

D   For a given cell line j, $OES_{G,D}$ scores across the available datasets are integrated using the Rank Product analysis to find statistically consistent genes that are at the top of the ranked list of genes ($CCS_{UP}$) or at the bottom ($CCS_{DOWN}$).

E   Finally, CLIP produces a profile of all the genes that are identified as CCS. In total, 13 different modality features were assessed by the CLIP framework, provided there are data available for a cell line for all the molecular datatypes. All genes identified as a CCS gene in any modality are highlighted, light orange for up-regulation and light blue for down-regulation. Genes that have CCS evidence across two or more modality types are considered in our analyses as robust Cancer Context-Specific (rCCS) genes, highlighted as light green.

F   A schematics of CLIP signature of a hypothetical gene, which summarizes its CCS evidence in a selected subset of cell lines, defined as a group based on any relevant criteria (the example shows all HER2+ breast cancer cell lines). Y-axis is the ratio of number of cell lines in which the gene is identified as a CCS gene vs. the total number of cell lines in the particular subset.

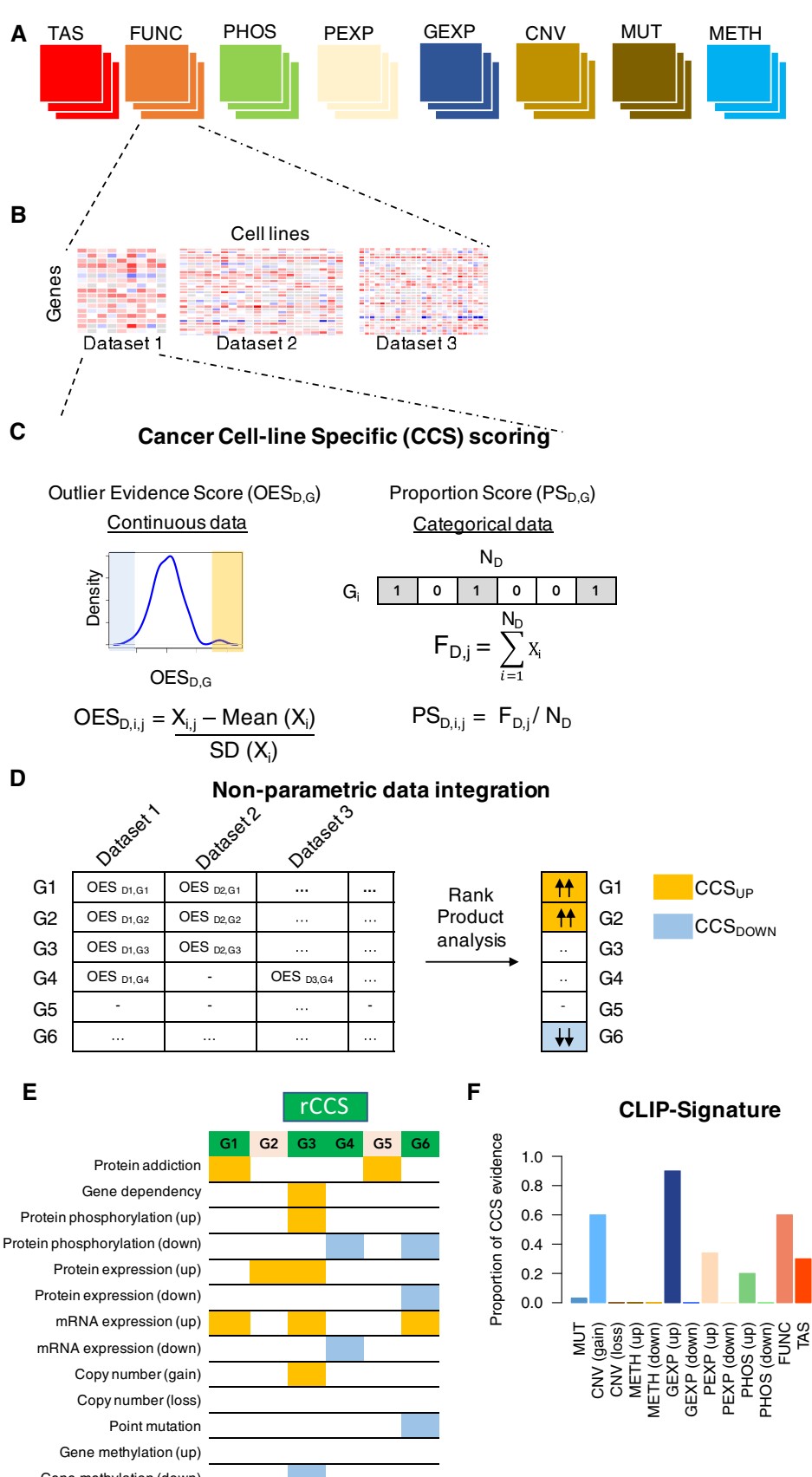

**Figure 3.**

for each gene in a dataset was quantified by estimating an Outlier Evidence Score (OES) for continuous variables, and Proportion Score (PS) for binary variables. The OES value for each gene was defined as a $z$-score over all the cell lines profiled in that study for a given data modality. Likewise, PS was defined as the proportion of cell lines in which a particular event is observed (Fig 3C). Next, for any given cell line, OES and PS scores of all the coding genes for a specific data modality from different studies were integrated to identify those genes that were consistently at the top ($CCS_{UP}$) or the bottom ($CCS_{DOWN}$) of the ranked list of genes in a given cell line (Fig 3D) (see Methods and Protocols).

For continuous data modalities, i.e., GEXP, PEXP. METH, PHOS, FUNC, and TAS, rank product analysis was performed to integrate over all the OES values for each gene-cell line combination from datasets across multiple laboratories. We also used the insights from our correlation analyses to fine tune the CLIP parameters for a robust integration of various modalities from multiple research sites. Genes with percentage of false positives (pfp) below a pre-specified threshold were considered statistically significant and defined as $CCS_{UP}$ or $CCS_{DOWN}$ genes for the respective data modality (see Methods and Protocols). For binarized data modalities, i.e., copy number gain (CNV_AMP) or loss (CNV_DEL) and MUT profiles, all the PS measures for that data modality from multiple studies were combined for a given gene-cell line combination. Specifically, any alteration that was observed in $\leq 10\%$ cell lines (arbitrary selected threshold) in any single dataset was considered as a CCS gene. Taken together, we quantified the CCS evidence of each gene across all eight types of data modalities.

Conceptually, the two categories $CCS_{UP}$ or $CCS_{DOWN}$ define a particular property of a gene, for instance, gene expression level higher or lower in the particular cell line compared to all the other cell lines. Ultimately, for each cell line, the CLIP meta-analysis framework provides a list of genes that show statistically robust evidence for being CCS genes by considering all the 8 types of molecular modalities in the gene space, where it generates a cell line-specific CLIP signature for each gene (Fig 3E). We further integrated the CCS evidence of each genes across multiple modalities to identify the robust CCS (rCCS) genes, based on the rationale that, if the CCS property persists through multiple modalities, then the likelihood for being a robust and reproducible CCS gene increases (Fig 3F).

### CLIP identifies established breast cancer cell line and subtype-specific drivers

To systematically test our meta-analysis approach, we reasoned that the list of rCCS genes for each cell line should be enriched for genes that determine the unique phenotypic or molecular characteristics of the particular cell line. As a proof-of-principle, we applied the CLIP framework specifically to 106 breast cancer cell lines (Appendix Fig S11), as they have been extensively profiled by multiple studies. Reassuringly, the meta-analysis approach was able to identify previously established driver kinases in several breast cancer cell lines (Szwajda et al, 2015; Fig 4A). The CLIP signature further revealed that most of the driver genes were identified based on the target addiction and gene dependency modalities, and a few others based on protein phosphorylation (up) and gene copy number (gain), as well as based on their point mutation views. rCCS

hits from CLIP were much more likely to have support from TAS or PHOS modality compared to the others (Fig 4B). Moreover, the rCCS genes supported by the GEXP modality were also likely to be supported by the CNV, METH, and PEXP modalities (Appendix Fig S12). Thus, in addition to identifying known drivers in breast cancer cell lines, the CLIP signature also provided insights into the mechanistic basis of the drivers based on multiple levels of supporting evidence across the data modalities.

Breast cancer cell lines are conventionally categorized based on the expression levels of ER and HER2 receptors into three subtypes, indicative of their clinical characteristics (Perou et al, 2000; Van't Veer et al, 2002; Koboldt et al, 2012) (Dataset EV11). We reasoned that the CLIP framework should be able to identify the relevant receptor proteins as rCCS genes in the cell lines belonging to these subtypes. Indeed, we observed that ER and HER2 were more frequently identified as rCCS genes in the subtype-specific cell lines (Fig 4C and E), suggesting that our data-driven approach to identifying context-specific players for each cell line was able to recapitulate the known molecular features of these cell lines. Furthermore, upon investigating the supporting rCCS evidence from the different molecular modalities, we found that these genes had shared support at functional, gene expression, and protein phosphorylation levels (Fig 4D and F). We further observed that methylation levels of ER were downregulated in a few of the rCSS-identified ER+ cell lines (Fig 4D). Similarly, in cell lines driven by ERBB2, the rCCS status was also supported by copy number gain, as it is known that ERBB2 is frequently amplified in HER2+ cell lines (Fig 4F). CLIP was also able to systematically identify a larger fraction of well-established driver genes compared to analyzing each data modality individually (Fig 4G), and also in comparison to an alternative approach based on multi-omics latent factor analysis method MOFA+ (see Methods) (Fig 4G).

A number of previously reported highly expressed genes, such as GATA3 in ER+ tumors (Perou et al, 2000; Koboldt et al, 2012) and PGAP3, GRB7, and STARD3 that are frequently co-amplified with HER2 (Perou et al, 2000; Koboldt et al, 2012), were also identified by CLIP for the ER+ and HER2+ subtypes, respectively (Dataset EV12). Similarly, SMAD7 was identified as one of the rCCS genes in the triple-negative breast cancer (TNBC) subtype (Dataset EV12). SMAD7 is known to play a role in metastasis and epithelial-to-mesenchymal (EMT) transition, a feature is frequently exhibited by the TNBC tumors (Valcourt et al, 2005; Katsuno et al, 2018). These results suggest that the CLIP framework is able to pinpoint the established cell line and subtype-specific drivers and also corroborate the mechanistic evidence for the genes involved in breast cancer progression from multiple data modalities. Importantly, many of these drivers would have been missed when looking at one of the studies or molecular modalities alone, but rather an integrative approach was necessary to identify the robust and reproducible driver signatures. In addition to the known markers, which were used here as positive controls, the CLIP framework also identified a number of novel genes specific to the established breast cancer subtypes (Dataset EV12), which provide leads for future research.

### CLIP identifies ECHDC1 as a novel tumor suppressor in breast cancer

While many of the known key players of breast cancer, such as BRCA1, ERBB2, ESR1, GATA3, CDH1, FOXA1, were frequently

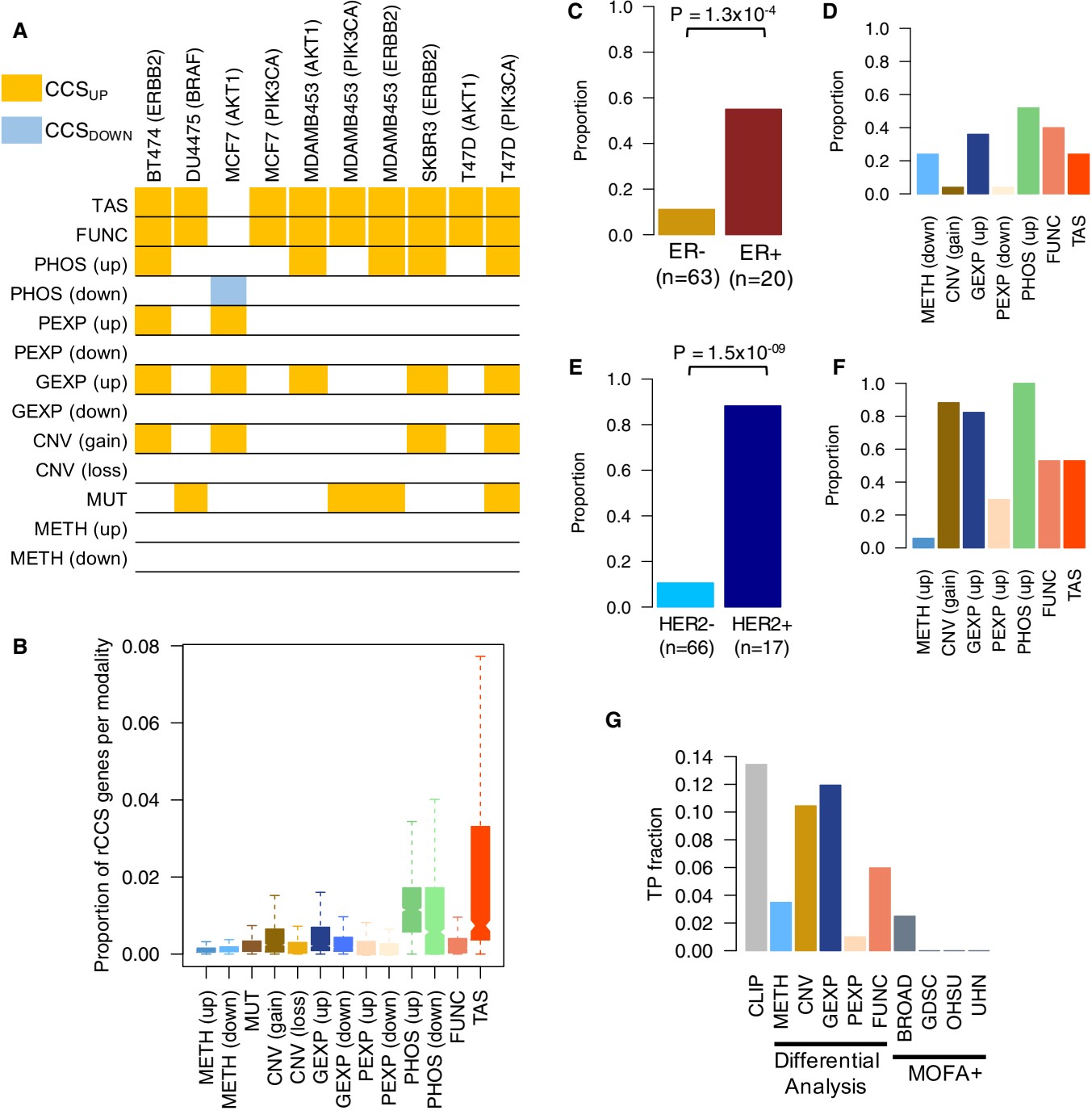

**Figure 4. CLIP signature of established breast cancer driver kinases.**

A  Subset of cell line-specific drivers that were identified as rCCS genes in this study. Highlighted entries indicate that the gene was identified as a rCCS gene in that modality.

B  Proportion of the rCCS genes identified by CLIP and supported by the various data modalities, relative to the average number of genes profiled for each modality in all cancer cell lines ($n = 1,047$). Boxes represent the interquartile range, notch in each box represents median value and whiskers the range of the values.

C  Proportion of ER+ and ER− breast cancer cell lines that have ESR1 as a rCCS gene. $P$-value was calculated with the Fisher's exact test.

D  The data modalities that supported the rCCS status of ESR1 and the proportion of cell lines having that evidence in the ER+ cell lines ($n = 20$).

E  Proportion of HER2+ and HER2− breast cancer cell lines that have ERBB2 as a rCCS gene. $P$-value was calculated with the Fisher's exact test.

F  The data modalities that supported the rCCS status of ERBB2 and the proportion of cell lines having that evidence in the HER2+ cell lines ($n = 17$).

G  Benchmarking the performance of CLIP to identify well-known breast cancer driver genes. True positive (TP) fraction of unique cancer driver genes ($n = 201$) for the three defined breast cancer subtypes as identified by CLIP and alternative approaches based on differential analysis in each specific modality alone, and using the latent factor-based Multi-Omics Factor Analysis (MOFA+) methods for data integration.

identified as rCCS genes by CLIP (Dataset EV13), we also observed novel genes, such as ECHDC1, SYCP2, GPX1, and MSN, whose role in breast cancer have not yet been studied extensively (Dataset EV13). In particular, ECHDC1 was identified as the most frequent rCCS gene among 24 out of the 106 breast cancer cell lines considered in our analysis. ECHDC1 encodes an enzyme, ethylmalonyl-CoA Decarboxylase 1, with a potential metabolite proofreading function (Linster et al, 2011). Interestingly, a previous report based on genome-wide association study implicated the genomic locus mapping to the ECHDC1 as a breast cancer risk locus in Jewish Asheknazi women (Gold et al, 2008). Notably, neither metabolic profiling nor germline genotyping data were used as part of the CLIP framework, thereby these studies provide an orthogonal support for a previously unappreciated role of ECHDC1 in breast cancer. Moreover, ECHDC1 was identified much less frequently as an outlier by a simpler approach to identify CCS in each individual study based on the METH data modality alone (Appendix Fig S3E), or using an integrative multi-omics factor analysis approach MOFA+ (see Methods and Protocols, Appendix Fig S3D), highlighting the usefulness of CLIP in identifying novel cancer-associated genes through robust integration of multi-modal multi-site datasets.

Our further analysis of the CLIP signature of ECHDC1 revealed that it was hypermethylated in all the breast cancer cell lines in which it was identified as a rCCS gene (Fig 5A). In the same cell lines, ECHDC1 mRNA was downregulated, suggesting that ECHDC1 could be a putative tumor suppressor. Moreover, we also observed that higher methylation levels or lower gene expression levels of ECHDC1 were associated with reduced breast cancer-specific survival probability in breast cancer patients ($P = 0.007$, log-rank test: Fig 5B), irrespective of their ER status (Appendix Fig S13), corroborating its putative tumor suppressive role. To experimentally challenge this finding, we used CRISPR/Cas9-mediated transcriptional silencing to knockout ECHDC1 in immortalized human MCF10A breast epithelial and malignant BT-474 cells (for the knockout efficiency, see Appendix Fig S14A and B). In the 5-day culture, ECHDC1-depletion induced MCF10A cell proliferation and growth already 72 h after embedding cells in 3D collagen matrix (Fig 5C and Appendix Fig S14C). However, BT-474 phenotype remained unaltered after the knockout (Fig 5C and Appendix Fig S14D), further supporting the tumor suppressive role of ECHDC1 in breast cancer cells.

To further illuminate the mechanistic basis of the tumor suppressive role of ECHDC1, we investigated the metabolic pathway in which ECHDC1 is involved, namely, the propanoate metabolism (Fig 5D and Appendix Fig S15). Propanoyl-CoA is an end product of catabolism of several branched chain amino acids, and oxidation of cholesterol side chains and odd-chain fatty acids. Propanoyl-CoA is further converted to succinyl-CoA, which is oxidized and fed into the TCA cycle. We reasoned that the down-regulation of ECHDC1 in breast cancer cells could lead to alteration in the levels of intermediate metabolites resulting in tumorigenesis. Subsequent metabolite profiling of three such intermediate metabolites revealed that succinate and 2OH-3MBA were significantly up-regulated in the breast cancer cell lines in which ECHDC1 was identified as a rCCS gene (Fig 5E). Succinate is known to be elevated in various cancers (Zhao et al, 2017; Dalla Pozza et al, 2020), and it may potentially contribute to tumor imitation and progression through regulation of mitochondrial function, hypoxia and reactive oxygen species production.

These observations further strengthen our data-driven approach and suggest that ECHDC1 is a novel tumor suppressor of breast cancer. This role was also supported by a pathway co-regulation analysis for predicting gene function (see Methods and Protocols), which suggests ECHDC1 is likely to play a role in TCA cycle and mitochondrial respiration, namely the electron transport chain, and fatty acid beta-oxidation pathway (Appendix Fig S16).

## CLIP predicts novel genetic interaction partners for known cancer drivers

To further extend the applicability of our integrative meta-analysis approach, we reasoned that the CLIP framework could also identify novel and robust genetic interaction (GI) partners of cancer driver genes. We considered specifically synthetic lethal (SL) interactions, i.e., the most negative end of GIs, which exhibit differential dependencies in context-specific genetic backgrounds; for instance, exclusively in the presence of a cancer driver mutation (Kaelin, 2005; Ashworth et al, 2011; Nijman & Friend, 2013). Such co-addictions are often observed only in certain cell lineages, making their identification challenging in smaller-scale studies (Nijman & Friend, 2013; Huang et al, 2020). As CLIP identifies context-specific rCCS genes in large panels of cell lines, and using multiple data modalities, we reasoned that a gene that is both supported by the gene dependency modality and identified robustly as a rCCS gene specifically in cancer cell lines mutated for a cancer driver could provide a multi-modal support for being a SL partner of the driver gene. We used Fisher's exact test to evaluate the difference in the proportion of rCCS genes between two groups of cancer cell lines, mutated and wild type, but we note that also other types of statistical tests for SL interactions could be utilized.

To examine this rationale for identifying context-specific and reproducible SL interaction partners, we first confirmed that CLIP was able to identify the known oncogenic addictions, such as KRAS, PIK3CA, and BRAF as rCCS genes, in the specific cell lines that harbor these oncogenic driver mutations (Fig 6A–D and Appendix Fig S17A and B, Dataset EV14). Cancer cell lines with such oncogenic driver mutations are known to be dependent on the same driver genes, due to oncogenic addiction (Weinstein & Joe, 2008), supporting the use of gene dependency modality in their detection. We also observed that known oncogenes were significantly more frequently identified as rCCS genes by CLIP (Fig 6E). Notably, even the removal of the FUNC modality did not affect the performance of CLIP. We observed a similar trend when the analysis was repeated for (i) all the driver genes, i.e., including both oncogenes and tumor suppressor genes (TSGs); and (ii) only TSGs (Appendix Fig S17C and D). Interestingly, when identifying TSGs in the setting for compulsory evidence of rCCS from FUNC modality, we observed that the difference in the frequency between known TSGs and non-TSGs was reduced, although it remained still statistically significant. This suggests that the multi-modal rCCS evidence for TSGs likely originates from the non-functional modalities.

We next extended this SL analysis to identify also co-addiction partners of other major cancer driver genes that are also frequently mutated in specific cell contexts, and in doing so, we identified a previously reported SL interaction between ARID1A and ARID1B (Helming et al, 2014), suggesting that the approach is able to recapitulate many confirmed SL interaction partners (Dataset EV14).

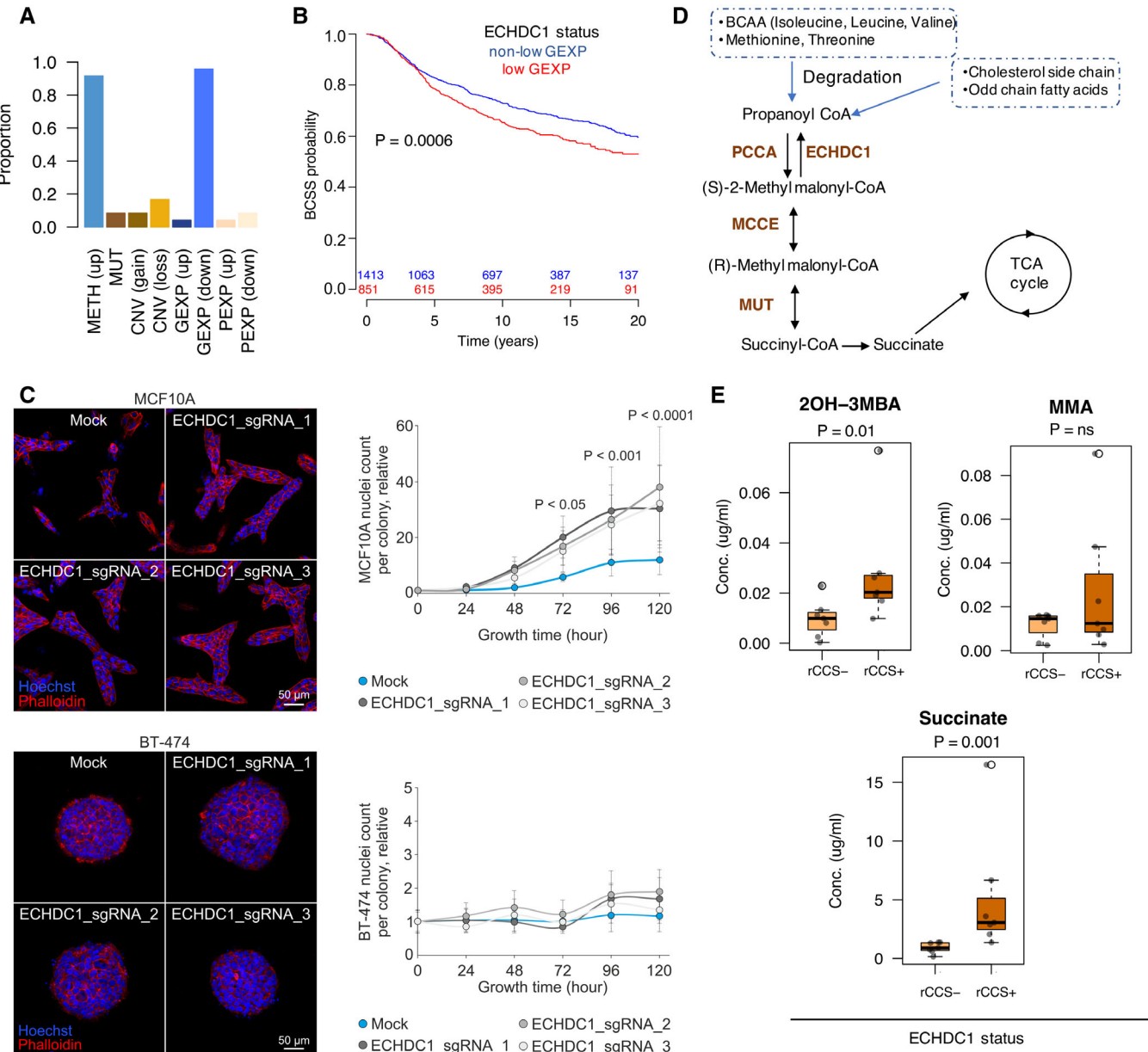

**Figure 5.  Identification of ECHDC1 as breast cancer tumor suppressor gene.**

A   The CLIP signature of ECHDC1 suggests that it was hypermethylated and down-expressed in all the breast cancer cell lines ($n = 24$) in which it was identified as rCCS gene.

B   Breast cancer-specific survival (BCSS) based on gene expression and methylation levels of ECHDC1 in breast cancer patient tumors in the combined Metabric and Oslo datasets ($n = 3,885$). Patients in the low GEXP category class have lower BCSS than those in the non-low GEXP group. Numbers above the *x*-axis line indicate the number of patients in each group, defined by the color code, at each time point. *P*-value from age-adjusted Cox-proportion hazard model.

C   Benign breast epithelial MCF10A and breast carcinoma BT-474 cells were embedded in 3D collagen as single cells or as spheroids, respectively, and the growth was followed for 5 days. Light micrographs show filamentous actin (phalloidin) and nuclei (Hoechst) in representative cell colonies. Quantitative assessment of the nuclei counts per colony show the induced proliferation in MCF10A cells after ECHCD1 sgRNA knockout. At 72 h, MCF10A mock vs. ECHDC1_sgRNA_1 and ECHDC1_sgRNA_2 $P < 0.05$; at 96 h mock vs. ECHDC1_sgRNA_1, ECHDC1_sgRNA_2 and ECHDC1_sgRNA_3 $P < 0.001$; at 120 h mock vs. ECHDC1_sgRNA_1, ECHDC1_sgRNA_2, and ECHDC1_sgRNA_3 $P < 0.0001$. Nuclei count relative to mock 0 h. Error bars indicate mean $\pm$ SEM; $n \geq 10$ colonies. Statistical significance was assessed with one-way ANOVA with Tukey's multiple comparison test. Scale bar 50 µm.

D   Metabolic pathway of propanoate metabolism.

E   Measured metabolite levels of intermediates in propanoate metabolism in select breast cancer cell lines with or without the ECHDC1 rCCS status ($n = 7$ in both groups). Boxes represent the interquartile range, whiskers represent the range of the values and solid line within the box correspond to the median value. Outlier points indicates values not included between the whiskers. Statistical significance was assessed with Wilcoxon test.

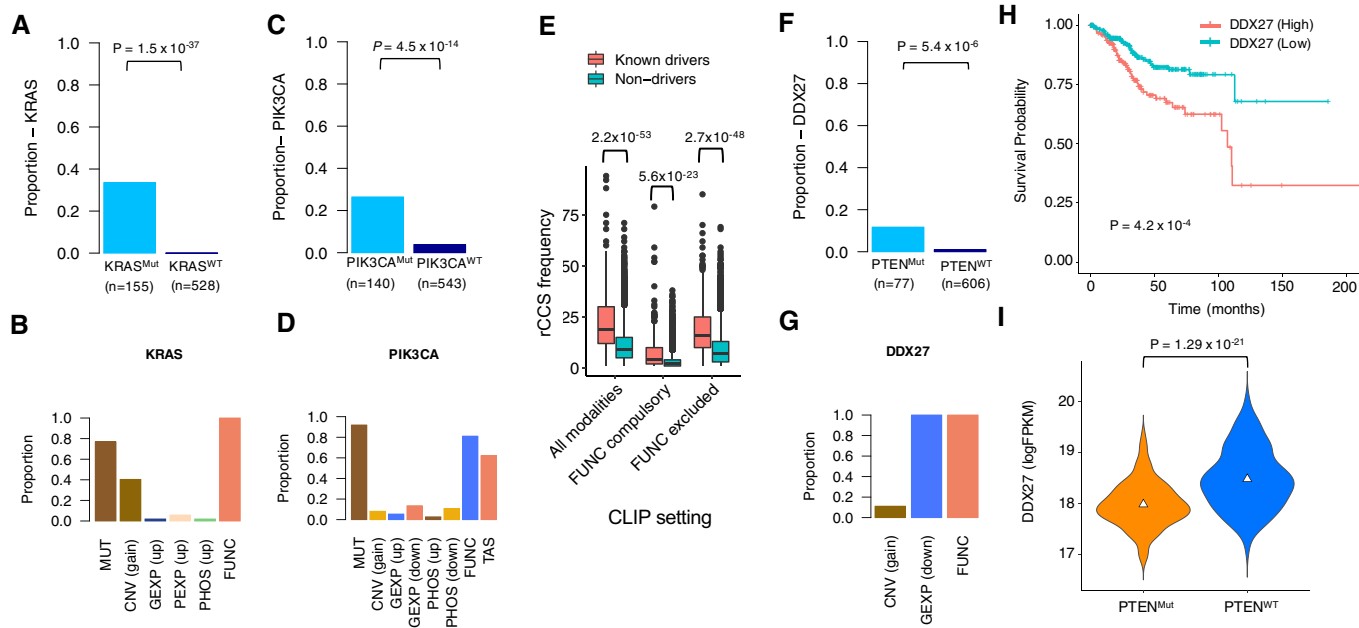

**Figure 6. Identification of novel synthetic lethal interactions.**

A   Proportion of KRAS-mutated (Mut) and KRAS wild-type (WT) cancer cell lines with KRAS identified as a rCCS gene. *P*-value was calculated with Fisher's exact test.

B   The modalities that support the rCCS status of KRAS and the proportion of cell lines having that evidence in the KRAS-mutated cell lines ($n = 155$).

C   Proportion of PIK3CA-mutated (Mut) and PIK3CA wild-type (WT) cancer cell lines with PIK3CA identified as a rCCS gene. *P*-value was calculated with Fisher's exact test.

D   The modalities that support the rCCS status of PIK3CA and the proportion of cell lines having that evidence in the PIK3CA-mutated cell lines ($n = 140$).

E   Systematic identification of cancer driver genes specific to epithelial cancer cell lines ($n = 737$) under multiple settings of CLIP run. rCCS genes identified by CLIP are enriched for known cancer drivers compared to non-driver genes, even after excluding the FUNC data modality from the CLIP approach. Boxes represent the interquartile range, whiskers represent the range of the values and solid line within the box correspond to the median value. Statistical significance was assessed with Wilcoxon test.

F   Proportion of PTEN-mutated (Mut) and PTEN wild-type (WT) cancer cell lines in which DDX27 was identified as a rCCS gene. *P*-value was calculated with Fisher's exact test.

G   The modalities that supported the rCCS status of DDX27 and the proportion of cell lines having that evidence in the PTEN-mutated cell lines ($n = 77$).

H   Survival analysis based on mRNA expression levels of DDX27 in patients with endometrial cancer in the TCGA dataset. Expression levels were divided into 2 classes, high ($n = 203$) and low ($n = 322$), based on mean expression level of DDX27 (logFPKM = 18.27). Patients in the high class showed lower survival probability than those in the low class ($P = 4.2 \times 10^{-4}$; log-rank test).

I   mRNA expression levels of DDX27 in PTEN-mutated (Mut, $n = 302$) and PTEN wild-type (WT, $n = 224$) endometrial patient tumors in the TCGA dataset. Triangles correspond to the median value. *P*-value was calculated with Wilcoxon test.

Overall, we also confirmed that CLIP is able to systematically identify SL interactions between paralog pairs more frequently compared to alternative approaches that use only the FUNC modality in each dataset (see Materials and Methods; Appendix Fig S17E). Moreover, even after excluding the FUNC modality, CLIP was able to identify a higher number of genetic interactions between paralogous genes compared to an analysis using the FUNC modality in each dataset (Fig 6E, Appendix Fig S17C and D).

Interestingly, we found a statistically strong evidence for DDX27 being a SL interaction partner of the known tumor suppressor PTEN ($P < 0.001$, Fisher's exact test) (Fig 6E, Appendix Fig S17F and G, Dataset EV14). Specifically, the CLIP signature of DDX27 suggested that all the PTEN mutant cell lines in which DDX27 was identified as a rCCS gene had downregulated mRNA levels of DDX27, which was also essential for their growth (Fig 6F). DDX27 belongs to the DEAD box nucleic acid helicase family of proteins that have recently been shown to modulate the formation of RNA molecular condensates, known as stress granules, thereby exerting a role in ribosomal translation (Fuller-Pace, 2013; Ivanov *et al*, 2019; Tauber *et al*,

2020). Recently, DDX27 was also shown to have a pro-tumorigenic function in colorectal cancer (Tang *et al*, 2018), and we observed using the TCGA data that it correlates with poor prognosis in endometrial cancer (Fig 6G), as well as in liver and renal cancer (Appendix Fig S18A and B).

It is noteworthy that PTEN is the most frequently mutated gene in endometrial cancers (Appendix Fig S18C), but with no drugs available for its direct reactivation. In line with the mutual exclusivity property of many SL partners (Unni *et al*, 2015; Varmus *et al*, 2016), we also found that endometrial tumors harboring loss-of-function PTEN mutations had much lower expression levels of DDX27 (Fig 6H, $P = 1.29 \times 10^{-21}$, Wilcoxon test), and this property was also significant in the TCGA Pan-Cancer dataset (Appendix Fig S18D, $P = 1.06 \times 10^{-19}$). In support of this observation, patients with downregulated PTEN and DDX27 tended to have better survival probability, compared to patients in which is not the case (Appendix Fig S18E). The mechanistic basis of this co-dependency is likely through the effects on protein synthesis and translation. The loss of PTEN induces an increased physical association of

mTORC2 and ribosome, which drives cancer growth while making the cells stress-prone and vulnerable to apoptosis (Keniry & Parsons, 2008; Zinzalla *et al*, 2011). Down-regulation of transcripts of RNA helicases together with a compounded loss of its activity could severely limit the ability of cells to cope with the induced stress by preventing the formation of RNA:RNA aggregates, thereby making the cells apoptosis prone (Ivanov *et al*, 2019; Tauber *et al*, 2020).

# Discussion

In this study, we first provided a comprehensive and quantitative view of the reproducibility of multiple data modalities of cancer cell lines by means of a systematic and non-parametric correlation analyses of molecular profiles of identical cell lines profiled in different laboratories. In particular, we found out that the profiles based on genomic technologies, such as transcriptome and copy number alterations (CNV), are highly consistent across the laboratories (Fig 1D). This is rather expected, given the robustness and maturity of the sequencing and hybridization techniques. Contrary to our expectations, we found that that the consistency of point mutational profiles was relatively low, even though the correlation difference in point mutation profiles between identical cell lines compared to non-identical cell lines was still highly significant (Fig 1D). This could be partly attributed to the correlation metric used for evaluating the reproducibility of binary and continuous profiles (Matthew's or Spearman's), as we observed also a decrease in correlation of binarized CNV calls compared to continuous copy number intensity calls (Appendix Fig S19).

While previous studies have also compared the consistency of genomic profiles between different laboratories (Barretina *et al*, 2012; Ben-David *et al*, 2018), their approaches are different from ours. For instance, Barretina *et al* (2012) compared the agreement between cancer cell lines based on mutational frequency of genes in CCLE with patient tumor-based mutational frequency from the COSMIC project. Likewise, Ben-David *et al* (2018) compared the allelic fraction of somatic variants in 106 cell lines common between CCLE and GDSC datasets. Interestingly, they observed that 10–90% of non-silent mutations identified in one dataset were not identified in the other, suggesting variability in the point mutational profiles of identical cell lines. We also observed a wide variation in the correlation of CNV profiles across the laboratories, suggesting that cell lines might have undergone clonal selection at different research sites. Such clonal divergence between identical cell lines has profound implications for the conclusions drawn from experimental assays performed on cultured cancer cell lines (Ben-David *et al*, 2018). We also investigated the various technical factors that contributed to the variability in the reproducibility estimates and found, for instance, significant discrepancies in the proteomic profiles between TMT-labeled and non-labeled MS techniques.

In the next phase, we developed a non-parametric integrative meta-analysis framework to identify robust molecular determinants, unique to an individual cancer cell line, that are shared among multiple data modalities and studies. A particular challenge for the integrative analysis is that the different research sites have profiled different panels of cell lines, making it difficult to derive robust estimates for every cell line. Nevertheless, we found extensive profiling information of breast cancer cell lines across multiple sites, which served here a purpose for evaluating the performance of our

approach. We demonstrated that our reproducibility-based integrative framework was able to identify well-established breast cancer driver genes as robust cancer cell line-specific (rCCS) genes using the available omics data from the set of breast cancer cell lines. Further, we extended this bottom-up approach for the identification of individual CCS genes and demonstrated that this approach also recapitulates the known drivers at a broader sub-group level, such as the well-established ER+ and HER2+ subtypes of breast cancer. We also benchmarked CLIP's performance and showed that CLIP was able to identify a much higher fraction of known breast cancer driver genes, compared the MOFA+ method (Fig 4F). MOFA+ could identify only five known driver genes of breast cancer subtypes in the integrative analysis of BROAD datasets, but was unable to identify any of the gold standard driver genes in other datasets, suggesting that MOFA+ framework may not be optimal for identifying subtype-specific driver genes. Moreover, CLIP was robust to the removal of each individual data modality and was able to identify a similar fraction of the driver genes (Appendix Fig S1D).

The CLIP framework relies on a meta-analysis approach that accounts for the variability in the observed measures for a gene across multiple studies using the rank product integration approach. Our motivation was to use the knowledge from the comparative evaluations to inform the noise parameters of the integration pipeline and to ultimately identify novel cancer-related genes and synthetic lethal interactions. While batch correction methods such as ComBat can be applied to computationally reduce the discordance of the heterogeneous datasets (Dempster *et al*, 2019), it assumes that the distribution of measured variables is largely similar between these datasets. However, in our study, we have used datasets generated from various platforms from multiple studies, both for genomic/molecular and functional profiling, which therefore have very different distributions of measured values. For instance, the distribution of array-based gene expression measurements is quite unlike RNA-seq based gene expression measurements, not to speak of binary point mutations or CNV profiles. Thus, instead of taking a traditional approach to computationally reduce the discordance of the heterogeneous datasets, we tried to utilize the data directly from the original studies and implemented rather simplistic methods of quantile normalization and other preprocessing steps that allows the data integration approach to be easily extendable also to future omics data and emerging studies, without the need to always test and implement specific discordance reduction strategy for each data modality separately. We hope the integrative approach and the harmonization procedures implemented by us and others (Appendix Fig S4) will become useful for the community for extending the CLIP approach to other omics data profiles from emerging studies, as well as for developing other types of meta-analysis approaches that require integrating multi-omics data from multiple studies.

As an application case, the CLIP framework identified a novel driver gene, ECHDC1, with a hitherto unknown tumor suppressive role in breast cancer. Notably, MOFA+ was unable to identify ECHDC1 as a latent factor shared across data modalities and driving heterogeneity in breast cancer cell lines (Appendix Fig S3D). Moreover, ECHDC1 could not be identified by analyzing the methylation modality only, hence demonstrating the need for multi-modal analysis. In particular, by forcing the constraint that a rCCS must be identified as a CCS gene in at least two modalities, CLIP improves the

likelihood of identifying true cancer-associated genes. We confirmed here the tumor suppressive role of ECHDC1 and highlighted a possible mechanism by which ECHDC1 may contribute to breast tumor growth. Specifically, we showed that the level of succinate increases in breast cancer cells with hypermethylated and lowly expressed ECHDC1. Moreover, it has been shown previously (Zhao *et al*, 2017; Dalla Pozza *et al*, 2020) that higher levels of succinate are associated with tumorigenesis and cancer progression via dysregulation of mitochondrial function, a hypoxic environment and production of ROS, which all have established roles in the etiology of cancer development. Additionally, several tumors are also known to have inactivating mutations in the SDH (Dalla Pozza *et al*, 2020), succinate dehydrogenase, the enzyme that processes succinate and feds into the TCA cycle.

We further applied the CLIP framework to identify context-specific synthetic lethal relationships with well-known cancer driver genes in a wider panel of cancer cell lines from multiple cell types. Notably, CLIP identified oncogenes as rCCS genes much more frequently, even after removing the FUNC modality from the model (Fig 6E), suggesting the advantage of integrating across multi-modal datasets. In addition to capturing the known aspects of oncogenic dependency, we identified a novel co-dependency relationship between PTEN loss-of-function mutations and RNA helicase enzyme DDX27. The cancer cell lines that had a higher prevalence of loss-of-function mutations in PTEN had lower expression of DDX27, and this mutually exclusive genetic interaction was particularly strong in endometrial cancers (Fig 6H and I). CLIP also identified a SL interaction between PTEN and DHX30, also a known RNA helicase. However, endometrial cancer patients with downregulated PTEN and DHX30 tended to have worse prognosis, compared to patients where this is not the case (Appendix Fig S19B); this suggests that not all the SL interactions identified by CLIP are supported by the survival analyses in the patient cohorts. This may be attributed either to the current statistical testing of SL interactions in the cell line data, that may identify SL interactions that are not clinically relevant, or to the limitations of the patient cohorts such as small sample sizes of patient subsets with the CLIP signature or limited patient annotations (Liu *et al*, 2018).

In summary, firstly, this study provides to date the most comprehensive perspective on the reproducibility of the genomic, molecular, and functional profiles of cancer cell lines and delineates specific factors that contribute to the consistency that should be considered in future studies. Secondly, to provide solution to the sub-optimal consistency, we developed an integrative meta-analytic framework for leveraging robust and reproducible signal from various modalities of molecular profiles that also accounts for the observed variation between datasets generated at different laboratories. The analytic choices of the CLIP approach were based on the reproducibility analysis of the multi-omics datasets. Finally, this study also demonstrates the potential of such integrative approaches for identification of novel molecular features having a confirmed role in breast cancer. We expect this approach will lead to many more exciting discoveries once more multi-omics profiling data become available also from other cancer types.

# Materials and Methods

### Reagents and Tools Table

| Reagent/Resource | Reference or Source | Identifier or Catalog Number |
|---|---|---|
| MDA-MB-231 | American Type Culture Collection (ATCC) | ATCC® HTB-26™ |
| BT-549 | American Type Culture Collection (ATCC) | ATCC® HTB-122™ |
| CAL-148 | American Type Culture Collection (ATCC) | ACC-460 |
| MFM223 | American Type Culture Collection (ATCC) | ACC-422 |
| DU4475 | American Type Culture Collection (ATCC) | ACC-427 |
| BT474 (HTB-20) | American Type Culture Collection (ATCC) | ATCC® HTB-20™ |
| CAL120 | American Type Culture Collection (ATCC) | ACC-459 |
| HS578T | American Type Culture Collection (ATCC) | ATCC® HTB-126™ |
| CAL51 | American Type Culture Collection (ATCC) | ACC-302 |
| CAL851 | American Type Culture Collection (ATCC) | ACC-440 |
| MCF7 | American Type Culture Collection (ATCC) | ATCC® HTB-22™ |
| SKBR3 | American Type Culture Collection (ATCC) | ATCC® HTB-30™ |
| BT20 | American Type Culture Collection (ATCC) | ATCC® HTB-19™ |
| T-47D | American Type Culture Collection (ATCC) | ATCC® HTB-133™ |
| MCF10A | American Type Culture Collection (ATCC) | ATCC® CRL-10317™ |
| HEK293T | American Type Culture Collection (ATCC) | ATCC® CRL-11268™ |
| **Recombinant DNA** | | |
| LentiCRISPRv2 | Addgene | Cat #52961 |
| pCMV-VSV-G | Addgene | Cat #8454 |

**Reagents and Tools table**  (continued)

| Reagent/Resource | Reference or Source | Identifier or Catalog Number |
|---|---|---|
| pCMV-dR8.2 | Addgene | Cat #8455 |
| **Oligonucleotides and other sequence-based reagents** | | |
| ECHDC1 guide RNAs | This study | Dataset EV5 |
| PCR primers | This study | Dataset EV6 |
| **Chemicals, Enzymes and other reagents** | | |
| OneTaq® DNA Polymerase | New England Biolabs | Cat # M0480 |
| Collagen type 1 from rat tail | Sigma-Aldrich | C7661-50MG |
| Alexa Fluor 568 phalloidin | Life Technologies | A12380 |
| Hoechst 33342 | Thermo Scientific | 62249 |
| **Software** | | |
| CorelDRAW 2020 | Corel | www.coreldraw.com |
| ImageJ | Abramoff *et al* (2004) | imagej.nih.gov/ij/ |
| **Other** | | |
| Applied Biosystems ABI3730XL DNA Analyzer | Applied Biosystems | www.thermofisher.com |
| Zeiss AxioImager.Z1 | Carl Zeiss | www.zeiss.com |
| Axiovert 200 | Carl Zeiss | www.zeiss.com |

## Methods and Protocols

### *Publicly available datasets reused in the multi-modal meta-analysis*

#### *The Broad Institute, Cambridge, USA (abbreviation: BROAD)*

The Broad institute is carrying out a number of large-scale cell line profiling projects such as the Cancer Cell Line Encyclopedia (CCLE) (Barretina *et al*, 2012; Ghandi *et al*, 2019), Cancer Dependency Map (DepMap) (Tsherniak *et al*, 2017; Meyers *et al*, 2017), and Cancer Therapeutic Response Portal (CTRP) (Basu *et al*, 2013; Seashore-Ludlow *et al*, 2015). Specifically, we reused the point mutation profiles of coding genes among 1,570 cancer cell lines from the DepMap project (DepMap Broad, 2019). We included point mutations that were either categorized as pathogenic by the authors or had FATHMM (Shihab *et al*, 2013) score ≤ −0.75 and binarized the genes for presence or absence of a mutation with a functional consequence. The processed copy number profiles of 1,080 cancer cell lines, generated using the Affymetrix SNP 6.0 arrays, were obtained from CCLE (DepMap Broad, 2019). Gene-level copy number gain and losses were called using a stringent threshold of ≥ 1 and ≤ −1, respectively. Genome-wide transcriptomic profiles for protein-coding genes generated with RNA sequencing for 1,156 cancer cell lines were obtained from the DepMap resource (DepMap Broad, 2019). Likewise, protein phosphorylation levels of 217 proteins in 899 cancer cell lines profiled using reverse phase protein arrays (RPPA) were obtained from the CCLE resource (Ghandi *et al*, 2019) and averaged for each protein. Quantitative proteomic profiles for 375 cell lines were generated using TMT-labeled multiplexed protocol for sample preparation (Nusinow *et al*, 2020). We re-analyzed both the bridge-normalized and non-bridge-normalized proteome profiles. For methylation profiles, we used averaged gene-level methylation profiles of promoters situated 1 kb upstream of transcription start sites for all coding genes in 843 cancer cell lines generated using reduced representation bisulfite sequencing (RRBS)

method), as provided by the authors of the study. Since the genome-wide CpG level data were not available, we used the original criteria for defining the promoter (Ghandi *et al*, 2019). For functional profiles, we used loss-of-function data from the Achilles Project that was generated with a pooled genome-wide shRNA screening of 501 cancer cell lines (Tsherniak *et al*, 2017; DepMap Broad, 2019). Gene dependency scores of each coding gene were estimated using the DEMETER2 algorithm (Tsherniak *et al*, 2017). We also re-analyzed gene dependency scores based on the genome-wide CRISPR-Cas9 knockout screens performed using various pooled sgRNA libraries from the DepMap portal. The Avana library was screened in 485 cancer cell lines (Meyers *et al*, 2017), the 120K sgRNA GeCKO v2 library was screened in 33 cancer cell lines (Aguirre *et al*, 2016), and the Sabatini library was screened in 15 acute myeloid leukemia (AML) cell lines (Wang *et al*, 2017). All the raw data for the knockout screens were processed by the Ceres algorithm (Aguirre *et al*, 2016) and downloaded from the DepMap or Achilles data portal. Drug response profiles of cancer cell lines for CCLE and CTRP v2 were obtained from the PharmacoDB database where the cell line identifiers were pre-harmonized (Smirnov *et al*, 2018), and the drug-induced viability response was estimated using the Drug Sensitivity Score ($DSS_2$), previously developed in the group (Yadav *et al*, 2014). While CCLE screened 24 compounds against 504 cell lines, the CTRP v2 dataset was generated by screening 544 compounds against 887 cell lines. Both of the drug sensitivity screens were based on CellTiter-Glo (CTG) assay to measure cell viabilities.

#### *The Sanger Institute, Hinxton, UK (abbreviation: SANGER)*

The Sanger Institute has also carried out several studies for molecular characterization of cancer cell lines, performed under the Genomics of Drug Sensitivity in Cancer (GDSC) project (Garnett *et al*, 2012; Yang *et al*, 2012; Iorio *et al*, 2016; Van Der Meer *et al*, 2018). We re-analyzed mutational profiles of 1,000 cancer cell lines for

coding genes from the COSMIC Cell Lines project generated by whole genome sequencing (Bamford *et al*, 2004). We selected mutations that were categorized as pathogenic or had FATHMM score ≤ −0.75 and binarized the genes for presence or absence of a point mutation with a functional consequence. Copy number profiles for 991 cancer cell lines generated using the Affymetrix SNP 6.0 arrays and processed with the PICNIC (Greenman *et al*, 2010) algorithm were obtained from the GDSC portal (Yang *et al*, 2012). The resulting total copy number calls were normalized by the ploidy level as follows:

$$\text{Copy Number(CN)}_{\text{Normalized}} = \frac{\text{Copy Number}_{\text{PICNIC}}}{\text{Ploidy level}} - 1$$

Gene copy number gains and losses were called by setting a threshold of ≥ 0.5 and ≤ −0.5, respectively, for the normalized copy number values. The RMA normalized gene expression profiles generated by Affymetrix Human Genome U219 array for 1,156 cancer cell lines were available from the GDSC portal (Yang *et al*, 2012). For methylation profiles (Yang *et al*, 2012), raw intensities generated by Illumina HumanMethylation450 BeadChip were processed using the Illumina Methylation Analyzer (IMA) R package (Wang *et al*, 2012). Quality control was performed by removing CpG sites with missing rate > 5% and detection $P > 0.05$. For the Illumina array-based methylation data, pre-annotated indices of CpG sites within the 1,500 kb (as supplied by the manufacturer) were used as the closest approximations for defining gene promoter methylation. Gene-level methylation intensities were obtained by averaging the methylation levels of CpG sites located in the annotated promoter site of each gene within a range of 200–1,500 base pairs (bp) upstream of the transcription start site. Ultimately, methylation levels for ~ 19,000 coding genes in 1,026 cancer cell lines were available for further analyses. Drug response profiles of 250 compounds tested in 1,075 cell lines were quantified using the $\text{DSS}_2$ method (Yadav *et al*, 2014) for GDSC1000 dataset, as obtained from the PharmacoDB database (Smirnov *et al*, 2018). Drug sensitivity screens were based on fluorescent nucleic acid stain probe Syto 60 assay to measure cell viabilities.

Additionally, proteomic and phospho-proteomic profiles of 50 colorectal cancer cell lines generated by multiplexed quantitative mass spectrometry (MS)-based proteomics technology were made available at the Sanger Institute (Roumeliotis *et al*, 2017). Multiplexing was performed by the isobaric labeling technology with tandem mass tag (TMT) reagents. Normalization step involved row-mean scaling of column-normalized calculation of intensities. Since we observed a minimal overlap between the phospho-peptides of proteins that were identified using MS in this study, when comparing to the protein residues profiled using targeted RPPA in other studies, we averaged multiple phospho-peptides corresponding to the same protein to generate protein-level phosphorylation estimates. Gene dependency profiles were generated at the Sanger Institute as a part of Project SCORE for 324 cell lines with a pooled genome-wide CRISPR-Cas9 knockout screens performed using two pooled sgRNA libraries, the Human CRISPR Library v.1.0 and v.1.1 (Behan *et al*, 2019). We used the copy number bias-corrected count fold changes as a measure of gene dependency in our analyses.

### Genentech Inc., USA (abbreviation: gCSI)
We reused gene expression profiles and mutation calls for 675 cell lines generated by RNA sequencing (Klijn *et al*, 2015). For the point

mutation data, we included the mutations whose annotations were already provided by the study authors, categorized as deleterious using variant function annotator methods, such as SIFT (Kumar *et al*, 2009), Condel (González-Pérez & López-Bigas, 2011), and PolyPhen (Ramensky, 2002). To be consistent, for the subset of unannotated mutations, we further annotated the variants using FATHMM (Shihab *et al*, 2013) and selected mutations with score ≤ −0.75, and binarized the genes for presence or absence of a mutation with a functional consequence. Gene copy number profiles for 668 cancer cell lines were generated using the Illumina HumanOmni2.5 4v1 arrays and processed with the PICNIC algorithm. We used ploidy-corrected copy number calls to categorize the amplifications and deletions. Copy number gains and losses were called by setting a threshold of ≥ 0.5 and ≤ −0.5, respectively. Drug response estimates for 16 compounds tested in 409 cell lines and quantified using the $\text{DSS}_2$ method (Yadav *et al*, 2014) were available from the PharmacoDB database (Smirnov *et al*, 2018). Drug sensitivity screens were based on CellTiter-Glo (CTG) assay to measure cell viabilities.

### National Cancer Institute, USA (abbreviation: NCI60)
The NCI-60 cancer cell line profiling data were extracted through the CellMiner data portal (Shankavaram *et al*, 2009), followed by further processing for the meta-analyses. Mutational profiles were generated by exome sequencing. We included the mutations that were categorized as deleterious by the SIFT (Kumar *et al*, 2009) and MA (Reva *et al*, 2011) variant function annotators, which were provided by the CellMiner data portal. We further annotated the variants using FATHMM (Shihab *et al*, 2013), selected those variants with score ≤ −0.75, and binarized the genes for presence or absence of a mutation with a functional consequence. We used summarized log-scale intensities representing copy number profiles generated by combining probe intensities from four platforms (Agilent Human Genome CGH Microarray 44A, Nimblegen HG19 CGH 385K WG Tiling v2.0, Affymetrix GeneChip Human Mapping 500k Array Set, and Illumina Human1Mv1_C BeadChip). A threshold of ≥ 0.4 and ≤ −0.4 was used to call copy number gains and losses, respectively. Similarly, processed GCRMA normalized gene expression profiles generated with Affymetrix Human Genome U133 plus 2.0 array was used. For methylation data, raw intensities generated by Illumina HumanMethylation450 BeadChip were processed. Gene-level methylation intensities were obtained by averaging the methylation levels of CpG sites located in the annotated promoter site of each gene within a range of 200 to 1,500 bp upstream of the transcription start site to calculate gene promoter level methylation as described in the earlier section. Log intensities of protein phosphorylation site levels on 94 proteins generated using 162 antibodies by RPPA were averaged for each protein. For proteomic profiles of the NCI60 panel cell lines, we used the label-free iBAQ-based quantitative estimates of protein levels (Gholami *et al*, 2013).

### University Health Network, Canada (abbreviation: UHN)
We downloaded the processed datasets from the Breast Functional Genomics data portal (Marcotte *et al*, 2016). Log ratios representing copy number profiles generated using the Human Omni-Quad BeadChip array and processed using the Circular Binary Segmentation (CBS) algorithm (Olshen *et al*, 2004) were available for 79 breast cancer cell lines. A stringent threshold of ≥ 0.4 and ≤ −0.4 was used

to call copy number gains and losses, respectively. Transcriptomic profiles were generated for 82 breast cancer cell lines. Log intensities of protein phosphorylation levels of 193 proteins were generated using 245 antibodies by RPPA and averaged for each protein. For gene dependency profiles, we used data from pooled genome-wide shRNA screen performed on 120 cancer cell lines from breast, pancreatic, and ovarian tissue types (Koh *et al*, 2012; Marcotte *et al*, 2016). Gene dependency scores of each coding gene were estimated using the DEMTER2 algorithm (Tsherniak *et al*, 2017).

### Oregon Health and Science University, USA (abbreviation: OHSU_BREAST)

All the processed datasets for breast cancer cell lines were downloaded from the Synapse portal (Daemen *et al*, 2013; Costello *et al*, 2014). Transcriptomic and genomic profiles were produced by RNA sequencing and exome sequencing, respectively. Point mutations were annotated as described in earlier sections using FATHMM (Shihab *et al*, 2013). Log ratios representing copy number profiles generated using the Affymetrix Genome-Wide Human SNP Array 6.0 and processed with the Circular Binary Segmentation (CBS) algorithm (Olshen *et al*, 2004) were available for 77 breast cancer cell lines. A threshold of $\geq 0.5$ and $\leq -0.5$ was used to call gene copy number gains and losses, respectively. For methylation data, raw intensities generated with Illumina Infinium Human Methylation27 BeadChip Kit were used for the genome-wide detection of 27,578 CpG loci, spanning a total of 14,495 genes. Probe intensities were processed to derive gene-level methylation intensities by averaging the methylation levels of CpG sites located in the annotated promoter site of each gene within a range of 200 to 1,500 bp upstream of the transcription start site. Log intensities of protein phosphorylation levels of 146 proteins generated by reverse phase protein lysate arrays were available. We utilized drug response profiles of 89 compounds tested in 71 cell lines and quantified using the $DSS_2$ method (Yadav *et al*, 2014), extracted from the PharmacoDB database (Smirnov *et al*, 2018). Drug sensitivity screens were based on CellTiter-Glo (CTG) assay to measure cell viabilities.

### University of Texas, MD Anderson Cancer Center, USA (abbreviation: MCLP)

We re-analyzed log intensities of protein phosphorylation levels of 382 proteins generated using 452 antibodies by RPPA for 650 cancer cell lines were (Li *et al*, 2017). Intensities for multiple phospho-sites from each protein were averaged.

### Massachusetts General Hospital Cancer Center, USA (abbreviation: MGHCC_BREAST)

Quantitative proteomic profiles of 41 breast cancer cell lines were generated using multiplexed quantitative mass spectrometry (MS)-based proteomics technology (Lapek *et al*, 2017). Multiplexing was performed using the isobaric labeling technology with ten-plex tandem mass tag (TMT) reagent, and the bridge-normalized intensities were used in the analyses.

### University of Washington, USA (abbreviation: UW_TNBC)

We processed the label-free iBAQ-based quantitative proteomic profiles of 20 breast cancer cell lines generated using non-multiplexed label-free quantitative mass spectrometry (MS)-based proteomics technology (Lawrence *et al*, 2015).

### Novartis, USA (abbreviation: DRIVE)

We made use of gene dependency profiles for 8,195 genes in 398 cancer cell lines for which raw data were generated by pooled genome-wide shRNA libraries (McDonald *et al*, 2017). shRNA level scores were collapsed to gene dependency scores of each coding gene using the DEMTER2 algorithm (Tsherniak *et al*, 2017).

### Institute for Molecular Medicine Finland (abbreviation: FIMM)

Drug response estimates for 52 compounds tested in 50 cell lines and quantified using the $DSS_2$ method were obtained from the PharmacoDB database (Mpindi *et al*, 2016; Smirnov *et al*, 2018; Gautam *et al*, 2019). Drug sensitivity screens were based on CellTiter-Glo (CTG) assay to measure cell viabilities.

### Max Planck Institute of Biochemistry, Germany (abbreviation: MPIB_HGSOC)

We re-analyzed label-free quantitative (LFQ) estimates of proteomic profiles of 30 ovarian cancer cell lines generated with a label-free quantitative mass spectrometry-based proteomics technology (Coscia *et al*, 2016).

### Calculation of target addiction score

Since drug response profiles exist in the compound space, we projected them into gene space to create an additional functional data modality. To do this, we used our previously described pipeline, target addiction scoring (TAS), which transforms the drug response profiles into target addiction signatures (Jaiswal *et al*, 2019). The TAS pipeline makes use of drug poly-pharmacology to integrate the drug sensitivity and target selectivity profiles through systems-wide interconnection networks between drugs and their targets, including both primary protein targets as well as secondary off-targets. The TAS approach is individualized in the sense that it uses the drug sensitivity profile of each cancer cell line screened separately against a library of bioactive compounds and then transforms the observed phenotypic profile into a cell line-specific target addiction profile, hence enabling ranking of potential therapeutic targets based on their functional importance in the particular cell line.

We applied the TAS pipeline separately to each cell line drug response profile considered in the study. First, we obtained the set of potent protein targets for each drug from various drug-target databases as described previously (Jaiswal *et al*, 2019; Dataset EV1). For instance, we retrieved at least one potent target for 349 of the 495 compounds profiled in the CTRP dataset. The rest of the compounds are either non-targeted drug treatments or compounds with unknown target profiles. Likewise, protein targets were identified for 201/250 compounds in GDSC; 44/52 compounds in FIMM; 33/89 compounds in OHSU; 13/16 compounds in gCSI; and 19/25 compounds in CCLE dataset. For each individual target $t$, $TAS_t$ was calculated by averaging the observed drug response (here, $DSS_2$) over all those $n_t$ compounds that target the protein $t$. Eventually, we were able to derive the functional TAS profiles for a median of 222 targets in each dataset.

$$TAS_t = \sum_{i=1}^{n_t} \frac{DSS_i}{n_t}$$

### Statistical analyses for reproducibility analyses

Spearman's correlation analysis was conducted to evaluate the reproducibility of continuous molecular data types between any two

studies. The reproducibility analyses were performed on the identical set of cell lines and on the common set of genes between any two datasets. Matthews correlation coefficient (MCC) was calculated to assess the consistency between binarized data types, such as gene-level mutational profiles, or copy number gain and loss profiles. Only the correlation values with $P < 0.05$ calculated based on the molecular profiles with $\geq 10$ genes or proteins were considered for further analysis.

### Data processing for meta-analysis and integration

Binarized datasets of gene copy number gains and copy number losses were generated from their continuous CNV profiles using the study-specific thresholds, as specified above. Protein phosphorylation intensities from multiple residues mapping to the same protein were averaged to generate protein-level phosphorylation estimates. Since the GDSC phospho-proteome study in colorectal cancer cell lines used global MS technique for protein phosphorylation profiles (Roumeliotis *et al*, 2017), only those proteins that were also profiled in other research sites using targeted RPPA were considered for the meta-analysis.

### Meta-analysis and data integration framework CLIP

In the cell line-specific gene Identification Pipeline (CLIP), we first define the notion of a Cancer cell line-specific (CCS) gene, and then quantify the strength of evidence for each gene across data modalities and datasets from different laboratories. In this study, we only considered data modalities in the gene/protein space, and exclude drug response-based profiles; however, TAS profiles were derived from the drug response phenotypes.

For continuous data modalities, i.e., gene expression, protein expression, gene methylation, protein phosphorylation, drug sensitivity, and gene dependency profiles, the strength of CCS evidence was estimated using one-class rank product analysis performed separately for each gene and cell line combination from datasets across multiple laboratories using RankProduct package (Del Carratore *et al*, 2017). For methylation, gene expression, protein expression, and protein phosphorylation data modalities, genes with pfp < 0.10 were considered significant. As we observed lower consistency for functional gene dependency profiles, we used a less stringent threshold of pfp < 0.25 to increase the coverage of identifying robust CCS genes. These choices were informed by the reproducibility analyses of the multi-modal profiles.

For binarized data modalities, i.e., copy number gain and loss and gene point mutation profiles, any alteration that was observed in $\leq 10\%$ cell lines in any single dataset was considered as a CCS gene. For cell lines that were profiled in only one study, the rank product analysis could not be performed, and therefore, we selected all the top-ranked genes: top-100 genes (0.5% of all genes) for the continuous variable datasets, except for functional gene dependency modality (FUNC), where the top-200 genes were considered (1% of all genes) to account for the higher noise in functional gene dependency profiles. For the protein phosphorylation datasets, we selected the top-10 genes (0.5% of all proteins), considering that 200 proteins were assayed by reverse phase protein arrays on average.

Genes that were identified as CCS genes in two or more data modality types are considered in our analyses as robust CCS (rCCS) genes. Fisher's exact test was performed to identify the genes that

are enriched in different breast cancer subtypes or pre-defined subgroups of cancer cell lines. Overall, we applied the CLIP approach to 1,047 cancer cell lines for which data was available for $\geq 6$ modalities (Dataset EV2).

### Selection of thresholds for binary CNV calls for CLIP

The categorized CNV modalities (CNV_AMP and CNV_DEL) were used as inputs in the CLIP pipeline, instead of the continuous CNV data. Notably, while datasets from BROAD, OHSU, UHN, and NCI60 have been originally processed using the CBS algorithm, data from GDSC and gCSI have been processed using the PICNIC algorithm. To make the integrative approach more systematic and unbiased, we inspected the copy number distributions of two well-known CNV alterations in breast tumors; ERBB2 which is known to be frequent amplification, and PTEN which is known to be a frequent deletion in breast cancer patients (Koboldt *et al*, 2012). For ERBB2, we looked at the distribution of its copy number values across all cell lines in each dataset (Appendix Fig S1A). We then identified the threshold for calling an amplification in each dataset at the value where we observed a sharp deflection. For BROAD, this analysis led to the threshold of $\geq 1$ for calling amplifications. For, GDSC, gCSI, and OHSU, values $\geq 0.5$; and for UHN and NCI60, value $\geq 0.4$ were considered as amplifications. Similarly, for calling deletions; BROAD < 1; GDSC, gCSI, and OHSU < 0.5; and UHN and NCI60 < 0.4 were selected for the CLIP meta-analysis based on the analysis of PTEN deletions (Appendix Fig S1B). We also executed CLIP at various detection thresholds and did not observe significant differences in its ability to identify breast cancer subtype-specific driver genes that were used for benchmarking the performance of CLIP (Appendix Fig S1C).

### Benchmarking the performance of CLIP against other methods

As the first reference approach, we assessed the relative performance of the CLIP framework in comparison to each independent data modality alone for their ability to identify breast cancer subtype-specific driver genes. For this benchmark, we first defined the set of "true" cancer driver genes as identified in the recent study (Bailey *et al*, 2018), separately for pan-cancer and breast cancer tumor types, totaling to a set of 201 genes (Dataset EV3). To evaluate the performance of each data modality, we assumed the gene or protein expression up-regulation, hypermethylation, and copy number gains as molecular phenotypes that are likely to correspond to subtype-specific driver genes. Using the empirical Bayes framework of *limma* R package (Ritchie *et al*, 2015), we performed differential analysis between the two groups of breast cancer cell lines, "subtype+" and "subtype−" groups for each data modality and each study site. Next, for each data modality, we performed rank product analysis to integrate the evidence for the odds of differential levels, measured by B-statistic (Ritchie *et al*, 2015), separately for datasets from every research site to identify robust subtype-specific driver genes. Then, we characterized the subtype-specific genes as up-regulated vs. downregulated, hypermethylated vs. hypomethylated, gain vs. loss, based on the direction of average fold changes. Finally, we compared the fraction of true cancer drivers that were identified by CLIP and each individual data modality for each breast cancer subtype.

To assess how much the different modalities contributed to the performance of CLIP, in addition to the base setting of CLIP in

which all the modalities were included, we also re-ran the CLIP by removing each data modality one at a time, and measured the fraction of true positive driver genes identified after the removal of the input datasets (Appendix Fig S1D).

In addition, we also benchmarked our approach against the Multi-Omics Factor Analysis (MOFA+) method, which models latent factors for integration of multi-omics datasets (Argelaguet *et al,* 2020). MOFA+ is an unsupervised framework for discovering hidden factors that capture biological variability across multiple data modalities as well as within an individual modality. While MOFA+ has been primarily applied for integrating multiomics datasets from a single study site, in principle it can be applied also to multi-omic datasets from multiple sites. However, due to the differences in distribution of the omics measurements, resulting from the various technological platforms that were used at the various sites for generating the datasets, we chose to apply MOFA+ to breast cancer cell lines from BROAD, GDSC, OHSU, and UHN only. Furthermore, we reasoned that the multi-group framework in MOFA+ could be used to identify the latent factors that are specific to each breast cancer subtype, even though the authors of MOFA+ acknowledge that the multi-group framework is not designed for capturing differential changes between the groups, but rather to identify the principal sources of variability that drive each group.

Specifically, we pursued the following strategy for the omics datasets from each site: (i) For BROAD, we considered CNV, MUT, METH, GEXP, PEXP, FUNC_CRISPR, and FUNC_RNAI data modalities. Each dataset was quantile-normalized and subset to the set of breast cancer cell lines, before running the multi-group MOFA+ model using its default settings. (ii) The proportion of variance explained for each group by each data modality/view for 15 latent factors were inspected (Appendix Fig S2), and the factors and views with the highest variance were then selected further. (iii) Top-20 (top-10 positive weights and top-10 negative weights) view-specific features loading on to each factor were extracted from the MOFA+ model and then compared against the established gold standard driver genes from a recent study (Bailey *et al,* 2018, Dataset EV3).

### ECHDC1 identification using alternative approaches

To further challenge the performance of CLIP as multi-modal and multi-site data integration approach, we assessed whether ECHDC1 could be identified by alternative methods that integrate multi-omics dataset. First, we applied the MOFA+ method (Argelaguet *et al,* 2020), which combines datasets from a single research site. Because ECHDC1 was supported by GEXP and METH modality from CLIP, we applied MOFA+ method independently to all the datasets that profiled for methylation in breast cancer cell lines, namely BROAD, OHSU and GDSC. GEXP modality data from BROAD and OHSU were log-transformed before input into MOFA+. Otherwise, the MOFA+ method was run using its default options without defining any groups. We performed the same steps (i)–(iii) as detailed in the previous section for the BROAD (CNV, GEXP, METH, MUT, FUNC_AVANA, FUNC_ACHILLES), GDSC (CNV, GEXP, METH, MUT, FUNC), and OHSU (CNV, GEXP, METH, MUT) datasets. The other modalities, such as PHOS and TAS, were not considered because of very low number of features compared to the other modalities (Appendix Fig S3).

Alternatively, we also devised a simplistic approach to investigate whether ECHDC1 could be identified by analyzing the METH modality only. As for CLIP, we performed $z$-scaling for each gene on quantile-normalized gene-averaged methylation levels on breast cancer cell lines from each dataset. Next, to identify the outlier genes (i.e., the CCS genes in CLIP), specific to each cell line, we selected all the genes that had $z$-scores above or below 1.66 standard deviations from the mean $z$-score in each dataset. Following this, we checked the frequency of ECHDC1 being identified as an outlier gene in all the breast cell lines that were profiled in each dataset. We implemented this approach on methylation datasets from BROAD and GDSC (Appendix Fig S3). The method could not be implemented in the OHSU dataset because ECHDC1 methylation levels were not measured.

### Systematic identification of cancer drivers and oncogenic addictions

We further assessed the ability of CLIP to identify known oncogenic addictions by utilizing the set of cancer drivers, as defined in Bailey *et al* (2018), as true positives. After sub-setting the list of driver genes to only those that were defined as oncogenes for pan-cancer and epithelial cancer types, we had a total of 227 oncogenic driver genes. Next, to benchmark CLIP, we counted the frequency of each rCCS gene across all the epithelial cancer cell lines. We reasoned that the true oncogenic addictions will be identified more frequently as a rCCS gene, when compared to the non-oncogenes. We ran the CLIP pipeline on the subset of epithelial cell lines, in the following three settings to identify rCCS genes: (i) including datasets of all modalities (number of epithelial cell lines = 737); (ii) including datasets of all modalities, but constraining rCCS gene selection criteria by compulsory identification as a rCCS gene in the FUNC modality (number of epithelial cell lines = 679); and (iii) including datasets of all modalities, except FUNC modality (number of epithelial cell lines = 736). Next, we compared the difference in rCCS frequency between oncogenes and non-oncogenes groups in each setting using the Wilcoxon test.

### Statistical detection of genetic interactions and synthetic lethal partners

To statistically identify candidate genetic interactions (Gis) between genes, Fisher's exact test was performed to evaluate the difference in the proportion of rCCS genes between two groups of cancer cell lines, mutated and wild type, for all the well-known cancer driver genes. For defining the subset of potential synthetic lethal (SL) interactions, we considered genetic interaction partners of only those rCCS genes that were also identified as essential genes, based on the evidence from the gene dependency modality (Dataset EV4). Out of the 2,018 cell lines that were included in the meta-analyses, we considered 1,047 cell lines for which molecular data were available in $\geq 6$ of the eight data modalities in gene space. Furthermore, we removed cell lines derived from bone, skin, nervous, and hematopoietic systems, restricting our interaction analyses to epithelial cancer cell lines ($n = 679$). For selecting the mutated driver genes with relevance in patient tumors, we considered the highly frequent driver genes in patient tumors from a recent pan-cancer study of mutational landscape in The Cancer Genome Atlas (TCGA) dataset (Kandoth *et al,* 2013) and used a subset of the driver genes that were mutated in at least five cell lines.

### Benchmarking CLIP for identification of genetic interactions between paralogs

We assessed the ability of CLIP to identify Gis between paralogous genes in comparison to using only the FUNC modality dataset. First, we obtained the list of paralogous genes in the human genome from the Duplicated Genes Database (DGD) (Ouedraogo *et al*, 2012). In total, we had a list 3,543 paralogs in the human genome that are constituted by 945 unique paralog groups. For instance, ARID1A and ARID1B are paralogs that belong to a unique group. Briefly, we conducted a Fisher's exact test to assess the difference in proportions for rCCS genes between mutant and WT cell lines for each paralog group. For example, if ARID1A is mutated, then we tested whether ARID1B is enriched in proportion as a rCCS gene in mutant vs. WT cell line comparison. Or, vice versa, if ARID1B is mutated, then we evaluated whether ARID1A is enriched as an rCCS gene in those cell lines. We performed the genetic interaction analyses on paralogs that are mutated in at least 10 epithelial cell lines to have sufficient power to detect robust associations. If any gene from a paralog group was detected as statistically significant, we record that as a paralog GI association. Finally, we plot the proportion of number of unique paralog groups that have a paralog SL/GI association in relation to the total number of unique paralog groups that were tested. Moreover, we ran this analysis in different settings to evaluate the contribution of FUNC modality to the performance of CLIP; (i) FUNC Compulsory: GI analysis on rCCS genes from CLIP, but rCCS gene should be compulsorily identified as a rCCS gene in the FUNC modality (number of epithelial cell lines = 679). This is the default setting to identify SL relationships, since a gene has to be essential and rCCS. (ii) All modalities: GI analysis on rCCS gene from CLIP (number of epithelial cell lines = 737). Since essentiality is not mandatory criteria for rCCS calling, these associations can be generally categorized as genetic interactions. (iii) FUNC excluded: GI analysis on rCCS gene from CLIP by including all modalities, except for FUNC modality (number of epithelial cell lines = 736). Since, gene essentiality is not the included criteria for rCCS calling, these associations can be generally categorized as genetic Interactions.

To benchmark the performance of CLIP, we performed GI analysis in each individual FUNC dataset. Using the *limma* package (Ritchie *et al*, 2015), we performed a linear model differential expression analysis between mutant vs. WT cell lines for difference in gene essentiality scores. As described above, we subset to paralogs that are mutated in at least 10 epithelial cell lines. Associations with $P < 0.05$ were considered statistically significant. To compare the differences in proportions of paralogous GI associations using multiple strategies, we performed two proportions $z$-test for difference in proportions of every comparison relative to CLIP setting (FUNC compulsory).

### Survival analysis in the patient tumor cohorts

To perform survival analysis in breast cancer patients with high statistical power, we combined two patient cohorts with 20-year follow-up data and molecular profiling: the Metabric cohort (Curtis *et al*, 2012), which contains around 2,000 breast cancer patients with gene expression data, and the Oslo cohort (Fleischer *et al*, 2014, 2017), which contains 334 breast cancer patients with DNA methylation data. For the patients with DNA methylation data, an average DNA methylation level for all pre-annotated indices of CpG

sites within the range of 200–1,500 bp upstream of the TSS was calculated for each patient. We grouped the patients into two groups based on gene expression or promoter methylation of ECHDC1. To capture the effect of loss of tumor suppressor function, we identified the patients with either low expression or high methylation by selecting the patient tertile (one-third of the patients) with the lowest expression or highest methylation (low GEXP group), which was compared to the rest of the patients (non-low GEXP group). Subsequently, we performed survival analysis on the two groups (log-rank test), where age was included as continuous covariate in a multivariate Cox model. The sex and race were not considered as relevant factors in Nordic breast cancer patients. Race information was not available for Metabric patients. The analyses were performed in R using the packages; *survival* and *rms*.

### Experimental validation of ECHDC1 knockout
#### Cell lines

Immortalized breast epithelial cell line MCF10A and breast carcinoma cell line BT-474 (both American Type Culture Collection; ATCC) were cultured according to manufacturer's instructions in a humidified incubator with 5% $CO_2$ at 37°C and routinely checked and tested negative for mycoplasma contamination using MycoAlertPlus™ Mycoplasma Detection Kit (Lonza) according to manufacturer's instructions.

#### CRISPR/Cas9-mediated ECHDC1 knockout

Oligonucleotides (Merck) encoding single guide RNAs (sgRNA) against *ECHDC1* (see Dataset EV5 for sequences) were cloned into LentiCRISPRv2 plasmid (#52961, Addgene) as described previously (Sanjana *et al*, 2014). Lentivirus particles were generated by seeding HEK293T cells at a density of 105 cells per $cm^2$. After 16 h, cells were transfected with transfer plasmid, packaging plasmids pCMV-VSV-G (Stewart *et al*, 2003; #8454, Addgene), and pCMV-dR8.2 (Stewart *et al*, 2003; #8455, Addgene) using Lipofectamine 2000 (Life Technologies). Supernatants were collected 48 h after transfection. Cells were infected in 24-well plates with lentiviral particles (MOI ~ 5) for 24 h in the presence of 8 µg/ml polybrene. The culture media were replaced for puromycin (Lentiguide; 1 µg/ml) containing media, and cells were selected for 3 days respectively.

#### PCR amplification and Sanger sequencing

To confirm the gene knockout, the *ECHDC1* target regions were amplified with corresponding primers (Dataset EV6) using OneTaq® DNA Polymerase (New England Biolabs; Cat: M0480). After the initial denaturation at 94°C for 9 min, 30 polymerase chain reaction (PCR) cycles were performed as follows 94°C for 40 s, 65°C for 30 s, and 72°C for 40 s. PCR-amplified products were purified using a PCR purification kit (Macherey-Nagel) and sequenced using Sanger sequencing (Applied Biosystems ABI3730XL DNA Analyzer).

#### Invasive growth assay

Control and *ECHDC1* knockout, MCF10A and BT-474 cells were embedded as $5 \times 10^3$ single cells (MCF10A) or $5 \times 10^2$ cell-spheroids (BT-474) in 3D collagen drops. BT-474 cells were allowed to form tumor-representing cell-spheroids for 24 h as 5 µl hanging drops in complete media. Rat tail collagen type I (Sigma-Aldrich) was dissolved in 0.25% acetic acid and diluted 1:1 with 2× MEM (Gibco) to final concentration of 2.25 mg/ml. Cells and pre-formed

spheroids were embedded in 30 μl collagen and incubated at 37°C up to 5 days. Cell growth was followed daily by phase-contrast imaging using inverted epifluorescence microscope (Axiovert 200; Carl Zeiss). Collagen 3D drops were fixed at 0, 24, 48, 72, 96, and at 120 h using 4% PFA for 1 h at room temperature (RT).

### Immunofluorescence and imaging

Fixed 3D collagen drops were post-fixed with ice-cold 1:1 acetone-methanol for 45 s and blocked with 15 % FBS—0.3% Triton-X in PBS for 2 h at RT. Drops were incubated with Alexa Fluor 568 Phalloidin (1:40; ThermoFisher Scientific) for 4 h at RT, washed with 0.45 % Triton-X in PBS followed by Hoechst 33342 staining (10 μg/ml; ThermoFisher Scientific) for 30 min at RT. After PBS, wash drops were mounted with Vectashield Antifade Mounting Medium (Vector laboratories).

Light micrographs were taken by using Zeiss AxioImager.Z1 upright epifluorescence microscope with Apotome combined with a computer-controlled Hamamatsu Orca R2 1.3-megapixel monochrome CCD camera and ZEN software. 20× Plan Apochromat 0.8 NA objective was used. Immunofluorescence quantifications were done by using ImageJ (Abramoff *et al*, 2004), and post-acquisition image processing was performed using CorelDRAW 2020 software (Corel). Statistical analysis was carried out using one-way ANOVA with Tukey's multiple comparison test.

### Metabolite assay of intermediates in ECHDC1 pathway

A total of 14 breast cell lines (see Dataset EV7), 7 for ECHDC1 rCCS positive (rCCS+) and 7 for rCCS negative (rCCS−), were selected for subsequent metabolite assays based on their ECHDC rCCS status. In total, 42 samples (14 lines with 3 replicates) were run with UPLC-MS (MRM) method, each sample containing *ca.* $4 \times 10^6$ cells. Culture conditions for each cell line are detailed in Dataset EV7.

Quenching protocol for adherent cells involved the following steps: Cells were washed with 2× volume of cold phosphate-buffered saline (PBS) and incubated with trypsin (TrypLE™ Express Enzyme (1×), #12605010, Invitrogen) at 37°C until the cells detached. Trypsin was inactivated by adding an equal volume of cold fetal bovine serum (FBS, #10270-106, Invitrogen). The cells were counted and centrifuged at $400 \times g$ for 5 min at 4°C. The cells were washed with 2× volume of cold PBS, each time centrifuging at $400 \times g$ for 5 min at 4°C. For each sample, $4 \times 10^6$ cells were resuspended in 500 μl of cold PBS and centrifuged at $400 \times g$ for 5 min at 4°C in 1.5 ml microcentrifuge tube. The cells were quickly washed with 2× volume of deionized water, not disturbing the pellet and not exposing the cells to water for more than 4–5 s. All water was aspirated from the tube. The cells were frozen in liquid nitrogen and stored at −80°C.

The protocol for cell disruption and extraction was adapted from a previously described method (Dettmer *et al*, 2011). Briefly, cells disrupted with a combination of freeze-thaw cycle (−80/+4°C) and sonication. 1,000 μl of 80% MeOH (Honeywell, Riedel-de-Haën™, Seelze. Germany) and 10 μl of internal standard (ISTD) (conc. 10 μg/ml) was added to the purified cells. Then samples were ultra-sonicated in ice bath for 10 min, put to liquid nitrogen back and forth three times. After cell disruption, samples were vortexed for an hour, centrifuged at 21,500 *g* for 5 min at +4°C. The second extraction was performed with 600 μl of 100% MeOH for 30 min. The supernatant (1,600 μl) was dried under vacuum (MiVac Duo

concentrator, GeneVac Ltd, Ipswich, UK), reconstituted to 50 μl of 0.1% formic acid in acetonitrile (ACN)/H$_2$O (Honeywell, Riedel-de-Haën™, Seelze. Germany), and run with UPLC-QTRAP/MS with ESI (+) and (−) switching (ExionLC UPLC, ABSciex; 6500+ QTRAP-MS, ABSciex).

Ten microliter (μl) of extract was injected into the LC column with the mobile phase flow of 0.4 ml/min at +35°C. The LC separation was carried out on a reversed-phase UPLC-column (Waters Acquity BEH C18, 150 × 2.1 mm, Ø 1.7 μm). A gradient elution of the analytes was achieved using a program with mobile phases A (aqueous 0.1% formic acid) and B (0.1% formic acid in ACN). The linear gradient started at 99% A and 1% B was held for 2 min and proceeded from 1% B to 10% in 2 min, 10% B to 90% in 2 min, held at 90% for 1 min, then switched back to 1% B and left to stabilize for 2 min. Total run time of 9 min.

In total, 12 metabolites were analyzed from the cells: aspartate (Asp), glutamate (Glu), glutamine (Gln), α-ketoglutarate (α-KG), fumarate, succinate, malate, citrate, isocitrate, 2-hydroxy-3-methyl-butyrate (2OH-3MBA), malonate, and methylmalonate (MMA). D$_3$-methylmalonate (D$_3$-MMA) was used as an ISTD. Multiple Reaction Monitoring (MRM) transitions for 13 analytes were monitored: *m/z* 134/74 for Asp, 148/56 for Glu, and 147/84 for Gln in positive mode; and *m/z* 145/83 for α-KG, 114.9/71 for fumarate, 117.1/73 for succinate, 132.8/71 for malate, 190.8/111 for citrate, 190.8/73 for isocitrate, 117.1/71 for 2OH-3MBA, 103/59 for malonate, 117/73 for MMA and 120/76 for D$_3$-MMA (ISTD) in negative mode. Calibration curves with five standard mixes (conc. 100, 10, 1, 0.1, and 0.01 μg/ml) were used for quantification for all 12 metabolites. The samples were not normalized to cell count, but only for ISTD, hence unit μg/ml.

### Prediction of ECHDC1 function with gene co-regulation analysis

We used the Gene-Module Association Determination (G-MAD) algorithm, implemented in the GeneBridge toolkit (Li *et al*, 2019), to predict the gene function of ECHDC1. G-MAD considers transcriptome data sets from six species (human, mouse, rat, fly, worm, and yeast), and performs a competitive gene set testing method—Correlation Adjusted mEan RAnk gene set test (CAMERA) (Wu & Smyth, 2012), which adjusts for inter-gene correlations to compute the enrichment between gene-of-interest and biological modules. Gene-module connections with enrichment *P*-values that survive multiple testing corrections of the gene or module numbers are scored and normalized (range: −1 to 1). We restricted our analysis to breast tissue datasets ($n = 153$), compiled in the GeneBridge expression database to enrich for breast tissue-specific associations (Dataset EV8).

## Data availability

The datasets and computer code utilized in this study are available in the following repositories:

- Computer scripts: GitHub (https://github.com/jaiswal-alok/clip-meta).
- Datasets: Figshare (https://doi.org/10.6084/m9.figshare.13473168).

**Expanded View** for this article is available online.

## Acknowledgements

We thank the authors of the various projects used in the study for making their data publicly available, and Prof. Therese Sørlie and Prof. Bjørn Naume as PIs of the Oslo patient cohorts for enabling the survival analyses. We also thank Dr. Anil Giri for his help with the methylation data analysis, and Dr. Vidya Velagapudi for her help with designing the metabolite assays. AJ was supported by the Integrative Life Science (ILS) doctoral program and the FIMM-EMBL PhD program. TA was supported by the Academy of Finland (grants 292611, 310507, 313267, 326238), the Cancer Society of Finland, the Sigrid Jusélius Foundation, and Helse Sør-Øst (grant 2020026). PG was supported by the Orion Research Foundation sr. KL and EAP were supported by the Finnish Cancer Foundation and K. Albin Johansson foundation, and EAP by the ILS Doctoral Programme in University of Helsinki. This work was also supported by the Breast Cancer Now's Catalyst Programme (grant 2017NovPCC1067 to TA and KW), which is supported by funding from Pfizer. Metabolomic analysis in Viikki Metabolomics Unit were supported by Biocenter Finland (BF) and Helsinki Institute of Life Science (HILIFE) (NS).

## Author contributions

AJ conceived, designed and conducted the study. AJ analyzed the data and drafted the manuscript. PG, EP, YA, ST and NN contributed to cell line laboratory experiments. NS contributed to metabolite measurements. ZT contributed to data analysis. TF contributed to the survival analyses. KL contributed to overseeing the cell line experiments. KW contributed to study design and critically reviewed the manuscript. TA contributed to study design, supervised the study and revised the manuscript. All authors read and approved the final manuscript.

## Conflict of interest

The authors declare that they have no conflict of interest.

## Ethics approval and consent to participate

The Oslo breast cancer patient cohort is covered by the REK approvals 2015/116 and 2010/498, and the compilation with the Metabric cohort is covered by the REK approval 2016/433. All the data on human material were generated with the approval of the Norwegian Regional Ethical Committee (REK), under the guidelines for handling of sensitive human data.

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
