## [Review Process File · Molecular Systems Biology]

Multi-modal meta-analysis of cancer cell line omics profiles identifies ECHDC1 as a novel breast tumor suppressor

Alok Jaiswal, Prson Gautam, Elina Pietilä, Sanna Timonen, Nora Nordstrom, Yevhen Akimov, Nina Sipari, Ziaurrehman Tanoli, Thomas Fleischer, Kaisa Lehti, Krister Wennerberg, and Tero Aittokallio
DOI: 10.15252/msb.202010126

Corresponding author(s): Alok Jaiswal (ajaiswal@broadinstitute.org) , Tero Aittokallio (tero.aittokallio@helsinki.fi)

Review Timeline:

Submission Date:	17th Feb 20
Editorial Decision:	25th Mar 20
Revision Received:	13th Nov 20
Editorial Decision:	7th Dec 20
Revision Received:	15th Jan 21
Editorial Decision:	11th Feb 21
Revision Received:	17th Feb 21
Accepted:	19th Feb 21

Editor: Maria Polychronidou

Transaction Report:

Thank you again for submitting your work to Molecular Systems Biology. We have now heard back from the three referees who agreed to evaluate your manuscript. As you will see from the reports below, the reviewers raise substantial concerns on your work, which, I am afraid to say, preclude its publication in Molecular Systems Biology.

The reviewers acknowledge that the proposed approach seems potentially interesting. However, they think that as it stands the study remains rather preliminary and point out that in absence of further analyses and direct comparisons to alternative approaches the superiority and advantages of the proposed approach remain unclear. They also express concerns regarding the validations of several of the reported conclusions, including the proposed role of ECHDC1. As such, at this point we see no choice but to return the manuscript with the message that we cannot offer to publish it.

Nevertheless, we would not be opposed to considering a substantially revised and extended manuscript based on this work, provided that the issues raised by the reviewers can be convincingly addressed. Some of the more essential issues that would need to be addressed include:

- Demonstrating the advantages and superior performance of CLIP compared to simpler approaches (and ideally other multi-omics approaches e.g. MOFA). As reviewer #1 mentions, it would also be important to show that CLIP is really required to identify relevant genes e.g. ECHDC1, and that ECHDC1 would have been missed by simpler approaches.
- Providing further validations for several of the identified context-specific synthetic lethal relationships and for the proposed role of ECHDC1.
- Making the code available for the reviewers to evaluate (and future users to use easily).
- Providing more global support for the ability of CLIP to identify oncogenes.
- As reviewer #2 mentions, the first part of the study remains somewhat incomplete, as no concrete conclusions emerge on how to tackle the problem of discordance between datasets. Follow up analyses e.g. providing computational solutions and informing on analytic choices in order to alleviate the lack of reproducibility/agreement would significantly enhance the impact of the study.

All three reviewers provide constructive suggestions on how to improve the study. We recognize that thoroughly addressing the referees' concerns would involve substantial further analyses with unclear outcome and we understand if in light of the substantial revisions required, you prefer to submit your study elsewhere.

A resubmitted work would have a new number and receipt date. It will be editorially evaluated afresh and its novelty will be re-assessed at the time of submission. As you probably understand, we can give no guarantee about its eventual acceptability. If you do decide to follow this course then we would ask you to enclose with your re-submission a point-by-point response to the points raised in the present review.

I am sorry that the review of your work did not result in a more favorable outcome on this occasion, but I hope that you will not be discouraged from sending your work to Molecular Systems Biology in the future. In any case, thank you for the opportunity to examine this work.

REFeree REPORTS

Reviewer #1:

Summary:

The authors provide an extensive meta-analysis of 53 omics profiling studies in more than 2000 cancer cell lines processed in 12 research laboratories, considering for each cell line up to 9 different data modalities. They describe a pipeline, which they name CLIP, for identifying robust cancer cell line specific (rCCS) genes that may be relevant to certain cancer types and use the rCCS data to infer synthetic lethal (SL) pairs. The study further provides a systematic correlation analysis between different methods within the same data modalities (eg. microarray vs RNA-seq) in order to identify inconsistencies observed for certain data modalities. They use the pipeline to identify a putative tumor suppressor ECHDC1 in breast cancer and provide some in vitro validation.

Aside from the identification of ECHDC1, the most useful contribution of the study to the cancer community would be full access to the meta analysis results, and ideally both in the form of full results, and with a release of the pipeline code. Though the authors have done a partial release of the results as supplementary tables, they don't suffice because unless we are mistaken one cannot fully trace the rCCS designations fully back to the modalities and data used to make the designations. There is a github site, but it appears to have no instruction or code, and the data on the site is in a format (Rdata) that is not universally friendly. For useful exploratory analysis based on the results, cancer researchers would need more than the authors have provided.

Major comments:

1. The authors failed to provide the necessary code and instructions to reproduce their analysis. Without instructions on how to use the processed data on their github page, it is impossible to judge the reproducibility of their analysis and whether it was properly implemented.
2. The methods can often be vague. For instance what does it mean to say "Fisher's exact test was performed to evaluate the difference in the proportion of rCCS genes between different cancer subtypes or pre-defined subgroups of cancer cell lines." We understand the statistics, but don't understand the intended meaning of the sentence.
3. Did this method (robust, multi-modality) really lead to the identification of ECHDC1, or would

analysis of methylation data from a single site have sufficed? The authors need to make the case.

4. We think the authors did not provide sufficient validation to show that their context-specific synthetic lethal relationships found between tumor suppressor PTEN and RNA helicase DDX27 and DHX30 has a clinical relevance. For instance, the authors could easily mine transcriptomic data for endometrial cancers (Uterine Corpus Endometrioid Carcinoma, 560 cases) in TCGA and test whether patients where both PTEN and DDX27 is downregulated within the same patient compared to patient where this is not the case, show a better survival.

5. We think the survival analysis shown in Figure 5B is not convincing. First, Figure 5B does not show that increasing methylation leads to poor survival -- the highly methylated group fares worse than the low methylated group. The authors only compare medium and high. Second, the authors did not account for the most common confounding variables (age, sex and race). The authors should repeat the analysis using a cox proportional hazard regression model and account for these confounding factors. Finally, the most natural data to test for proportional hazards for this gene might be gene expression, and Figure 5A at least suggests that gene expression is available.

Reviewer #2:

The authors present a computational integrative approach aiming at harmonisation and joint analysis of data from the chemo/functional-genomic characterisation of cancer in vitro models spanning 9 different data modalities, and 12 different studies.

The main aim of the presented approach is to dilute/remove biases and inconsistencies due to different laboratory specific experimental practices and analytical protocol across the considered studies and to provide a unified analytical framework for the identification of cancer-driver multi-omic signatures, hence cancer driver genes and genomic-context specific genetic dependencies.

After converting all the analysed data omics at the level of gene patterns, the authors devised an elegant strategy to quantify the 'signal-strength' of each gene in each cell line across each omic, then they computed cell line specific (CSS) gene scores based on how consistent their signal-strength is in a given cell line across considered omics.

Then they show how the resulting signatures of top CSS genes are predictive of established and novel cancer driver genes, and can be used to efficiently predict synthetic lethal gene pair.

Very briefly, the problem tackled by the authors is important and timely, and the attempt they made to devise a method to unify/harmonise multiple data omics across different studies should be lauded for its simplicity and elegance.

In addition, the discovery of a new tumour suppressor genes for breast cancer and synthetic lethal gene pairs add value and demonstrate the utility of the presented approach. However I have a number of major concerns (detailed below) preventing me to warrant publication of this manuscript in Molecular Systems Biology, at least not before a convincing extensive revision.

First of all, the authors have included a systematic comparisons of data omics across studies, whose purpose or utility is left unclear. In fact they do not attempt any computational reduction of the observed discordance across studies for certain omics nor they use any of the findings from this

analysis to guide the follow up analytical choices.

In addition, as the author themselves demonstrated in a previous publication, the major problem with the assessments of consistency in drug screening and cancer dependency datasets is failing to account that often that the lack of agreement is due to a lack of variance in the observed drug-response/gene-essentiality signals. That's why for example in a recent publication measuring the agreement between cancer dependency datasets [PMID: 31862961] correlation metrics employed to compare gene dependency profiles across studies are weighted by the 'context-specificity' of gene essentiality (as never-essential/always-essential genes would add noise only to these estimation).

In the light of these consideration, it is surprising that here the authors use crude Pearson's correlations and Mathew Coefficients to compare omics across studies, without further digging into the identified discordancies, trying to reduce them or gaining insights for the follow up integrative analysis.

Moreover, in this work there is a general lack of performance comparisons when considering the ability in detecting cancer drivers, between the CLIP signatures and the individual omics (when used alone). The authors should compulsorily show, for example, that the hits coming out from contrasting CLIP scores across breast cancer subtype are in a larger number or more reliable or with a larger Recall of true cancer drivers, with respect to using for example expression, or methylation or cnv or gene dependency alone. In addition, comparisons with other multi-omic integrative methods, such as MOFA (PMID: 29925568) for example, should be included, or at least discussed if this is not feasible.

Also the claim that CLIP is able to identify known oncogenic addictions is not supported by any systematic analysis. The authors should compulsorily show that cell line specific rCCS genes are indeed enriched for essential genes. Ideally, keeping the gene-dependency omic out of their integration effort, at what extent the rCCS gene rank in each cell line is predictive of that gene essentiality in the same cell line? This should be quantified, for example by means of ROC indicators. A similar analysis should be performed to assess the ability to detect associations between paralogs, such as the mentioned AIRD1A/B case but, again, more systematically

Other major points:

- * The author used inconsistent and seemingly arbitrary thresholds across datasets to call for copy number amplification/deletions. They should specify how these threshold values were picked and why.
- * Differently from the other omics, for which the authors extracted data processed via the protocol adopted in their native study, the drug response datasets from multiple studies were commonly extracted from a unique on line resource (PharmacDB), which applies its own processing method. This introduces an extent of inconsistency and makes the integrative effort presented by the authors not 100% complete. This choice should be motivated.
- * Not clear what is the criteria used to pick the genes with high/low CSS and the subset of Breast cancer cell lines, to be shown/discussed a more systematic assessment should be performed in order to show that indeed the proposed approach confirms established cancer type specific drivers, with True/False-positive rates
- * At least from the picked example, it seems that the proposed approach seems to be biased toward

the discovery/confirmation of gain-of-function drivers. Arguably this might be true, as gain-of-functions mutations, might lead to constitutive expression hence sustained protein abundance and dependency, is this the case? I think that this would deserve a dedicated analysis/investigation

* In addition, aren't the CCS strength biased toward concordant expression/proteomic level? (aren't these expected to be more frequent with respect to other pairs of omics?)

* It is not clear how the authors converted the proteomic/phosphoproteomic datasets at the gene-level domain as done for the drug response dataset. I would expect they have used an approach similar to the TAS method, but a description of this in the methods is totally missing

Finally, I have the following minor points to highlight:

* references are inconsistently formatted/non-formatted;

* The author cite Iorio et al, 2016 when referencing large scale phosphoproteomic portraits of cancer cell lines. However in Iorio et al, no phosphoproteomic data is presented/used;

* While citing published papers dealing with inconsistencies between phenotypic datasets derived from screening large-scale panel of cell lines, the authors include Dempster et al. 2016. However this study reports a fairly good agreement between CRISPR-cas9 based gene-dependency datasets. This might be briefly mentioned, as opposed to the inconsistencies reported in previous analyses of RNAi phenotypic datasets.

* part of the text in the 'Calculation of target addiction score' section is highlighted in grey (?)

* when mentioning the Sanger dataset, a citation to the Cell Model Passports resource (PMID: 30260411), containing the molecular characterisation of the GDSC cell lines (including proteomic data) might be added

* abbreviations in figure 1D are not specified

* The CRISPR-based datasets generated at Sanger Institute are not part of the GDSC but of Project SCORE. This should be corrected

Reviewer #3:

Summary

Jaiswal and colleagues describe in their manuscript a new pipeline for a combined analysis of various omics data sets that were generated for a large number of cancer cell lines by multiple research laboratories. The authors presented their observations on reproducibility of the data obtained by different experimental techniques and suggested a simple, non-parametric methodology that allowed them to aggregate biological evidence, in order to find genes relevant for individual cell lines, tissues or tumor types.

To illustrate usability of their approach, the authors focused on breast cancer cell lines and identified ECHDC1, a gene not previously associated with breast cancer, as a potential tumor suppressor. Moreover, the authors used their framework to discover new synthetic lethality interactions between known cancer drivers and other genes, and found DDX27 and DHX30 to be essential for cells carrying PTEN mutations.

General remarks

The authors made a tremendous effort of collecting a large number of data sets from various databases. To my knowledge, this is the first attempt to combine nine different types of high-throughput omics data (gene methylation, point mutation, copy number variation, mRNA expression, protein levels and phosphorylations, functional profiling, drug sensitivity, etc.).

The work by Jaiswal and colleagues demonstrates that by using simple data aggregation techniques it is possible to extract novel and potentially clinically relevant information. Considering a wide range of omics studies presented in this manuscript, and general remarks on experimental reproducibility, I find it to be of high interest for a broad scientific audience. Generated results, provided as supplementary tables, can serve as a valuable source of information for further exploratory studies.

Nevertheless, the authors should address several issues before their work can be published in *Molecular Systems Biology*.

Major points

1. It is not clear why authors applied different cut-offs for defining copy number gains and losses in different data sets (e.g. {greater than or equal to} 1 and {less than or equal to} -1 in case of "BROAD" data and {greater than or equal to} 0.5 and {less than or equal to} -0.7 in case of "UHN"). At the same time, the authors report a much higher correlation between continuous copy number profiles calculated for identical cell lines, than their binarized calls (page 23 and Supplementary Figure S11). This raises a concern, whether binarization of the copy number data makes sense at all. At the very least, the differences in used cut-offs should be explained.

2. The gene-level methylation profiles were calculated by analyzing promoters of protein-coding genes, but there is a discrepancy in how promoters were defined for different data sets (e.g. 1 kb up-stream of TSS in case of the "BROAD" data-set and 1.5 kb in case of "GDSC"). The authors should either unify the promoter definitions or explain the differences.

3. In order to identify genes carrying deleterious mutations, the authors combined predictions reported by multiple tools. It is a very sensible approach but for the sake of consistency, the same set of methods should be applied to all data sets. If, for any reason, it was not possible, the authors should explain why.

4. I find the data acquisition and pre-processing steps to be very important for the manuscript and highly interesting for the scientific community. In order to make it easier for the readers to understand and evaluate the pipeline, I would suggest including one or multiple supplementary figures depicting in details all methods used for preparing the input data. One scheme for each modality. This would be very useful, especially because data sets provided by different research institutes were not processed identically. The flow charts could include names of the tools used, important parameters and counts of genes and cell lines acquired from each data source.

5. Throughout the manuscript, the authors compare correlations between pairs of results obtained from identical cell lines to pairs of non-identical cell lines (Figure 1D). This approach is highly biased, considering that many non-identical cell lines derived from same tumor types are, in fact, very similar in their profiles. Splitting pairs of non-identical cell lines into two groups: non-identical but same-tissue (e.g. breast), and non-identical and different tissue would put the data consistency in a better perspective.

6. Page 11: The authors defined OES as a z-score, which is a difference between a raw value and a mean, divided by a standard deviation. This is not consistent with Figure 3, where the denominator is omitted. Either the figure should be corrected, or the z-score term should not be mentioned in the text. It also seems to me that OES could as well be defined as a percentile rank of a gene-specific raw value in a given cell line, among all cell lines in the same data set. This would make OES more

robust against heavily skewed distributions.

7. Page 19 and Figure 3: The use of "down-regulated" and "up-regulated" terms is not suitable for many analyzed modalities, and may confuse the readers. For example, high methylation of a gene is, in fact, a sign of its down-regulation. Instead, I would rather suggest referring to CSSup and CSSdown as genes enriched at the top or the bottom of a set of ranked lists of genes.

8. Page 11: In case of cell lines profiled in a single study, rank product analysis was not performed but, instead, an arbitrary number of top scoring genes were selected. In case of almost all continuous modalities, 100 genes were picked. Except functional data sets, for which the top 200 genes were selected. Why the inconsistency? Is it because of noisiness of the gene dependency data? The reason should be mentioned in the Methods section.

9. Page 16: The authors claim that a high variability of CNV correlations between identical cell lines suggest clonal variability of cell lines cultured at different research sites. It would be very interesting to see if such cell lines also show much weaker correlations of mutational profiles. In my opinion, this would make the statement stronger. Additionally, the authors should provide a table of these cell lines. It would be highly valuable for the scientific community.

10. Page 17: Comparison of different technical platforms is a very interesting aspect of the manuscript. However, to be able to draw any conclusions about reproducibility of some technologies, more sets of correlation analyses are needed. For example: in case of phosphorylation profiles, correlations within MS-based profiles are missing. In case of drug sensitivity assays, it is a lack of within-Syto60 correlations (Supplementary Figure S4K and S4L). Similarly, the methylation studies based on bisulfite sequencing and Illumina 27K are missing their respective "within technology" correlations. I assume it is due to a lack of appropriate data sets, but the authors should make it clear for the reader. I would even suggest a significant reduction of the first two paragraphs and even removal of PHOS- and DSS-related statements from the main text.

11. Page 18, line 19 and Figure 2E: Why these two studies were picked? Do they represent a typical correlation between coefficients of variation of NL and TMT profiles?

12. Page 19, line 34-38: The authors claim that some driver genes were selected based on TAS and FUNC modalities and others based on PHOS and CNV. However, judging from Figure 4A, it looks like almost all shown genes could be selected only by CSS scores from TAS and FUNC modalities. The only exception, AKT1 in MCF7 cell line, was selected by scores from PHOS and GEXP studies. Other modalities would not be needed for this hit calling. The authors should correct their conclusions or explain their point better.

13. The authors should measure expression levels of ECHDC1 (e.g. with qPCR) in MCF10A and BT-474 cell lines, as well as knock-down efficiency.

14. Microscopy images are not convincing enough to support the claimed proliferative and invasive phenotype of ECHDC1 knock-down. If possible, a time-course cell count quantification should be provided, or a cell competition assay. Considering reported poor reproducibility of RNAi screens (Figure 2B), the authors should validate the phenotype with an independent silencing trigger, or by analyzing a CRISPR knock-out.

Minor points

1. In general, the number formatting should be corrected to be uniform across the whole text (i.e. digit grouping and use of thousand separators). Also, at numerous places, a non-breaking space should be added between a number and a unit (e.g. page 5, line 8; page 13).

2. Page 3, line 7: The reference needs to be fixed.

3. Page 5, line 34: "Pathogenic" probably should be replaced by "deleterious".

4. Page 8, line 12: It is not clear to which "earlier section" the authors refer to. In previous sections, varying methods were used.

5. Page 9, line 32: "DSS2" should be spelled without a subscript.

6. Page 12, line 38 and page 13, line 5: Exponents at the cell count should be positive.
7. Page 14, line 12: The sentence ending with "from 1 or -1" needs to be corrected.
8. Page 15, line 37. The text says ~215 cell lines, but the Figure S1A shows over 700 cell lines analyzed by a single study site. The number should be corrected. This sentence also refers to Figure S2, which does not fit the text.
9. Page 22, line 41: The sentence refers to Figure S10E, but the figure panel does not match the text.
10. Figure 1: Please, unify the colors between panels. Differences in shades become especially apparent on printouts. References in panel A are not consistent with the journal style.
11. Figure 1B and Supplementary Figure S1A: I would suggest using black bar to avoid confusion with the DSS modality.
12. Figure 2A, 2C and Supplementary Figure S4J: Considering the group labeling, it would be easier to read the matrices if their lower triangles were shown.
13. Figure 2B and 2D: The figure legend should explain that triangles represent means.
14. Figure 3F: Either a gene name should be mentioned, or it should be made clear that this is just an example of how a CLIP signature may look like.
15. Figure 4A: Some cell line labels are not aligned with their respective columns.
16. Figure 4B, 4D, Figure 5A and Figure S10: The percent signs in brackets should be removed. They suggest that upper limits of Y-axes is 1%. Alternatively, multiply labels by 100.
17. Figure 5E: The legend refers to Figure S7, which does not show metabolite levels across 14 cell lines.
18. Figure S6: Descriptions of "columns" and "rows" in the figure legend should be swapped.
19. Table S2: Typo in the header ("each genes").

Response to Reviewers comments

Reviewer #1:

Summary:

The authors provide an extensive meta-analysis of 53 omics profiling studies in more than 2000 cancer cell lines processed in 12 research laboratories, considering for each cell line up to 9 different data modalities. They describe a pipeline, which they name CLIP, for identifying robust cancer cell line specific (rCCS) genes that may be relevant to certain cancer types and use the rCCS data to infer synthetic lethal (SL) pairs. The study further provides a systematic correlation analysis between different methods within the same data modalities (eg. microarray vs RNA-seq) in order to identify inconsistencies observed for certain data modalities. They use the pipeline to identify a putative tumor suppressor ECHDC1 in breast cancer and provide some in vitro validation.

Aside from the identification of ECHDC1, the most useful contribution of the study to the cancer community would be full access to the meta analysis results, and ideally both in the form of full results, and with a release of the pipeline code. Though the authors have done a partial release of the results as supplementary tables, they don't suffice because unless we are mistaken one cannot fully trace the rCCS designations fully back to the modalities and data used to make the designations. There is a github site, but it appears to have no instruction or code, and the data on the site is in a format (Rdata) that is not universally friendly. For useful exploratory analysis based on the results, cancer researchers would need more than the authors have provided.

Response:

We thank the reviewer for appreciating the importance of our multi-omics data resource and the meta-analysis integration approach. Please find our responses to each point below.

Major comments:

1. The authors failed to provide the necessary code and instructions to reproduce their analysis. Without instructions on how to use the processed data on their github page, it is impossible to judge the reproducibility of their analysis and whether it was properly implemented.

Response:

We thank the reviewer for raising this concern. We have now extensively updated our code on Github website (<https://github.com/jaiswal-alok/clip-meta>), with scripts for performing reproducibility analysis (in the folder *reproducibility_analysis*), and also the code for running the CLIP pipeline (in the folder *clip*). We also provide detailed instructions on how to use the processed datasets, and execute the CLIP pipeline so that it can be used by the community. The reason the datasets have been provided as *Rdata* files is because the csv format requires much more memory for the large data files. Github does not allow files with sizes > 100 Mb, and most of the CCLE and SANGER GEXP and CNV data would easily go beyond that in csv format.

2. The methods can often be vague. For instance what does it mean to say "Fisher's exact test was performed to evaluate the difference in the proportion of rCCS genes between different cancer subtypes or pre-defined subgroups of cancer cell lines." We understand the statistics, but don't understand the intended meaning of the sentence.

Response:

We apologize for the poor wording in the original manuscript, and have now updated the statements to better reflect the meaning we want to convey. For instance, this sentence has been reworded (on page 12): "Fisher's exact test was performed to identify the genes that are enriched in different breast cancer subtypes or pre-defined subgroups of cancer cell lines". We have further clarified many places of the Methods section (please see the changes made that are highlighted in red text).

3. Did this method (robust, multi-modality) really lead to the identification of ECHDC1, or would analysis of methylation data from a single site have sufficed? The authors need to make the case.

Response:

This is a good point. Using the multi-modal meta-analysis approach CLIP, we were able to identify ECHDC1 as a robust CCS (rCCS) gene because it was identified as CCS gene in both methylation and gene expression modalities. If we only had looked at CCS hits from the methylation data from a single study, this would lead to many CCS genes, making it impossible to prioritize the robust CCS genes from a long list of putative hits. By forcing the constraint that a rCCS must be identified as a CCS gene in at least two modalities, CLIP improves the likelihood of identifying true cancer associated genes.

To address the reviewer's valid concerns, we devised an alternative and more simplistic approach to test whether ECHDC1 is identifiable in methylation data from single site only. Specifically, we performed the following steps on breast cancer cell line methylation data from each site to follow the CLIP preprocessing approach without the multi-modal and meta-analysis option:

1. Quantile normalization of gene-averaged methylation levels as performed in CLIP.
2. Z-scaling of the methylation levels for each gene as performed in CLIP.
3. To identify the outlier genes, we selected all the genes that had z-scores above or below 1.66 standard deviations from the mean z-score in each dataset separately (this corresponds to ~90% quantile when the z-scores are normally-distributed).
4. Following this, we checked the frequency of ECHDC1 being identified as an outlier gene in any of the breast cancer cell lines that were profiled in the dataset.
5. We also provided the code of our analysis in Github for the reviewer's perusal (in the folder *clip-meta/R/clip*).

In summary, we implemented the above pipeline on methylation datasets from BROAD and GDSC. As a result, ECHDC1 was identified much less frequently as an outlier in any of the site-specific methylation datasets (Figure S3 E). This new comparison method is detailed in Methods (pages 13-14), and the result is mentioned briefly in the revised Results (page 26) and Discussion (page 29-30).

Moreover, we also challenged our CLIP approach using an alternative, previously-published approach for integration of multi-omics dataset, named MOFA+ method (Argelaguet *et al*, 2020), which combines datasets from a single research site. Because ECHDC1 was supported

by GEXP and METH modality from CLIP, we applied MOFA+ method independently to all the datasets that profiled for methylation in breast cancer cell lines, namely BROAD, OHSU and GDSC. Because GEXP modality data from BROAD and OHSU were in RPKM units, log-transformation was done before input into MOFA+. Otherwise, the MOFA+ method was run using its default options.

Specifically, we performed the following steps:

1. Each dataset was quantile normalized and subset to the set of breast cancer cell lines, before training the unsupervised MOFA+ method under the default settings with the number of latent factors initialized to 15.
2. For BROAD dataset, the following modalities were considered as input for MOFA+: CNV, GEXP, METH, MUT, FUNC_AVANA, FUNC_ACHILLES (see Fig. 1a for the descriptions). The other modalities such as PHOS and TAS were not considered because of very low number of features compared to the other modalities.
3. For GDSC dataset, the following modalities were considered as input for MOFA+: CNV, GEXP, METH, MUT, FUNC. The MOFA+ method was trained under the default settings with the number of factors initialized to 15.
4. For OHSU dataset, the following modalities were considered as input for MOFA: CNV, GEXP, METH, MUT. The MOFA+ method was trained under the default settings with the number of factors initialized to 15.
5. Plots of the proportion of variance explained for each group by each data modality/view for 15 latent factors were inspected (Figure S3), and the factors and views with the highest variance were then selected further.
6. Top 20 (top 10 positive weights + top 10 negative weights) view specific-features loading on to each factor were extracted from the method.

In summary, we did not observe that ECHDC1 was loading into any of the top factors in the models from any of the three sites, suggesting that MOFA+ was unable to identify ECHDC1 (Figure S3 A-D) as a latent factor shared across data modalities and also driving heterogeneity in breast cancer cell lines. This new comparison method is detailed in Methods (page 13-14), and the MOFA+ result is mentioned in the revised Results (page 26) and Discussion (page 29-30).

We have also provided our codes for conducting these analyses for the reviewer purposes in the `clip-meta/R/clip/clip_benchmark.R` file in the GitHub repository (<https://github.com/jaiswal-alok/clip-meta>).

4. We think the authors did not provide sufficient validation to show that their context-specific synthetic lethal relationships found between tumor suppressor PTEN and RNA helicase DDX27 and DHX30 has a clinical relevance. For instance, the authors could easily mine transcriptomic data for endometrial cancers (Uterine Corpus Endometrioid Carcinoma, 560 cases) in TCGA and test whether patients where both PTEN and DDX27 is downregulated within the same patient compared to patient where this is not the case, show a better survival.

Response:

We thank the reviewer for this great suggestion. To further investigate the synthetic lethal relationship between PTEN and RNA helicases, which was identified by CLIP, we performed a survival analysis on TCGA data as suggested. We classified patients into two groups: high expression of both PTEN and DDX27, $PTEN^{hi}DDX27^{hi}$ (n=129), and low expression of both

PTEN and DDX27, PTEN^{lo}DDX27^{lo} (n=112) based on their mean mRNA expression levels. As expected, we observed that patients with lower mRNA levels of PTEN and DDX27 had better survival probability albeit the statistical significance was nominal (P = 0.1). Interestingly, we observed an opposite trend between DDX30 and PTEN expression levels, i.e., PTEN^{hi}DDX30^{hi} patients had poorer survival compared to the PTEN^{lo}DDX30^{lo} group. We provide the survival analysis plot as the new supplementary Figure S18 E-F, and added the following sentence in the results section of the revised manuscript (page 28): “In support of this observation, patients with downregulated PTEN and DDX27 tended to have better survival probability, compared to patients where this is not the case (Figure S18 E), and the trend was reversed for DDX30 and PTEN (Figure S18 F)”.

5. We think the survival analysis shown in Figure 5B is not convincing. First, Figure 5B does not show that increasing methylation leads to poor survival -- the highly methylated group fares worse than the low methylated group. The authors only compare medium and high. Second, the authors did not account for the most common confounding variables (age, sex and race). The authors should repeat the analysis using a cox proportional hazard regression model and account for these confounding factors. Finally, the most natural data to test for proportional hazards for this gene might be gene expression, and Figure 5A at least suggests that gene expression is available.

Response:

We concur with the reviewer’s assessment that the results shown in Figure 5B are somewhat difficult to interpret, which is a quite general trend in this kind of survival analyses. To simplify the analysis, we further classified the patients into two groups: high and low methylation based on the mean methylation intensities. In a Cox regression analysis adjusted for age and race, we did not observe a statistically significant difference in the survival probability between the two groups of breast cancer patients in the TCGA cohort. Similar results were observed when the patients were categorized into two groups based on gene expression levels. We reasoned that this could be because of the censoring due to poor follow up and a lower median survival, a characteristic of the TCGA BRCA cohort that has been also observed by the TCGA authors (PMID: 29625055).

Therefore, to perform the survival analysis in breast cancer patients with higher statistical power, we combined two breast cancer patient cohorts with 20 year high-quality follow-up data and molecular profiling: the Metabarc cohort (Curtis *et al*, 2012), which contains around 2000 breast cancer patients with gene expression data, and the Oslo cohort (Fleischer *et al*, 2014, 2017), which contains 334 breast cancer patients with DNA methylation data. For the patients with gene expression data, the patients were divided in tertiles based on the expression of ECHDC1, and the upper tertile was denoted high GEXP group while the lower tertile was denoted low GEXP group. For the patients with DNA methylation data, an average DNA methylation level for all pre-annotated indices of CpG sites within the range of 200 to 1500 bp upstream of the TSS was calculated for each patient, and the patients were divided in tertiles; the upper tertile was denoted high METH group and the lower tertile was denoted low METH group. For the survival analysis, the patients were stratified as either 1) high GEXP or low METH or 2) low GEXP or high METH and compared for breast cancer specific survival (BCSS) using a log-rank test. To assess the potential confounding with age at diagnosis, a Cox proportional hazard multivariate model with age (above or below 60) as covariate was performed. The sex and race were not considered as relevant factors in Nordic breast cancer patients. Race information was not available for the Metabarc patients. In this combined dataset, we observed a strong statistically significant decrease in the survival probability in breast cancer patients that belong to the low GEXP or high METH category (P=0.007). Moreover, we also

observed a similar result within the subsets of the breast cancer patients, defined as ER+ or ER-, demonstrating the robustness of the association (Figure S13).

We think this new analysis is of much better quality, hence we have replaced the original Figure 5B with the corresponding figure from the new breast cancer cohort meta-analysis, and added descriptions to the Results (page 26) and Methods sections (page 16).

Reviewer #2:

The authors present a computational integrative approach aiming at harmonisation and joint analysis of data from the chemo/functional-genomic characterisation of cancer in vitro models spanning 9 different data modalities, and 12 different studies.

The main aim of the presented approach is to dilute/remove biases and inconsistencies due to different laboratory specific experimental practices and analytical protocol across the considered studies and to provide a unified analytical framework for the identification of cancer-driver multi-omic signatures, hence cancer driver genes and genomic-context specific genetic dependencies.

After converting all the analysed data omics at the level of gene patterns, the authors devised an elegant strategy to quantify the 'signal-strength' of each gene in each cell line across each omic, then they computed cell line specific (CSS) gene scores based on how consistent their signal-strength is in a given cell line across considered omics.

Then they show how the resulting signatures of top CSS genes are predictive of established and novel cancer driver genes, and can be used to efficiently predict synthetic lethal gene pair.

Very briefly, the problem tackled by the authors is important and timely, and the attempt they made to devise a method to unify/harmonise multiple data omics across different studies should be lauded for its simplicity and elegance.

In addition, the discovery of a new tumour suppressor genes for breast cancer and synthetic lethal gene pairs add value and demonstrate the utility of the presented approach. However I have a number of major concerns (detailed below) preventing me to warrant publication of this manuscript in Molecular Systems Biology, at least not before a convincing extensive revision.

Response:

We thank the reviewer for appreciating the relevance of our multi-omics data integration approach and the novel findings made. Please find our responses to each point below.

1. First of all, the authors have included a systematic comparisons of data omics across studies, whose purpose or utility is left unclear. In fact they do not attempt any computational reduction of the observed discordance across studies for certain omics nor they use any of the findings from this analysis to guide the follow up analytical choices.

In addition, as the author themselves demonstrated in a previous publication, the major problem with the assessments of consistency in drug screening and cancer dependency datasets is failing to account that often that the lack of agreement is due to a lack of variance in the observed drug-response/gene-essentiality signals. That's why for example in a recent

publication measuring the agreement between cancer dependency datasets [PMID: 31862961] correlation metrics employed to compare gene dependency profiles across studies are weighted by the 'context-specificity' of gene essentiality (as never-essential/always-essential genes would add noise only to these estimation).

In the light of these consideration, it is surprising that here the authors use crude Pearson's correlations and Mathew Coefficients to compare omics across studies, without further digging into the identified discordancies, trying to reduce them or gaining insights for the follow up integrative analysis.

Response:

We thank the reviewer for this excellent comment on the assessment of consistency of omics data. Our rationale for performing the systematic comparisons of reproducibility between multiple studies was to first get a comprehensive understanding of the level of variability between the multiple modalities profiled in the datasets. This is highlighted in the Discussion (page 30). Based on what we learned from these comparative analyses, we used the information to inform our choices on the false discovery rate (FDR) thresholds for selecting the CCS genes based on different modalities. For instance, in most of the continuous data modalities, we used an FDR threshold of < 0.10 to select CCS genes, whereas in the FUNC (gene essentiality) modality, we used the cutoff of $FDR < 0.25$ pertaining to higher noise in the functional profiles. Similarly, in the remaining continuous modalities, we selected the top 100 genes if a cell line was profiled in only one dataset, except for functional gene essentiality data sets, where we selected top 200 genes as CCS genes for each cell line due to higher noise of functional profiles. These analytical choices were made based on the initial analysis of the level of variability and missing data points in the studies and data modalities. This is now emphasized in the revised manuscript (several paragraphs in Methods section on page 11, and Results, pages 20 and 24).

While it is true that this study did not directly attempt to develop a computational approach to reduce the discordance of the datasets before their multi-omics integration steps, the CLIP pipeline that we developed does consider the variability in observations between multiple studies and datasets. The CLIP framework relies on a meta-analysis approach, that accounts for the variability in the observed measures for a gene across multiple studies using the rank product integration approach. Moreover, the reason we did not directly develop a computational approach to reduce the discordances is because there are already a number of previous studies that have tried to do that for specific omics data (e.g., PMIDs: 31862961, 28928933, 32175273), and we felt it would make the work difficult to follow if we had also included systematic comparison of these approaches in the first part of the work. As pointed out before, our motivation was to use the know-how from the comparative evaluations to inform the noise parameters of the integration pipeline to go on and identify novel cancer-related genes and synthetic lethal interactions. Another reason for not trying to reduce the discordance of the datasets was that by using the data directly from the original studies (and basically just implementing quantile normalization and other simple preprocessing steps beyond that), the data integration CLIP approach is easily extendable also to future omics data and studies, without the need to always test and implement specific discordance reduction strategy for each data modality separately. These aspects of the approach are now better clarified in the revised manuscript (page 29). We have also we now provided the details of ours as well as the original authors' data processing and harmonization procedures for multi-omics data in the new Figure S4. We believe these steps will become useful for the community for extending the CLIP approach to other omics data profiles from emerging studies, as well as for developing other

types of meta-analysis approaches that require integrating multi-omics original data from multiple studies.

The authors of Dempster et al. 2019 [PMID: 31862961] employed a batch correction method to reduce the discordance between two functional screens based on CRISPR-Cas9 knockout generated at BROAD and SANGER. The experimental technique and platforms employed for generating the functional profiles for these two specific datasets are largely similar, hence a batch corrected method such as ComBat can be easily applied because the distribution of measured variables is largely similar between these two datasets. However, in our study, we have used datasets generated from various platforms from multiple studies, both for genomic/molecular and functional profiling, which therefore have very different distributions of measured values. For instance, the distribution of array-based gene expression measurements is quite unlike RNA-seq based gene expression measurements, not to speak of binary point mutations or CNV profiles. Hence, we believe it would be impossible to perform a computation reduction using the ComBat method in a case like ours. Nevertheless, we have now cited the work of Dempster et al. 2019 in the revised manuscript (page 30), while discussing the practical benefits of the CLIP approach.

We also think that by using the original data from various sites as original as we can, and by using rather simple comparison metrics, such as Spearman's correlation and Matthews correlation coefficient, that are easy to interpret, we keep the comparative analyses as unbiased and objective as possible, and minimize the speculation that we would favor any of the specific data modalities, for instance, drug testing or gene essentiality, that we know best from our previous studies. Several studies have utilized such metrics for comparison of omics datasets (PMIDs: 24284626, 22460905). We would also like to point out that we have not used the Pearson's correlation coefficient anywhere in our analyses, mainly because the distributions of measured variables are quite heterogenous depending on the data modality and technical platform used for the measurement. Moreover, we show that even such simplistic comparisons can lead to many unexpected findings. For instance, for the PEXP modality, we found the unexpected lack of consistency between MS-based proteomic profiles based on different techniques (Figures 2D-E). We also show how data normalization contributes to variability in consistency (Figure S9). In addition, for the drug sensitivity modality, we also found that the use of different experimental assay systems for profiling cell viability in drug screens, either Cell Titer Glo (CTG) or Syto60 contributes to differences in reproducibility (Figure S8 L). These novel results based on the correlation analyses are described in Results (page 22) and Discussion (page 29).

2. Moreover, in this work there is a general lack of performance comparisons when considering the ability in detecting cancer drivers, between the CLIP signatures and the individual omics (when used alone). The authors should compulsorily show, for example, that the hits coming out from contrasting CLIP scores across breast cancer subtype are in a larger number or more reliable or with a larger Recall of true cancer drivers, with respect to using for example expression, or methylation or cnv or gene dependency alone. In addition, comparisons with other multi-omic integrative methods, such as MOFA (PMID: 29925568) for example, should be included, or at least discussed if this is not feasible.

Response:

To address the request of benchmarking the performance of the multi-modal CLIP against the use of single data modalities alone, we pursued the following approach:

1. First, we defined the set of “gold standard” true cancer driver genes by sub-setting the known cancer driver genes from a recent study (Bailey et al. Cell 2018), to those that were detected in either pan-cancer tumors (PANCAN) or breast tumors (BRCA). This gave us a total of 201 cancer genes.
2. Then, we analyzed each modality separately to assess the true positive identification rate of drivers for each breast cancer subtype. Briefly, we performed differential analysis between “subtype+” and “subtype-” groups of cell lines for each modality from every site using the *limma* package.
3. For each modality, we merged the differential analysis results from various sites using the rank product analysis to identify robust differential genes based on each modality separately. Based on the dataset from each individual site, we obtained the B-statistic for every gene from differential analysis for a given subtype (“subtype+” vs. “subtype-”). Then, we performed the rank product analysis on the B-statistic and selected genes with $pdf < 0.05$.
4. One could ask, why not to perform a rank product analysis of log-fold changes. This is because uncertainty in fold-changes cannot be easily quantified, whereas B-statistic is a measure of statistical confidence for each gene. However, after merging the B-statistic from each study, we then considered the general direction of fold change, i.e. average fold change.
5. We further subset the genes based on their average log fold-change for each omics modality. For GEXP and PEXP modality, only “up-regulated” genes in “subtype+” cell lines were considered as driver genes. Similarly, hyper-methylated genes for METH modality, amplifications for CNV modality, and essential genes for FUNC modality in “subtype+” cell lines were considered as driver genes.

In summary, we observed that CLIP identified a higher fraction of the known true positive cancer driver genes (Figure 4G), when compared to each individual data modality alone for all the breast cancer subtypes. We have now described this analysis strategy in the methods (page 12-13), and the results in the revised results section (page 25) and discussion (page 29-30).

As requested, we have also benchmarked the performance of CLIP against the well-known method for multi-omics integration called MOFA+ (Argelaguet *et al*, 2020). While MOFA+ has been primarily applied for integrating multi-omics datasets from a single site, in principle it can also be applied to multi-omics data integration from multiple sites. However, due to the differences in distribution of the omics measurements, resulting from the various technological platforms that were used at the various sites for generating the datasets, we chose to apply MOFA+ to breast cancer cell lines from BROAD, GDSC, OHSU and UHN. Furthermore, we reasoned that the multi-group framework in MOFA+ could be used to identify the latent factors that are specific to each breast cancer subtype, even though the authors of MOFA+ acknowledge that the multi-group framework is not designed for capturing differential changes between the groups, but rather to identify the principal sources of variability that drive each group.

Specifically, we pursued the following strategy for the datasets from each site:

1. For BROAD, we considered CNV, MUT, METH, GEXP, PEXP, FUNC_CRISPR, and FUNC_RNAI data modalities. Each dataset was quantile normalized and subset to the set of breast cancer cell lines, before running the multi-group MOFA+ model using its default settings.

2. Plots of the proportion of variance explained for each group by each data modality/view for 15 latent factors were inspected (Figure S2, and factors and views with the highest variance were then selected further).
3. Top 20 (top 10 positive weights + top 10 negative weights) view-specific features loading on to each factor were extracted from the MOFA+ model.
4. The list was then subset to the established gold standard driver genes a recent study (Bailey et al. Cell 2018).

In summary, we observed only a few overlaps between the features with gold standard driver genes and those identified by MOFA+. In the BROAD dataset, we observed 5 genes that were identified by the MOFA+ model. However, the model was not able to identify any of the gold standard driver genes in the other datasets (Figure 4G). These results suggest that MOFA+ framework may not be optimal for identifying subtype-specific driver genes.

Please also see our reply to the comment 3 of reviewer 1, where we demonstrate that MOFA+ could not identify the novel breast cancer gene, ECHDC1, which was identified by CLIP and experimentally validated in our work. These comparative results are now described in the methods section (page 12-13), and in the revised results (page 25) and discussion (page 29).

We have also provided our codes for conducting these analyses for the reviewer purposes in the *clip-meta/clip* folder of the GitHub repository (<https://github.com/jaiswal-alok/clip-meta>)

3. Also the claim that CLIP is able to identify known oncogenic addictions is not supported by any systematic analysis. The authors should compulsorily show that cell line specific rCCS genes are indeed enriched for essential genes. Ideally, keeping the gene-dependency omic out of their integration effort, at what extent the rCCS gene rank in each cell line is predictive of that gene essentiality in the same cell line? This should be quantified, for example by means of ROC indicators. A similar analysis should be performed to assess the ability to detect associations between paralogs, such as the mentioned AIRD1A/B case but, again, more systematically

Response:

We are not aware of any comprehensive listing of oncogenic addictions in breast cancer tumors that could be used for such systematic analysis. We would appreciate if the reviewer pointed out such a resource. We do not think we should use essential genes of cell lines as the true oncogenic addictions of breast tumors, as the cell line addictions are different from those of patient tumors and the definition of cell line addictions is dependent on the profiling technology (e.g. CRISPR or RNAi) and the metrics used. Therefore, when investigating the clinical relevance of the CLIP predictions, that were made in the cell lines, we chose to use patient tumor data such as TCGA or METABRIC cohort instead.

To address the first half of the request, regarding the ability of CLIP to identify known oncogenic addictions, we followed a strategy outlined below:

1. We defined the set of known oncogenic addictions as known cancer driver genes from the study of Bailey et al. 2018, as described above.
2. After sub-setting to pan-cancer PANCAN and epithelial tumour types, we had a total of 227 known oncogenic driver genes.
3. Next, to benchmark CLIP, we counted the frequency of each rCCS gene across all the epithelial cancer cell lines. We reasoned that the true oncogenic addictions will be identified more frequently as a rCCS gene, when compared to the non-oncogenes.

4. We ran the CLIP pipeline on the subset of epithelial cell lines, in the following three settings to identify rCCS genes:
 - a) All modalities: including datasets of all modalities (the number of epithelial cell lines is 737)
 - b) FUNC compulsory: including datasets of all modalities, but constraining rCCS gene selection criteria by compulsory identification as a rCCS gene in the FUNC gene essentiality modality (the number of epithelial cell lines is 679)
 - c) FUNC excluded: including datasets of all modalities, except for FUNC gene essentiality modality (the number of epithelial cell lines is 736)
5. Finally, we compared the differences in rCCS frequency between the oncogenes and non-oncogenes groups in each setting using the Wilcoxon test.

Overall, we observed that the known oncogenes were more frequently identified as rCCS genes by CLIP (Figure 6E and S17 C-D). Moreover, the performance of CLIP was not much reduced even after removing the FUNC modality. The option of constraining the rCCS gene selection criteria by compulsory requiring that the gene was identified with FUNC gene essentiality modality performed actually the worst, indicating that cell gene essentiality alone may not be the best predictor of breast cancer oncogenes. We also repeated the analysis for all the driver genes, i.e., including both oncogenes and tumor suppressor genes, a total number of 631 driver genes, and only on tumor suppressor genes (TSGs) (Figure S17 C-D). We observed a similar trend in both cases.

This new comparison is detailed in Methods (page 14), and the results described in Results (page 25) and Discussion (page 30-31).

To address the second half of the request, regarding ability of CLIP to identify synthetic lethal associations between paralogous genes, we carried out the following steps:

1. We obtained the list of paralogous genes in the human genome from the Duplicated Genes Database (DGD) (Ouedraogo *et al*, 2012). In total, we ended up with a list 3543 paralogs in the human genome that are constituted by 945 unique paralog groups. For instance, ARID1A/ARID1B are part of OR form a unique paralog gene group. Similarly, STAT1 and STAT4 are another unique group.
2. Next, we analyzed the rCCS genes from CLIP runs for the genetic interaction analysis on epithelial cancer cell lines, as described in the original work. Specifically, we conducted a Fisher's exact test to assess the difference in proportions for rCCS genes between mutant and WT cell lines for each paralog group. For example, if ARID1A is mutated, then we asked whether its paralog, ARID1B, is enriched in proportion as a rCCS gene in mutants compared to WT cell lines. Or, vice versa, i.e., if ARID1B is mutated, then we evaluated ARID1A difference in enrichment.
3. We performed the genetic interaction analyses on paralogs that are mutated in at least 10 epithelial cell lines, so that we have sufficient power to detect robust associations.
4. If any gene from a paralog group was detected as statistically significant, we counted that as a paralog SL/GI association.
5. Then, we plotted the proportion of number of unique paralog groups that have a paralog SL/GI association in relation to the total number of unique paralog groups that were tested.
6. We ran this analysis in different settings of CLIP:
 - a) Running SL analysis on rCCS genes from CLIP on all modalities, but rCCS gene should be compulsorily identified as a rCCS gene in the FUNC modality (number of

- epithelial cell lines is 679). This is the default setting to identify SL relationships. A gene has to be essential and rCCS.
- b) Running SL analysis on rCCS genes from CLIP on all modalities (number of epithelial cell lines is 737). Since, essentiality is not mandatory criteria for this rCCS calling, these associations can be generally categorized as genetic interactions
 - c) Running SL analysis on rCCS genes from CLIP by including all modalities, except the FUNC gene essentiality modality (number of epithelial cell lines is 736). Since, essentiality is not included as criteria for rCCS calling, these associations can be generally categorized as genetic interactions.
7. To benchmark the performance of CLIP, we performed SL analysis in each individual FUNC dataset (BROAD_ACHILLES, BROAD_AVANA, DRIVE, UHN and SANGER) to compare the performance for detecting paralog SL associations. Using the *limma* package, we performed a linear model differential essentiality analysis between mutant vs. WT cell lines for their difference in gene essentiality scores. As described above, we subset to paralogs that are mutated in at least 10 epithelial cell lines, so that we have sufficient power to detect robust associations. P-value < 0.05 was considered statistically significant (Figure S16C).
 8. To compare the differences in proportions using multiple strategies, we performed two proportions z-test for difference in proportions of every setting relative to the CLIP (FUNC compulsory) setting.

We observed that none of the proportion comparisons were statistically significant, although the proportion of SL associations detected in individual datasets were systematically lower with nominal statistical significance (Figure S17E). These results show that even after excluding the FUNC modality, CLIP is able to systematically identify genetic interactions between paralogous genes. We have now detailed the comparison approach in Methods (page 15), and described these results in the Results (page 27).

Other major points:

4. The author used inconsistent and seemingly arbitrary thresholds across datasets to call for copy number amplification/deletions. They should specify how these threshold values were picked and why.

Response:

We thank the reviewer for raising this point. The categorized CNV modalities (CNV_AMP and CNV_DEL) were used as inputs in the CLIP pipeline, instead of the continuous data. The primary reason for such discrepancies between the datasets to call for copy number amplification/deletions is because the scaling of copy number intensities is rather different from one dataset to another. Moreover, like mentioned above, we wanted to use the processed data from each study, rather than re-processing every dataset from raw data with our own processing pipeline that may be sub-optimal for a particular study (i.e., we trust the original authors of the studies know best how to preprocess their data). Notably, while datasets from BROAD, OHSU, UHN and NCI60 have been originally processed using the CBS algorithm, data from GDSC and gCSI have been processed using the PICNIC algorithm. Based on the initial consistency analysis, we had previously selected the thresholds that maximized the overlap of amplifications and deletions between identical cell lines from different study sites.

Nevertheless, to make the approach more systematic and unbiased, we have now looked at the copy number distributions of two well-known CNV alterations in breast tumors; ERBB2 which is known to be frequent amplification, and PTEN which is known to be a frequent deletion in breast cancer patients_(Koboldt *et al*, 2012). For ERBB2, we looked at the distribution of its copy number values across all cell lines in each dataset (Figure S1). We then selected the threshold for calling an amplification in each dataset at the value where we observed a sharp deflection. For BROAD, this analysis led to the threshold of ≥ 1 for calling amplifications. For, GDSC, gCSI and OHSU, values ≥ 0.5 ; and for UHN and NCI60 value ≥ 0.4 were considered amplifications. Likewise, for calling deletions; BROAD < 1 ; GDSC, gCSI and OHSU < 0.5 ; and UHN and NCI60 < 0.4 were selected based on this analysis. We further executed CLIP at various thresholds, and did not observe significant differences in its ability to identify breast cancer subtype-specific driver genes that we used for benchmarking the performance of CLIP.

We have now added a new section describing the details of our rationale for the CNV threshold selection for the CLIP approach (page 12).

5. Differently from the other omics, for which the authors extracted data processed via the protocol adopted in their native study, the drug response datasets from multiple studies where commonly extracted from a unique online resource (PharmacDB), which applies its own processing method. This introduce and extent of inconsistency and makes the integrative effort presented by the authors not 100% complete. This choice should be motivated.

Response:

The drug-response datasets used in PharmacDB were analyzed with a common method, drug sensitivity score (DSS v2), which was developed by our group, and have been shown to lead to consistent drug sensitivity profiles between multiple cancer cell line screens (Mpindi *et al*, Nature 2016). Therefore, the use of the PharmacDB did not lead to further inconsistencies, but this database was mainly used because there the cell line names were harmonized. This is now mentioned in the methods section (page 5). We believe the PharmacDB database will be also remain in the future the main resource of drug response sensitivity data from cell lines, based on which one can apply specific drug response metrics suitable for the particular meta-analysis and multi-omics approaches.

6. Not clear what is the criteria used to pick the genes with high/low CSS and the subset of Breast cancer cell lines, to be shown/dicussed a more systematic assessment should be performed in order to show that indeed the proposed approach confirms established cancer type specific drivers, with True/False-positive rates

Response:

We hope we have addressed this comment in our aforementioned response to the comment #2, where we found that CLIP was able to systematically identify a higher proportion of breast cancer driver genes compared to the other comparison methods. These comparisons are now described in the methods (page 12-13) and the results in the results sections (page 25, Figure 4G) and discussion (page 29-30).

7. At lest from the picked example, it seems that the proposed approach seems to be biased toward the discovery/confirmation of gain-of-function drivers. Arguably this might be true, as gain-of-functions mutations, might lead to constitutive expression hence sustained protein

abundance and dependency, is this the case? I think that this would deserve a dedicated analysis/investigation

Response:

To address this question, we took a similar approach as described earlier in section for benchmarking the performance of CLIP in identifying oncogenic additions (Figure 6E). Gain-of-function driver genes are usually categorized as oncogenes, whereas loss-of-function driver genes are typically characterized as tumor suppressor genes (TSGs). We observed the CLIP does a good job at identifying both classes of cancer driver genes (Figure S17C-D). Interestingly, in the setting for compulsory evidence of rCCS from FUNC modality, we observed that the difference in the frequency between known/true TSGs and non-TSGs was reduced, although it was still statistically significant. This suggests that the support for rCCS evidence for TSGs might actually come from non-functional modalities.

8. In addition, aren't the CCS strength biased toward concordant expression/proteomic level? (aren't these expected to be more frequent with respect to other pairs of omics?)

Response:

To address this question, we looked at the modality sources of rCCS evidence for each gene from the CLIP run of all cancer cell lines. Specifically, for each cell line, we counted the number of evidences from each data modality for each rCCS gene that was identified by the CLIP pipeline. Next, to get a relative estimate per modality we divided the number of rCCS genes for each cell line by the average of the number of genes from multiple datasets that were used in CLIP for each specific modality. This was done because the number of genes per modality varies because of the differences in the size of datasets, for instance PHOS and TAS profiles are available for much lower number of proteins (Figure 4B). We observed that the TAS and PHOS modalities were the most frequent source of rCCS evidence among the single omics data. Further, to investigate the inter-modality relationships, we inspected for all rCCS genes that were supported by the GEXP modality the frequency of the other supporting modalities (Figure S12). Interestingly, we found that the GEXP + CNV rCCS genes were the most frequently identified across the cell lines, followed by the GEXP + METH, GEXP + PEXP and GEXP + FUNC combinations. We therefore conclude that the concordant expression/proteomic profiles do not dominate the rCCS identifications, rather multiple modalities are needed for the best results. We have now added descriptions of these analyses in the manuscript in the Results section (page 24)

9. It is not clear how the authors converted the proteomic/phosphoproteomic datasets at the gene-level domain as done for the drug response dataset. I would expect they have used an approach similar to the TAS method, but a description of this in the methods is totally missing

Response:

We averaged multiple phospho-peptides corresponding to the same protein to generate protein-level phosphorylation estimates. We thank the reviewer for pointing this out, as this was previously mentioned in the methods description section for datasets from Sanger Institute only, but we have now added also the text to other sections where phospho-proteomic datasets were used (page 5,7,8 and Figure S4).

10. Finally, I have the following minor points to highlight:

** references are inconsistently formatted/non-formatted;*

Response: We have now made the reference formatting systematic.

** The author cite Iorio et al, 2016 when referencing large scale phosphoproteomic portraits of cancer cell lines. However in Iorio et al, no phosphoproteomic data is presented/used;*

Response: We thank the reviewer for pointing out this error. We have removed the reference of Iorio et al, 2016 for phosphoproteome profiles. We now refer to: Shankavaram et al, 2009; Ghandi et al, 2019; Barretina et al, 2012; Daemen et al, 2013; Marcotte et al, 2016 when citing studies for phosphoproteome profiles. Iorio et al, 2016 is cited for genomic, transcriptomic, epigenomic and drug sensitivity profiles.

** While citing published papers dealing with inconsistencies between phenotypic datasets derived from screening large-scale panel of cell lines, the authors include Dempster et al. 2016. However this study reports a fairly good agreement between CRISPR-cas9 based gene-dependency datasets. This might be briefly mentioned, as opposed to the inconsistencies reported in previous analyses of RNAi phenotypic datasets.*

Response: We thank the reviewer for highlighting this point. We have now added the following statement in the Introduction section (page 3), " while the CRISPR-Cas9 based gene dependency screens have been shown to have fairly good agreement (Dempster et al., 2019)".

** part of the text in the 'Calculation of target addiction score' section is highlighted in grey (?)*

Response: We have corrected the text formatting (page 9).

** when mentioning the Sanger dataset, a citation to the Cell Model Passports resource (PMID: 30260411), containing the molecular characterisation of the GDSC cell lines (including proteomic data) might be added*

Response: We added the article as a new reference Van Der Meer et al, 2018 (page 5).

** abbreviations in figure 1D are not specified*

Response: We thank the reviewer for pointing this error out. We have added the abbreviations to the figure legend.

** The CRISPR-based datasets generated at Sanger Institute are not part of the GDSC but of Project SCORE. This should be corrected*

Response: We thank the reviewer for pointing out this error out. We have modified the method descriptions to accommodate this fact (page 6).

Reviewer #3:

Summary

Jaiswal and colleagues describe in their manuscript a new pipeline for a combined analysis of various omics data sets that were generated for a large number of cancer cell lines by multiple research laboratories. The authors presented their observations on reproducibility of the data obtained by different experimental techniques and suggested a simple, non-parametric

methodology that allowed them to aggregate biological evidence, in order to find genes relevant for individual cell lines, tissues or tumor types.

To illustrate usability of their approach, the authors focused on breast cancer cell lines and identified ECHDC1, a gene not previously associated with breast cancer, as a potential tumor suppressor. Moreover, the authors used their framework to discover new synthetic lethality interactions between known cancer drivers and other genes, and found DDX27 and DHX30 to be essential for cells carrying PTEN mutations.

General remarks

The authors made a tremendous effort of collecting a large number of data sets from various databases. To my knowledge, this is the first attempt to combine nine different types of high-throughput omics data (gene methylation, point mutation, copy number variation, mRNA expression, protein levels and phosphorylations, functional profiling, drug sensitivity, etc.).

The work by Jaiswal and colleagues demonstrates that by using simple data aggregation techniques it is possible to extract novel and potentially clinically relevant information. Considering a wide range of omics studies presented in this manuscript, and general remarks on experimental reproducibility, I find it to be of high interest for a broad scientific audience. Generated results, provided as supplementary tables, can serve as a valuable source of information for further exploratory studies.

Response:

We thank the reviewer for appreciating the importance of our multi-omics data resource and meta-analysis integration approach. Please find our responses to each point below.

Nevertheless, the authors should address several issues before their work can be published in Molecular Systems Biology.

Major points

1. It is not clear why authors applied different cut-offs for defining copy number gains and losses in different data sets (e.g. {greater than or equal to} 1 and {less than or equal to} -1 in case of "BROAD" data and {greater than or equal to} 0.5 and {less than or equal to} -0.7 in case of "UHN"). At the same time, the authors report a much higher correlation between continuous copy number profiles calculated for identical cell lines, than their binarized calls (page 23 and Supplementary Figure S11). This raises a concern, whether binarization of the copy number data makes sense at all. At the very least, the differences in used cut-offs should be explained.

Response:

We thank the reviewer for raising this point. The categorized CNV modalities (CNV_AMP and CNV_DEL) were used as inputs in the CLIP pipeline, instead of the continuous data. The primary reason for such discrepancies between the datasets to call for copy number amplification/deletions is because the scaling of copy number intensities is rather different from one dataset to another. Moreover, like mentioned above in response to the Reviewer 2, comment 1, we wanted to use the originally-processed data from each study, rather than re-processing every dataset from raw data with our own processing pipeline that may be sub-optimal for a particular study (i.e., we trust the original authors of the studies know best how to preprocess their data). Notably, while datasets from BROAD, OHSU, UHN and NCI60 have been originally processed using the CBS algorithm, data from GDSC and gCSI have been

processed using the PICNIC algorithm. Based on the initial consistency analysis, we had previously selected the thresholds that maximized the overlap of amplifications and deletions between identical cell lines from different study sites.

Nevertheless, to make the approach more systematic and unbiased, we have now looked at the copy number distributions of two well-known CNV alterations in breast tumors; ERBB2 which is known to be frequent amplification, and PTEN which is known to be a frequent deletion in breast cancer patients (Koboldt *et al*, 2012). For ERBB2, we looked at the distribution of its copy number values across all cell lines in each dataset (Figure S1). We then selected the threshold for calling an amplification in each dataset at the value where we observed a sharp deflection. For BROAD, this analysis led to the threshold of ≥ 1 for calling amplifications. For, GDSC, gCSI and OHSU, values ≥ 0.5 ; and for UHN and NCI60 value ≥ 0.4 were considered amplifications. Likewise, for calling deletions; BROAD < 1 ; GDSC, gCSI and OHSU < 0.5 ; and UHN and NCI60 < 0.4 were selected based on this analysis. We further executed CLIP at various thresholds, and did not observe significant differences in its ability to identify breast cancer subtype-specific driver genes that we used for benchmarking the performance of CLIP.

We have added a new section describing the details of our rationale for the CNV threshold selection for the CLIP approach (page 12).

The difference in correlation between continuous CNV intensities as opposed to the binary copy number calls can be attributed to the differences of the metric used for estimating correlations, namely, Spearman and Matthew's correlations, respectively. We have often observed that correlation for binary metrics are lower, compared to the continuous measurements, due to the differences in the distributions of the binary and continuous variables. However, we would like to point out that the differences in the consistency do not affect the performance of CLIP for identifying breast cancer subtype specific driver genes. More specifically, even after evaluating different cut-off thresholds for calling CNV alterations, we did not observe any deterioration of the performance of CLIP (Figure S1C).

2. The gene-level methylation profiles were calculated by analyzing promoters of protein-coding genes, but there is a discrepancy in how promoters were defined for different data sets (e.g. 1 kb up-stream of TSS in case of the "BROAD" data-set and 1.5 kb in case of "GDSC"). The authors should either unify the promoter definitions or explain the differences.

Response:

We thank the reviewer for raising this point. The BROAD methylation dataset, which was based on reduced representation bisulfite sequencing, was made available by the authors of the study at the gene-level by averaging the CpG sites 1 kb up-stream of TSS. Because the genome wide CpG level data were not available, we were not able to change the criteria for defining the promoter in this dataset. In contrast, methylation datasets from all other studies were generated using Illumina platform microarrays and made available at probe intensity levels. Importantly, we used pre-annotated indices of CpG sites with the range of 200 to 1500 bp upstream of the TSS that were supplied by the manufacturer. Hence, we processed the array datasets using the closest approximations for defining promoter methylation, i.e. by averaging the beta intensities of CpG sites located within 200 to 1500 bp upstream of the TSS. We have modified the descriptions in the method section to accommodate this detail (page 5,6,8).

3. In order to identify genes carrying deleterious mutations, the authors combined predictions reported by multiple tools. It is a very sensible approach but for the sake of consistency, the same set of methods should be applied to all data sets. If, for any reason, it was not possible, the authors should explain why.

Response:

We thank the reviewer for raising this point. Since we used processed datasets, we chose to keep the annotations of functional predictions of mutations provided by authors of each study. This is because we wanted to make as minimal further processing of the data as possible to avoid biasing these data and the original results. The functional prediction annotations were provided for most of the datasets, except for UHN and OHSU. In order to be consistent, we have processed each dataset using the FATHMM prediction pipeline. We have modified the methods descriptions to accommodate this detail (pages 4,5,7,8, Figure S4).

4. I find the data acquisition and pre-processing steps to be very important for the manuscript and highly interesting for the scientific community. In order to make it easier for the readers to understand and evaluate the pipeline, I would suggest including one or multiple supplementary figures depicting in details all methods used for preparing the input data. One scheme for each modality. This would be very useful, especially because data sets provided by different research institutes were not processed identically. The flow charts could include names of the tools used, important parameters and counts of genes and cell lines acquired from each data source.

Response:

We thank the reviewer for the great suggestion. We have now provided details of the original authors' and our data processing and harmonization procedures for multi-omics data in the new Supplementary Figure S4. As mentioned above, we chose to use the originally-processed data as far as possible, since we believe the authors of the original studies know best how to process their data. We believe these steps in the supplementary figure will be useful for the community for extending the CLIP approach to other omics data profiles from emerging studies, as well as for developing other types of meta-analysis approaches that require integrating multi-omics original data from multiple studies. These new additions are described in the revised discussion (page 30).

5. Throughout the manuscript, the authors compare correlations between pairs of results obtained from identical cell lines to pairs of non-identical cell lines (Figure 1D). This approach is highly biased, considering that many non-identical cell lines derived from same tumor types are, in fact, very similar in their profiles. Splitting pairs of non-identical cell lines into two groups: non-identical but same-tissue (e.g. breast), and non-identical and different tissue would put the data consistency in a better perspective.

Response:

This is an interesting point raised by the reviewer. The non-identical cell line correlations were used as background distribution for the identical cell line correlations, as the number of genes and cell lines may have a drastic effect on the correlation coefficients. As requested by the reviewer, we have now further re-analyzed the correlation estimates categorized by tissue identity of non-identical cell lines. In general, for all the data modalities, we observed that the non-identical cell lines from identical tissues had slightly elevated correlation, compared to non-identical cell lines from non-identical tissues (please see new Figure S7A). However, the difference between the tissue identical and tissue non-identical cell lines was relatively small, compared to the difference between the identical and non-identical cell lines. Therefore, we

chose to use the original background distribution from non-identical cell lines, but have added the following brief statement about the analysis in the manuscript (page 21): “For all the data modalities, we observed that non-identical cell lines from same tissue types had slightly elevated correlation compared to non-identical cell lines from different tissues, but the opposite for the PEXP modality (Figure S7 A)”.

6. Page 11: The authors defined OES as a z-score, which is a difference between a raw value and a mean, divided by a standard deviation. This is not consistent with Figure 3, where the denominator is omitted. Either the figure should be corrected, or the z-score term should not be mentioned in the text. It also seems to me that OES could as well be defined as a percentile rank of a gene-specific raw value in a given cell line, among all cell lines in the same data set. This would make OES more robust against heavily skewed distributions.

Response:

We thank the reviewer for pointing out this mistake. We have now corrected the formula in Figure 3, and added the SD term on the denominator. We agree with the reviewer’s alternative definition for OES. However, we think that this would not change the downstream results since we perform a rank product analysis of the OES scores across multiple datasets (please see Methods, page 11).

7. Page 19 and Figure 3: The use of "down-regulated" and "up-regulated" terms is not suitable for many analyzed modalities, and may confuse the readers. For example, high methylation of a gene is, in fact, a sign of its down-regulation. Instead, I would rather suggest referring to CSSup and CSSdown as genes enriched at the top or the bottom of a set of ranked lists of genes.

Response:

We agree with the reviewer’s suggestion, and have now modified the text to accurately reflect the meaning we want to convey throughout the text (page 24 which was page 19 in the original manuscript). We modified the sentence as: “Next, for any given cell line, OES and PS scores of all the coding genes for a specific data modality from different studies were integrated to identify those genes that were consistently **at the top or the bottom of the ranked list of genes** in a given cell line, CCS_{UP} and CCS_{DOWN} . We have also modified all the figures accordingly.

8. Page 11: In case of cell lines profiled in a single study, rank product analysis was not performed but, instead, an arbitrary number of top scoring genes were selected. In case of almost all continuous modalities, 100 genes were picked. Except functional data sets, for which the top 200 genes were selected. Why the inconsistency? Is it because of noisiness of the gene dependency data? The reason should be mentioned in the Methods section.

Response:

We thank the reviewer for pointing this out. Indeed, we used the insights from the initial reproducibility analyses to inform our choices on setting the thresholds for selecting genes. This is now mentioned on pages 11 and 24. As we observed lower reproducibility between functional datasets, we used a more relaxed threshold to select the top CCS genes from this modality. We have now modified the statements in the Methods section to better inform the readers of our rationale (page 11).

9. Page 16: The authors claim that a high variability of CNV correlations between identical cell lines suggest clonal variability of cell lines cultured at different research sites. It would be very

interesting to see if such cell lines also show much weaker correlations of mutational profiles. In my opinion, this would make the statement stronger. Additionally, the authors should provide a table of these cell lines. It would be highly valuable for the scientific community.

Response:

This is a very interesting question. We interrogated our dataset to explore whether identical cell lines that had lower CNV correlation for continuous intensities between identical cell lines also had weaker correlation of mutational profiles. To do this, we first categorized cell lines into two groups: low CNV correlation (loCNV) [$r_{\text{CNV}} < 0.2$] and high CNV (hiCNV) [$r_{\text{CNV}} \geq 0.2$]. This cut-off was defined based on an inspection of the distribution of CNV correlation values of identical cell lines, with the rationale to identify cell lines towards the extreme of inconsistent CNV profiles. We then checked the distribution of correlations between mutational profiles in the same set of identical cell lines pairs. As was hypothesized, we found that loCNV cell lines have significantly weaker correlation of mutational profiles, compared to hiCNV cell lines. We have added the results of this analysis to the manuscript supplementary section (Figure S7B), and mentioned this result in the text (page 21). As suggested, we also provided the list of cell lines that were used in the analysis for the community (Table S9).

10. Page 17: Comparison of different technical platforms is a very interesting aspect of the manuscript. However, to be able to draw any conclusions about reproducibility of some technologies, more sets of correlation analyses are needed. For example: in case of phosphorylation profiles, correlations within MS-based profiles are missing. In case of drug sensitivity assays, it is a lack of within-Syto60 correlations (Supplementary Figure S4K and S4L). Similarly, the methylation studies based on bisulfite sequencing and Illumina 27K are missing their respective "within technology" correlations. I assume it is due to a lack of appropriate data sets, but the authors should make it clear for the reader. I would even suggest a significant reduction of the first two paragraphs and even removal of PHOS- and DSS-related statements from the main text.

Response:

We have now mentioned it explicitly in the Figure legends why these comparisons could not be made. We added the following text: "MS/MS-based comparisons could not be performed because MS-based profiling was performed by only one study...", and "Syto60/Syto60 comparisons could not be performed because Syto60 based profiling was performed by only one study" for Figure S8 legend. We also shortened the first two paragraphs as was suggested. The correlation of methylation profiles between Illumina 27K and bisulfite sequencing have been already provided in Figure S8J. However, we did not discuss this minor detail explicitly in the manuscript, as we feel there are other, more interesting results to point out from the correlation analyses. We think the results section is already relatively dense, so we chose not to add more results that might make the results unnecessarily complicated.

11. Page 18, line 19 and Figure 2E: Why these two studies were picked? Do they represent a typical correlation between coefficients of variation of NL and TMT profiles?

Response:

These two studies were picked because they were done exclusively on breast cancer cell lines, providing sufficient overlap between the two studies for robust estimation of the coefficient of variation (n=15). This is now clarified in the revised text of the legend of Figure 2E.

12. Page 19, line 34-38: The authors claim that some driver genes were selected based on TAS and FUNC modalities and others based on PHOS and CNV. However, judging from Figure 4A, it looks like almost all shown genes could be selected only by CSS scores from TAS and FUNC modalities. The only exception, AKT1 in MCF7 cell line, was selected by scores from PHOS and GEXP studies. Other modalities would not be needed for this hit calling. The authors should correct their conclusions or explain their point better.

Response:

A similar question was also raised by Reviewer #2, comment 8. To answer this point regarding bias in data modality in rCCS genes selected by CLIP, we systematically looked at the frequency of the data modality sources of rCCS evidence for each gene from the CLIP run of all the cancer cell lines (Figure 4B). Specifically, for each cell line, we counted the number of evidences from each data modality for each rCCS gene that was identified by the CLIP pipeline. Next, to get a relative estimate per modality we divided the number of rCCS genes for each cell line by the average of the number of genes from multiple datasets that were used in CLIP for each specific modality. This was done because the number of genes per modality varies because of the differences in the size of datasets, for instance, PHOS and TAS profiles are available for much lower number of proteins. We observed that the TAS and PHOS modality were the most frequent source of rCCS evidence (Figure 4B). We have now incorporated the results of these analyses in the Results section (page 24).

Figure 4A shows the results for driver kinases that are known to exhibit oncogenic addiction, and this analysis shows that the multi-omics CLIP can efficiently identify such oncogenic addictions of cancer cell line. We note that this example figure was based on a relatively small number of known driver kinases only. In response to the Reviewer 2, comment 8, we have now made much more systematic analyses of the importance of the single omics data to the identification of known breast cancer drivers. To assess how much each data modality contributed to the performance of CLIP, in addition to the base setting of CLIP in which all the modalities were included, we also re-ran the CLIP by removing each data modality one at a time, and measured the fraction of true positive driver genes identified after the removal of the input datasets (Figure S1D). Overall, we observed that the performance of CLIP was relatively robust, however, we observed a slight decrease after removal of the GEXP modality. These new results demonstrate the importance of using all the data modalities in the best performance of the CLIP approach. This is now mentioned in the revised Methods (Page 13) and discussion (page 30).

13. The authors should measure expression levels of ECHDC1 (e.g. with qPCR) in MCF10A and BT-474 cell lines, as well as knock-down efficiency.

Response:

To address the reviewer's concern with regards to expression and knock-down efficiency of ECHDC1, originally done using siRNA assay, we have performed a CRISPR knockout assay to study the functional relevance of ECHDC1. As suggested in the comment #14 below, we have performed the CRISPR knockout assay using three distinct single guide RNAs for ECHDC1 in BT474 and MCF10A cell lines, and confirmed the KO by Sanger sequencing. These new results are provided in Figure 5C and S13, and the methods and results are described in the revised manuscript (pages 16-17 and 26). We have excluded the original siRNA results from the revised manuscript, as we think the CRISPR-based results are of better quality.

14. *Microscopy images are not convincing enough to support the claimed proliferative and invasive phenotype of ECHDC1 knock-down. If possible, a time-course cell count quantification should be provided, or a cell competition assay. Considering reported poor reproducibility of RNAi screens (Figure 2B), the authors should validate the phenotype with an independent silencing trigger, or by analyzing a CRISPR knock-out.*

Response:

To address the reviewer's concerns, we have performed a CRISPR knockout assay of ECHDC1 in BT474 and MCF10A cell lines. In order to assess the growth, these KO cells were embedded in 3D collagen and the cell count and colony area was observed and quantified over a time-course of 120 hours. We recapitulate the same phenotype with this alternative loss-of-function method than was originally observed with siRNA knock-down (Figure 5C and S13). We have also provided better quality microscopy images from the experiments and provide the updated data and images as Figures 5C. The new experimental methods and results are described in the revised text (pages 16-17 and 25).

Minor points

1. *In general, the number formatting should be corrected to be uniform across the whole text (i.e. digit grouping and use of thousand separators). Also, at numerous places, a non-breaking space should be added between a number and a unit (e.g. page 5, line 8; page 13).*
2. *Page 3, line 7: The reference needs to be fixed.*
3. *Page 5, line 34: "Pathogenic" probably should be replaced by "deleterious".*
4. *Page 8, line 12: It is not clear to which "earlier section" the authors refer to. In previous sections, varying methods were used.*
5. *Page 9, line 32: "DSS2" should be spelled without a subscript.*
6. *Page 12, line 38 and page 13, line 5: Exponents at the cell count should be positive.*
7. *Page 14, line 12: The sentence ending with "from 1 or -1" needs to be corrected.*
8. *Page 15, line 37. The text says ~215 cell lines, but the Figure S1A shows over 700 cell lines analyzed by a single study site. The number should be corrected. This sentence also refers to Figure S2, which does not fit the text.*
9. *Page 22, line 41: The sentence refers to Figure S10E, but the figure panel does not match the text.*
10. *Figure 1: Please, unify the colors between panels. Differences in shades become especially apparent on printouts. References in panel A are not consistent with the journal style.*
11. *Figure 1B and Supplementary Figure S1A: I would suggest using black bar to avoid confusion with the DSS modality.*
12. *Figure 2A, 2C and Supplementary Figure S4J: Considering the group labeling, it would be easier to read the matrices if their lower triangles were shown.*
13. *Figure 2B and 2D: The figure legend should explain that triangles represent means.*
14. *Figure 3F: Either a gene name should be mentioned, or it should be made clear that this is just an example of how a CLIP signature may look like.*
15. *Figure 4A: Some cell line labels are not aligned with their respective columns.*
16. *Figure 4B, 4D, Figure 5A and Figure S10: The percent signs in brackets should be removed. They suggest that upper limits of Y-axes is 1%. Alternatively, multiply labels by 100.*
17. *Figure 5E: The legend refers to Figure S7, which does not show metabolite levels across 14 cell lines.*
18. *Figure S6: Descriptions of "columns" and "rows" in the figure legend should be swapped.*
19. *Table S2: Typo in the header ("each genes").*

Response:

We thank the reviewer for pointing out these minor errors. We have corrected them all.

Thank you for sending us your revised manuscript. We have now heard back from the three reviewers who were asked to evaluate your study. The three reviewers were those who also evaluated the initial version of your work. Overall, the reviewers think that the study has improved as a result of the performed revisions. However, as you will see below, reviewer #1 still raises several remaining concerns, which will need to be addressed.

During our pre-decision cross-commenting process in which the reviewers are given the chance to make additional comments, including on each other's reports, all three reviewers agreed that you should be given a chance to address these remaining issues. Specifically, reviewers #2 and #3 mentioned that points 4 and 5 of reviewer #1 are important to address and they would recommend that you follow the suggestions of reviewer #1 in this regard. In brief, to address point 4 reviewer #1 recommended: "The authors should either change their test for synthetic lethality to ensure that they obtain SL pairs, or change the text to indicate that they obtain interactions that may not be SL. In particular, the putative SL pair that they found using the current test, (PTEN low, DDX30 low), is not supported by the survival test in patient data Fig S18F, and this needs to be explained." Regarding point 5 they recommended "improving the survival analysis of ECHDC1 low expression and high methylation patients (Figure 5b) so that it a) takes very young age into account as a covariate, and b) is a proper analysis of ECHDC1 low expression and high methylation against average patients."

Regarding the readability of the manuscript, reviewer #2 mentioned "I agree with Reviewer #1 that the manuscript is long and not easy to read, but a careful reader will not have difficulties in understanding the pipeline. Nevertheless, the whole methods section "Meta-analysis and data integration framework CLIP" (p. 10-11) would benefit from re-writing. Initially, the authors partially describe CCS and rCCS in the context of continuous variables, and mention an OES score before it is properly defined (as noticed by the Reviewer #1). A proper definition of rCSS is not given until the last paragraph of the section. Besides, there's a massive redundancy between this methods section and a description of the pipeline in the results section "An analytical framework for meta-analysis and integration of multi-modal datasets" (p. 23). The manuscript would indeed benefit from reorganizing these two parts. Possibly by reducing the methods text to a bare minimum - I would consider the pipeline more a "result".

Regarding the documentation of the code reviewer #2 mentioned: "R is a widely spread and freely available programming environment and I consider it sufficient to have all data sets deposited in the RData format. Although CSV is human readable, it has its own challenges (even risks) and I would advise against using it. Alternatively, authors could use HDF5 as a programming language-independent format, but I do not think that it would be particularly useful, considering the complete pipeline code being written in R. Either way, it should not be difficult to implement, for example by using the hdf5r package. Finally, I agree that the deposited source code should be polished (e.g. by

fixing references to files located in a "Desktop" directory) and that all input tables should be annotated." In line with these recommendations, we would ask you to make sure that the code is documented in detail and in a format that is easily accessible to the readers and future users.

Taken together, we would like to offer you a chance to address the remaining issues raised by reviewer #1 in an exceptional second round of revisions. I hope the comments above are helpful, do let me know if there is anything you would like to discuss when preparing your revision.

On a more editorial level, we would like to ask you to address the following issues.

REFEREE REPORTS

Reviewer #1:

This manuscript reflects a great amount of work on the part of the authors. In part because it is a multi-omics manuscript, but also in part due to the style in which it was written, the length of the manuscript is daunting. Our response will focus largely on whether the authors sufficiently addressed our comments. Regrettably, aside from comment 3, the authors did not. Unfortunately, some of the changes have also either created or revealed further problems with the manuscript. Therefore, in the current state, we do not recommend publication in MSB.

The authors have clearly invested significant amount of work following the revision. However, it is evident that the manuscript has gained considerably in length. Before moving on to address the comments the authors are encouraged to shorten the manuscript. This would greatly improve the reading experience. They have already created a large table in supplementary showing what the how different datasets were handled, which is a good start. Continuing this process by organizing

common themes in the main manuscript and moving the content of several sections would be very helpful.

The following comment about the contribution of this manuscript remains relevant: "Aside from the identification of ECHDC1, the most useful contribution of the study to the cancer community would be full access to the meta-analysis results, and ideally both in the form of full results, and with a release of the pipeline code." The changes the authors made with respect to the output data are insufficient to make the data accessible to most readers. The output files remain RData files with undocumented contents. Without reasonable access to the processed data, the project is of less practical use and therefore of less widespread interest. Some access to the data has been provided as supplemental tables, but it is not evident that the supplemental tables entirely duplicate the processed data.

Whether code must be released in a format that can be reused in different contexts by others is up to the journal. But the work would be more interesting if readers could easily change the sites and/or data modalities used, possibly adding different dataset or modalities and changing parameters. The format of the input files is not specified, steps are missing from the documentation (replaced by '..'), and there is no discussion on how one might add or change datasets or adjust any parameters in the pipeline. The ideas of pipeline are interesting, but currently one would need to rewrite or reverse-engineer the code to put them in practice. A more polished code release would strengthen the work.

Previous comment 1) Reproducibility: this appears to be partially addressed with the additional code that the authors added to GitHub. However, the authors' argument about the github file upload size limitations is unconvincing. First, the authors uploaded a file under /data/GEXP.RData and made it available for download using Git LFS which is 283MB. It seems to be clearly possible to provide data resources large than 100MB. In addition, the output files, as opposed to the input files, are not large and while the data would be somewhat larger in csv, and should not approach a data limit, particularly if the authors make use of compression. The authors are encouraged to provide preprocessed data resources in a format that is independent of a specific programming language.

Previous comment 2) Methods are vague: the new text that the authors quoted in their p2p reply is on page 11. The quoted text remains vague, in that it says "to identify enriched genes" in a location at which "genes" might be rCCS genes or CCS genes. The vagueness of the Methods is more widespread and could be fixed by careful attention to the writing. Take for instance this quote from lines 35-37, "Specifically, we estimate the CCS evidence of a gene using its deviation from the mean over a panel of cell lines, also called as Outlier Evidence Score (OES)." There are three obvious interpretations of this sentence: the OES is the square root of the variance over relevant cell lines, the OES is the mean absolute deviation, or the OES is the difference of the numerical of the gene in a specific cell line and the mean over cell lines. In fact, Figure 3 shows the OES as the Z-score, which is none of these.

Previous comment 3) CLIP performs better than methylation alone: In principle the authors' response is fine. They show that CLIP is more specific than using individual methylation data and that different sites are inconsistent about finding ECHDC1 as a top hit. That answers the question.

Previous comment 4) Synthetic lethality: The authors added text on page 14, lines 24-26 that suggested they used Fisher's exact test to determine interaction partners, and then define SL pairs as only those genes that appear on a list of essential genes. This is test for interactions, but not the proper test for synthetic lethality. Some aspect of this method needs to be changed; either by using a proper test for synthetic lethality used or dropping claims of synthetic lethality. Independent of that, the authors took the positive step of performing the analysis suggested in the review. However, the analysis could be improved even further. Instead of dichotomizing the expression data for each gene, it would be preferable to tertile the expression data (low, mid, high) so as to not make fine distinctions among genes whose expression is near the mean/median. Next, the patients should be stratified into those with an 'active' SL interaction (PTEN low, DDX27 low) within the same patient and those where this interaction is not active NOT (PTEN low, DDX27 low). By introducing a more fine-grade distinction of low and high expression the authors will likely improve the signal for gene DDX27. Lastly, it should be highlighted that the above analysis is again only an indirect evidence of synthetic lethality between two genes. Finally, the authors' suggestion that (PTEN low, DDX27 low) patients have worse survival is inconsistent with synthetic lethality, so some aspect of their methods appears to be leading to inconsistent results and does not seem to be supported by patient data. The authors should mention this limitation in the discussion.

Previous comment 5) Survival analysis: Is worse because a) They do not control for age in a way that accounts for the fact that pre-menopausal cases of breast cancer are different. They introduce a control variable that splits the cohort into under 60 and over 60, which is insufficient. One might discretize more finely to account for those under 45, or just use numerical age in Cox regression. b) Their comparison is not of practical utility because it does not include all persons. One can say that patients with ECHDC1 low expression and high methylation fair worse than those with ECHDC1 high expression and ECHDC1 low methylation. They cannot say if a person with ECHDC1 low expression and high methylation has a worse prognosis than an average patient. A plausible answer is partition patients into two groups (similar as above): those with ECHDC1 low expression and high methylation and those that do not meet these criteria.

Reviewer #2:

The authors have performed an excellent job while revising their original submission and address mine and other reviewers' points satisfactorily. In response to my previous comments, beside addressing the lack of clarity of few aspects and addressing all the minor requested changes, they have stressed the utility of the preliminary comparative analysis, added results from a comprehensive benchmarking effort contrasting their outcomes with those attainable while analysing single omics data and/or an existing tool such as MOFA+, and also robustly assessed the ability of their method in predicting cancer driver genes and dependencies. With respect to this last point, the author mentioned the lack of a resource containing established

oncogenetic additions. For this reason, to conduct their analysis anyway, they have manually assembled a set of positive cases derived from a published work. Optionally, they might consider re-performing this analysis deriving the oncogenetic additions to be used as gold-standard from <https://www.oncokb.org/>, if this increases the number of considered positives.

Reviewer #3:

The authors have revised and significantly improved the manuscript, addressing all points raised in the previous revision. Their approach to data analysis and presented results should be of interest to the readers of MSB.

Reviewer #1:

Response to the Authors

The authors have clearly invested significant amount of work following the revision. However, it is evident that the manuscript has gained considerably in length. Before moving on to address the comments the authors are encouraged to shorten the manuscript. This would greatly improve the reading experience. They have already created a large table in supplementary showing what the how different datasets were handled, which is a good start. Continuing this process by organizing common themes in the main manuscript and moving the content of several sections would be very helpful.

Response: We have now shortened the Methods section, as well as reorganized and moved most of the CLIP methods text to Results to remove the redundancies as suggested (please see below)

The following comment about the contribution of this manuscript remains relevant: "Aside from the identification of ECHDC1, the most useful contribution of the study to the cancer community would be full access to the meta-analysis results, and ideally both in the form of full results, and with a release of the pipeline code." The changes the authors made with respect to the output data are insufficient to make the data accessible to most readers. The output files remain RData files with undocumented contents. Without reasonable access to the processed data, the project is of less practical use and therefore of less widespread interest. Some access to the data has been provided as supplemental tables, but it is not evident that the supplemental tables entirely duplicate the processed data.

Response: We thank the Reviewer for these recommendations and agree that these points are important for open science. We have now provided the processed datasets also in the h5 format, as instructed by the Reviewer #2. All the study data can be downloaded from figshare platform: <https://doi.org/10.6084/m9.figshare.13473168>. We also provide now annotations for data tables that were utilized in the study.

Whether code must be released in a format that can be reused in different contexts by others is up to the journal. But the work would be more interesting if readers could easily change the sites and/or data modalities used, possibly adding different dataset or modalities and changing parameters. The format of the input files is not specified, steps are missing from the documentation (replaced by '..'), and there is no discussion on how one might add or change datasets or adjust any parameters in the pipeline. The ideas of pipeline are interesting, but currently one would need to rewrite or reverse-engineer the code to put them in practice. A more polished code release would strengthen the work.

Response: We thank the Reviewer for the suggestions. We corrected the file referencing locations in the code, and have provided a detailed documentation of the code by adding commented lines to the R scripts. We have also provided annotations for tables that were utilized in the study to show how and where additional data modalities should be added for future use. Additionally, we also added the descriptions of the format of input parameters for each function utilized in the CLIP pipeline. We hope that with these changes, the end users can easily follow the code and the steps in the pipeline.

Previous comment 1) Reproducibility: this appears to be partially addressed with the additional code that the authors added to GitHub. However, the authors' argument about the github file upload size limitations is unconvincing. First, the authors uploaded a file under /data/GEXP.RData and made it available for download using Git LFS which is 283MB. It seems to be clearly possible to provide data resources large than 100MB. In addition, the output files, as opposed to the input files, are not large and while the data would be somewhat larger in csv, and should not approach a data limit, particularly if the authors make use of compression. The authors are encouraged to provide preprocessed data resources in a format that is independent of a specific programming language.

Response: We have now provided the processed datasets in the h5 format as suggested by the Reviewer #2. Instead of uploading the data to GitHub, which has size restrictions, we now provide the processed datasets in the figshare platform. All the study data can be downloaded from: <https://doi.org/10.6084/m9.figshare.13473168>. We removed the RData files uploaded in the previous version to avoid confusion of having data files in multiple formats. We also added text in the Github Readme file describing the path for keeping the downloaded files before processing, and also modified the scripts for processing the h5 data files.

Previous comment 2) Methods are vague: the new text that the authors quoted in their p2p reply is on page 11. The quoted text remains vague, in that it says "to identify enriched genes" in a location at which "genes" might be rCCS genes or CCS genes. The vagueness of the Methods is more widespread and could be fixed by careful attention to the writing. Take for instance this quote from lines 35-37, "Specifically, we estimate the CCS evidence of a gene using its deviation from the mean over a panel of cell lines, also called as Outlier Evidence Score (OES)." There are three obvious interpretations of this sentence: the OES is the square root of the variance over relevant cell lines, the OES is the mean absolute deviation, or the OES is the difference of the numerical of the gene in a specific cell line and the mean over cell lines. In fact, Figure 3 shows the OES as the Z-score, which is none of these.

Response: We thank the reviewers for pointing out these concerns with the readability of the CLIP methods section. As suggested, we have now reorganized the text in the Methods and Results sections to more accurately describe the CLIP pipeline, as well as shortened the Methods section and moved most of the text to Results to remove the redundancies (pages 12 and 24-25). We have made our best effort to make the revised text free of any ambiguity for the reader.

Previous comment 4) Synthetic lethality: The authors added text on page 14, lines 24-26 that suggested they used Fisher's exact test to determine interaction partners, and then define SL pairs as only those genes that appear on a list of essential genes. This is test for interactions, but not the proper test for synthetic lethality. Some aspect of this method needs to be changed; either by using a proper test for synthetic lethality used or dropping claims of synthetic lethality.

Response: We thank the Reviewer for this comment, and apologize for the lack of clarity of the text in the Methods section relevant to this topic. We have now modified the text to accurately reflect our methodology (page 15-16). Briefly, we termed SL genes as those rCCS genes that are obligatorily identified as a CCS gene in the FUNC modality, i.e., essential genes, which are also differentially enriched in the mutated cancer cell lines only. Based on our terminology, the test for synthetic lethality assesses differential dependencies of genes between mutated samples vs. wild type samples, i.e., genes that are uniquely more essential in the mutated samples. We think our approach matches well with the traditional "dead vs. alive" phenotypic

classification of SL interactions, compared to the quantitative analyses of gene dependencies used in more broad classification of genetic interactions (from negative SL interaction to positive ones). More specifically, in our analyses, rCCS genes are also binary 0 or 1, where 1 indicates a gene that is both essential to that cell line (based on the CCS evidence from FUNC modality), and was also identified by another data modality as a CCS gene (to provide multi-modal support for rCCS gene). However, we agree that one could design also other types of statistical testing of SL interactions based on quantitative phenotypic data, which is now also mentioned in the revised version (page 28).

Independent of that, the authors took the positive step of performing the analysis suggested in the review. However, the analysis could be improved even further. Instead of dichotomizing the expression data for each gene, it would be preferable to tercile the expression data (low, mid, high) so as to not make fine distinctions among genes whose expression is near the mean/median. Next, the patients should be stratified into those with an 'active' SL interaction (PTEN low, DDX27 low) within the same patient and those where this interaction is not active NOT(PTEN low, DDX27 low). By introducing a more fine-grade distinction of low and high expression the authors will likely improve the signal for gene DDX27. Lastly, it should be highlighted that the above analysis is again only an indirect evidence of synthetic lethality between two genes.

Response: We agree that similar to the cell line profiling data (see our above response), also the patient profiling data can be utilized in many ways when providing support for the SL interactions identified using the cell line profiling data. The additional limitation with the patient data also comes from rather limited smaller sizes of patients with the particular CLIP signature (e.g., "PTEN low, DDX27 low"), which make statistical testing often underpowered in the patient data. This is the reason why we originally chose to dichotomize the expression and other quantitative data when analyzing patient molecular profiles. When performing the more fine-graded stratification of low and high expression of DDX27, as suggested by the Reviewer, the sample size of the 'active' group become smaller as we remove the patients whose expression is near the mean. We have now discussed the limitations of the patient data in the revised manuscript (page 32).

Finally, the authors' suggestion that (PTEN low, DDX27 low) patients have worse survival is inconsistent with synthetic lethality, so some aspect of their methods appears to be leading to inconsistent results and does not seem to be supported by patient data. The authors should mention this limitation in the discussion.

Response:

We assume that the Reviewer meant to write "PTEN low, DHX30 low" instead of "PTEN low, DDX27 low", since our previous data rather demonstrate that patients with "PTEN low, DDX27 low" CLIP signature have actually better survival (Appendix Figure S18E), which is consistent with the SL interaction. However, this was not true for the "PTEN low, DHX30 low" patients (Appendix Figure S19B). As suggested, we have now indicated that some of the interactions identified by the statistical analysis of the cell line data may not turn out to be clinically relevant SL interactions (page 32). Specifically, we now discuss the results on DHX30 in the revised Discussion section, pointing out this potential limitation of the CLIP pipeline (page 32). We have also modified the text in the Results section accordingly (pages 28-29). We agree that some of the identified pairs are not supported by the survival analyses in the patient data, but this may be also due to limitations of the current patient cohorts. This was also evident for the ECHDC1 finding (see below), for which the TCGA breast cancer patient cohort was not enough for demonstrating statistically significant survival effects, but we had to use the Oslo breast cancer

patient cohort with better follow-up data and clinical annotations to validate the clinical relevance of ECHDC1, after making detailed cell line validations using metabolite assays and CRISPR-Cas9 knock-out assays.

Previous comment 5) Survival analysis: Is worse because a) They do not control for age in a way that accounts for the fact that pre-menopausal cases of breast cancer are different. They introduce a control variable that splits the cohort into under 60 and over 60, which is insufficient. One might discretize more finely to account for those under 45, or just use numerical age in Cox regression. b) Their comparison is not of practical utility because it does not include all persons. One can say that patients with ECHDC1 low expression and high methylation fair worse than those with ECHDC1 high expression and ECHDC1 low methylation. They cannot say if a person with ECHDC1 low expression and high methylation has a worse prognosis than an average patient. A plausible answer is partition patients into two groups (similar as above): those with ECHDC1 low expression and high methylation and those that do not meet these criteria.

Response: We have now repeated the survival analyses as suggested. We grouped the breast cancer patients into two groups based on gene expression or promoter methylation of ECHDC1. To capture the effect of loss of tumor suppressor function, we identified the patients with either low expression or high methylation by selecting the patient tertile (one third of the patients) with the lowest expression or highest methylation (low GEXP), which was compared to the rest of the patients (non-low GEXP). Subsequently, we performed survival analysis on the two groups (log-rank test), and in a multivariate Cox model we also included age as a continuous covariate, as was suggested. We obtained similar results as reported previously (Figure 5B). In addition to showing the new survival results in modified Figure 5B, we have now also updated the text in the methods section accordingly (page 17). We would also like to mention that we do not claim that ECHDC1 should be used directly as a biomarker with prognostic utility. Instead, what we argue in the manuscript is that the association of ECHDC1 expression and methylation with patient survival supports tumor-suppressive role in breast cancer.

Reviewer #2:

The authors have performed an excellent job while revising their original submission and address mine and other reviewers' points satisfactorily. In response to my previous comments, beside addressing the lack of clarity of few aspects and addressing all the minor requested changes, they have stressed the utility of the preliminary comparative analysis, added results from a comprehensive benchmarking effort contrasting their outcomes with those attainable while analysing single omics data and/or an existing tool such as MOFA+, and also robustly assessed the ability of their method in predicting cancer driver genes and dependencies.

With respect to this last point, the author mentioned the lack of a resource containing established oncogenetic addictions. For this reason, to conduct their analysis anyway, they have manually assembled a set of positive cases derived from a published work. Optionally, they might consider re-performing this analysis deriving the oncogenetic addictions to be used as gold-standard from <https://www.oncokb.org/>, if this increases the number of considered positives.

Response: We thank the Reviewer for appreciating the efforts we made when addressing the original comments. We are happy to hear the Reviewer found the revised version already acceptable.

With respect to this last point, we repeated the analysis with the set of oncogenetic additions suggested by the Reviewer. We observed only a slight improvement in the ability of CLIP to identify more oncogenes and these results did not differ drastically from the previous results. Hence, we have persisted with the results shown in the previous version.

Reviewer #3:

The authors have revised and significantly improved the manuscript, addressing all points raised in the previous revision. Their approach to data analysis and presented results should be of interest to the readers of MSB.

Response: We thank the Reviewer for appreciating the efforts we made when addressing the original comments. We are happy to hear the Reviewer found the revised version already acceptable.

Thank you for sending us your revised manuscript. We have now heard back from reviewer #1 who was asked to evaluate your manuscript. As you will see below, they are now satisfied with the modifications made and are supportive of publication. They only raise a minor issue which we would ask you to address in a minor revision.

We would also ask you to address the following editorial issues listed below.

REFEREE REPORTS

Reviewer #1:

The authors have addressed all our comments to our satisfaction. We recommend publishing it in Molecular Systems Biology.

We believe there is a small inconsistency, at the level of a typo, between the threshold for the FUNC dataset: in the paper it is 0.20 and on the GitHub page it is 0.25. Either number is fine, but it should be resolved in favor of data presented in the manuscript.

Reviewer #1:

Response to the Authors

The authors have addressed all our comments to our satisfaction. We recommend publishing it in *Molecular Systems Biology*.

We believe there is a small inconsistency, at the level of a typo, between the threshold for the FUNC dataset: in the paper it is 0.20 and on the GitHub page it is 0.25. Either number is fine, but it should be resolved in favor of data presented in the manuscript.

Response: We are glad that the Reviewer is content with our responses. We appreciate the Reviewer for pointing out the typo in our manuscript, which was supposed to be 0.25. We have now made the required change in the manuscript (page 12).

Thank you again for sending us your revised manuscript . We are now satisfied with the modifications made and I am pleased to inform you that your paper has been accepted for publication.

Corresponding Author Name: Alok Jaiswal and Tero Aittokallio

Manuscript Number: MSB-20-9526R-Q